# Clustering wind profile shapes to estimate airborne wind energy production

Mark Schelbergen[1], Peter C. Kalverla[2], Roland Schmehl[1], and Simon J. Watson[1]

[1]Faculty of Aerospace Engineering, Delft University of Technology, Kluyverweg 1, 2629 HS Delft, The Netherlands
[2]Meteorology and Air Quality Section, Wageningen University, PO box 47, 6700 AA Wageningen, The Netherlands

**Correspondence:** Mark Schelbergen (m.schelbergen@tudelft.nl)

**Abstract.** Airborne wind energy (AWE) systems harness energy at heights beyond the reach of tower-based wind turbines. To estimate the annual energy production (AEP), measured or modelled wind speed statistics close to the ground are commonly extrapolated to higher altitudes, introducing substantial uncertainties. This study proposes a clustering procedure for obtaining wind statistics for an extended height range from modelled datasets that include the variation of the wind speed and direction with height. K-means clustering is used to identify a set of wind profile shapes that characterise the wind resource. The methodology is demonstrated using the Dutch Offshore Wind Atlas for the locations of the met masts IJmuiden and Cabauw, $85\,\mathrm{km}$ off the Dutch coast in the North Sea and in the centre of the Netherlands, respectively. The cluster-mean wind profile shapes and the corresponding temporal cycles, wind properties, and atmospheric stability are in good agreement with literature. Finally, it is demonstrated how a set of wind profile shapes is used to estimate the AEP of a small-scale pumping AWE system located at Cabauw, which requires the derivation of a separate power curve for each wind profile shape. Studying the relationship between the estimated AEP and the number of site-specific clusters used for the calculation shows that the difference in AEP relative to the converged value is less than three percent for four or more clusters.

## 1 Introduction

Airborne wind energy (AWE) systems employ tethered flying devices to harness energy above the operational height range of tower-based wind turbines. Typically these devices operate above $150\,\mathrm{m}$ (Malz et al., 2019; Salma et al., 2019), where wind is generally stronger and more persistent than in the surface layer. To estimate the annual energy production (AEP), measured or modelled wind speed statistics close to the ground are commonly extrapolated to higher altitudes to obtain the wind speed statistics in the full operational height range of the AWE system using either the wind profile power law or the logarithmic profile (e.g., Heilmann and Houle, 2013). This way of representing the wind resource introduces substantial uncertainties since the aforementioned wind profile relationships are not strictly valid beyond the surface layer. Moreover, within this layer, not all wind profiles can be described well with these relationships.

The power law is a simple empirical relationship which can be used to relate the wind speed $v$ at one height $z_1$ to that at a different height $z_2$ and has the form:

$$v(z_2) = v(z_1) \left( \frac{z_2}{z_1} \right)^{\alpha} \quad , \tag{1}$$

where $\alpha$ is an empirical shear exponent factor related to the surface properties. The power law is normally applied up to around 100–200 m (Peterson and Hennessey, 1978), and does not offer enough flexibility to describe the variety of measured wind profiles (e.g., Park et al., 2014).

The logarithmic wind profile is frequently used to estimate the variation in wind speed with height over a flat surface. This profile is based on physical arguments and a form of the profile has been well established based on Monin-Obukhov similarity theory (Monin and Obukhov, 1954). In this non-adiabatic form, the mean wind speed $v$ at height $z$ is given by:

$$v(z) = \frac{v_*}{\kappa} \left[ \ln\left(\frac{z}{z_0}\right) - \Psi\left(\frac{z}{L}\right) \right] \quad , \tag{2}$$

in which $v_*$ is the friction velocity, $\kappa$ is the von Karman constant, $z_0$ is the roughness length, $\Psi$ is a stability correction function, and $L$ is the Obukhov length that is often used to evaluate atmospheric stability and is positive (negative) for stable (unstable) stratification and infinite for neutral stratification. Holtslag et al. (2014) proposes the following stability correction functions:

$$\Psi(L \leq 0) = 2\ln\left(\frac{1+x}{2}\right) + \ln\left(\frac{1+x^2}{2}\right) - 2\arctan(x) + \frac{\pi}{2}, \qquad x = \left(1 - 19.3\frac{z}{L}\right)^{\frac{1}{4}} \quad , \tag{3}$$

$$\Psi(L \geq 0) = -6.0\frac{z}{L} \quad . \tag{4}$$

Although the logarithmic wind profile is less accurate under stable stratification above the surface layer (e.g., Optis et al., 2014), the relationship is often applied to any condition in wind resource estimation.

The value of $L$ is not easily measured or derived from model data and is generally inferred indirectly. One way to do this is to fit a functional form of the logarithmic wind profile with stability correction to the wind velocity magnitude profile. Such an approach is outlined by Basu (2018), using three levels of wind speed. Another common way of estimating $L$, is by inferring it from the gradient Richardson number, $\text{Ri}_\text{G}$. We approximate this number using a finite difference, yielding the bulk Richardson number, $\text{Ri}_\text{B}$, which expresses the ratio between the temperature stratification and the wind shear:

$$\text{Ri}_\text{B} = \frac{g}{\overline{\theta}_\nu} \frac{\Delta\theta_\nu \Delta z}{\Delta v^2} \quad , \tag{5}$$

in which $g$ is the gravitational acceleration, $\overline{\theta}_\nu$ is the mean virtual potential temperature, and $\Delta\theta_\nu$ and $\Delta v$ are the virtual potential temperature and the horizontal wind speed difference, respectively, determined over the height difference $\Delta z$. Positive (negative) $\text{Ri}_\text{B}$ values indicate stable (unstable) stratification and values close to zero indicate neutral stratification. By assuming a functional form of the stability correction, $L$ can be derived from $\text{Ri}_\text{B}$ (Holtslag et al., 2014):

$$\frac{\overline{z}}{L} = \begin{cases} \frac{\text{Ri}_\text{B}}{1 - 5\text{Ri}_\text{B}}, & \text{if } \text{Ri}_\text{B} \geq 0 \\ \text{Ri}_\text{B}, & \text{otherwise} \end{cases} \quad , \tag{6}$$

in which $\overline{z}$ is a reference height which is commonly taken as either the arithmetic or geometric mean of the heights used to determine the temperature and wind speed differences.

The wind direction can vary substantially with height in the lower atmosphere (e.g., Brown et al., 2005; Floors et al., 2015). A limitation of both the power law and logarithmic profile is that they provide no information about any wind direction

dependence with height. In addition, the relationships assume that wind speed increases monotonically with height. In practice, low-level maxima in wind speed, with decreasing wind speed above (low-level jets), are likely to occur, which is also observed in reanalysis data (e.g., Ranjha et al., 2013; Kalverla et al., 2019). To extend the validity of wind profile relationships to higher altitudes, several modifications have been proposed (e.g., Gryning et al., 2007; Holtslag et al., 2017). However, these theoretical

formulations are only validated up to heights relevant for conventional, tower-based wind turbines.

Alternatively, computationally expensive brute-force energy production calculations do not assume any wind profile relationship and are performed using historical wind data for the full operational height range. Bechtle et al. (2019) use ERA5 reanalysis data to map out the wind resource available to AWE systems over a large part of Europe, but do not touch upon the respective power production of an AWE system. Ranneberg et al. (2018) combine COSMO-DE reanalysis data with power

curves for multiple heights, that are independent of the wind profile shape, to estimate the AEP. This is a valid approach if the system is operating at a nearly constant height. However, the wind profile shape has to be considered if the system operates in a larger height range, as is the case for a flexible-kite AWE system (Van der Vlugt et al., 2019). AEP calculations become more computationally expensive if the wind profile shape is considered, especially when identifying the optimal cycle settings for all time points. Malz et al. (2020a) use three months of three-hourly MERRA-2 reanalysis data and speed up the computation

by a factor 20 by using the solution of the previous optimisation to initialise the next. In a follow-up study, Malz et al. (2020b) use this approach to determine the AEP of an AWE system for 16 locations in Europe. The current state-of-the-art is lacking a methodology that can be confidently used to make efficient AEP calculations for a pumping AWE system that sweeps a non-negligible height range.

Previously, clustering techniques have been used for identifying wind profile patterns. Sommerfeld et al. (2019) applies k-

means clustering to subdivide stable and unstable wind profile datasets from lidar observations into two clusters for a location in a mostly flat area in northern Germany. Duran et al. (2019) use self-organising maps to characterise wind profile data for two locations (Cabauw in the centre of the Netherlands and the FINO-1 platform in the North Sea, 45 km north of the German/Dutch coast) from Weather Research and Forecasting modelled data using 2300 clusters. The clusters are used for forecast verification and to investigate diurnal and seasonal cycles.

This study proposes a clustering procedure for obtaining representative wind profile shapes from measured or modelled data that include the vertical variation of the wind speed and direction. The data is partitioned into a small number of clusters and the corresponding cluster-mean wind profile shapes are determined. We have chosen an empirical approach for identifying these shapes such that they are not restricted by physical assumptions. Nevertheless, we try to physically interpret the observed features. In contrast to earlier studies that use clustering, we investigate normalised wind profiles, as these are often described

by wind profile relationships and yield a more compact wind resource representation. Moreover, the variation of the wind direction with height is included as it affects the operation of an AWE system.

The following sections of this paper outline the process of making an efficient AEP estimation for an AWE system based on historical wind data. Section 2 introduces the Dutch Offshore Wind Atlas (DOWA) and ERA5 datasets. Section 3 discusses the data processing and clustering techniques, complemented by interim results. Section 4 first addresses the clustering of DOWA

data and presents the results for an on- and offshore location. Subsequently, the DOWA data of 45 other locations is clustered

altogether to generate a generalised set of wind profile shapes that is applicable for an area which includes a wide range of location types. Although the resulting wind resource representation can be used for other applications, we illustrate its use for estimating the AEP of pumping AWE systems. Section 5 demonstrates the AEP estimation for a flexible-kite AWE system and assesses how many wind profile shape clusters are required for an accurate estimation. Finally, Sect. 6 summarises the conclusions of this study.

## 2   Wind datasets

In principle, any dataset containing time series of wind speeds and directions for multiple altitudes can be used as input for the proposed methodology. For the AEP calculation, we focus on the sensitivity of the AWE system power production to the wind profile, which is assumed to be non-varying in the calculation. An hourly temporal resolution of the datasets suffices for capturing the diurnal cycle of the wind profile. While smaller scale atmospheric phenomena might have an adverse effect on the power production, these effects can be superimposed on a mean time-invariant wind profile using separate models for assessing, e.g., the associated loss in power production (Fechner, 2016). This power loss is device-specific and depends on control strategy, and is therefore not considered here. The first commercial AWE initiatives envisage a maximum operational height of 500 m because operation at higher altitudes requires more complex system designs (Watson et al., 2019, p. 4) and legislative procedures (Salma et al., 2018). For the wind resource representation for AWE, it is thus desirable to have wind data at least up to this height. The vertical resolution should be adequate to assess the shape of the wind profile with sufficient detail for the performance calculations. Both long-term lidar observations and modelled data qualify as input. This study focuses on using modelled data, which provides good spatial and temporal coverage.

An on- and offshore location in the Netherlands and the North Sea, respectively, are selected for demonstrating the methodology. The offshore location, that of the met mast IJmuiden, is located 85 km off the Dutch coast in the North Sea. The onshore location, namely, the met mast Cabauw, is located in the centre of the Netherlands. The area directly surrounding the mast is flat open grassland for at least 400 m in all directions and up to 2 km in the dominant wind direction, i.e. west-south-west. Furthermore, within a radius of 20 km, the terrain is predominantly grassland and virtually flat. The met mast sites, shown in Fig. 1, are selected because they are well-known in the literature. We do not use the anemometer or lidar measurements of the met masts in this study. The other 45 depicted locations are used to evaluate the full DOWA domain and are selected such that onshore, coastal, and offshore locations are equally represented. The datasets for the met masts Cabauw and IJmuiden and the 45 locations are referred to as the onshore, offshore, and multi-location datasets, respectively.

### 2.1   ERA5

ERA5 (Copernicus Climate Change Service, 2017) is a global reanalysis produced by the European Centre for Medium-Range Weather Forecasts (ECMWF) using their atmospheric model and data assimilation system. At the time of writing, ERA5 data is available from 1979 to the present time. The data includes hourly modelled values of a large number of atmospheric variables on a 30 km horizontal grid with 137 vertical pressure levels up to a height of roughly 80 km. The level heights are

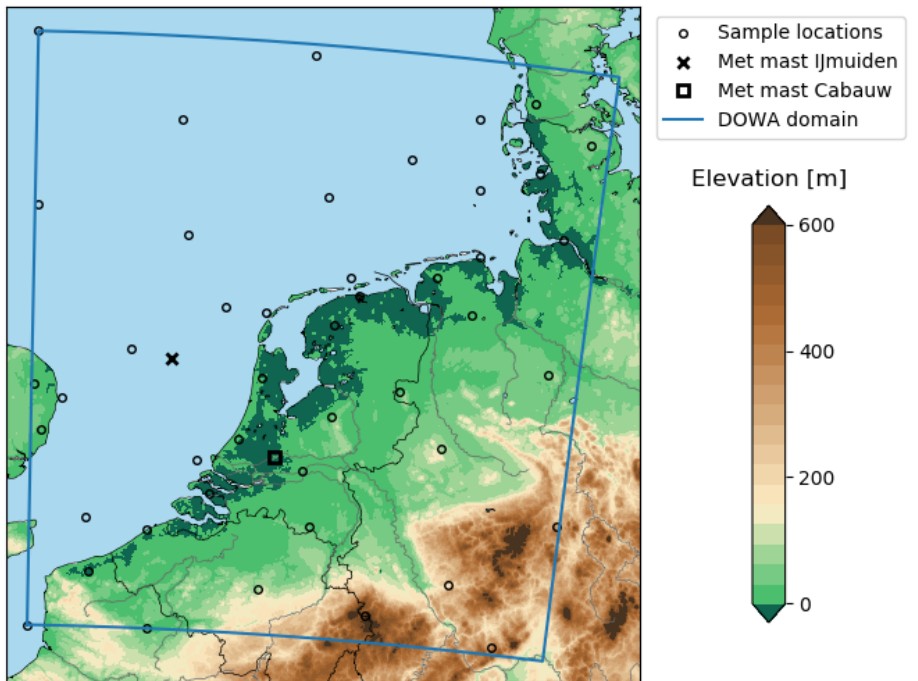

**Figure 1.** The DOWA domain, framed by the blue line, covers the Netherlands, a substantial part of the North Sea, and adjoining coastal areas. The ×, □, and ○ markers depict the locations analysed in Sect. 4.1, 4.3, and 4.4, respectively. The sea is depicted in light blue and the colour scale shows the elevation of the land surface (Amante and Eakins, 2009).

time dependent and interpolation is needed to obtain the wind data for fixed heights as required by the presented methodology. The clustering of the wind profile shapes is performed on the DOWA data, which is described in the following section. ERA5 is only used to determine the atmospheric stability at the time and location of the analysed wind profiles.

## 2.2 Dutch Offshore Wind Atlas

5  DOWA (Wijnant et al., 2019) is produced by the Royal Netherlands Meteorological Institute (KNMI) by downscaling ERA5 data to a finer-resolution surface grid using their mesoscale weather model HARMONIE-AROME (Bengtsson et al., 2017). The downscaled reanalysis is performed for 10 years from 2008 until 2017. Hourly values for temperature, wind speed and direction, pressure and relative humidity are made available on a 217 x 234 grid with 2.5 km spacing and 17 heights between 10 and 600 m. The DOWA domain is illustrated in Fig. 1. Due to the higher resolution and non-hydrostatic nature

10  of HARMONIE-AROME, DOWA benefits from an improved representation of the coastline, land surface heterogeneity, and mesoscale circulations, such as the sea breeze. Furthermore, additional observations from the KNMI's network of automated weather stations, satellite retrievals (ASCAT), and aircraft sensors (MODE-S EHS) have been assimilated by the HARMONIE-

AROME model. Kalverla (2019) shows that DOWA improves on ERA5 in terms of wind speed, wind shear, and directional accuracy, as well as the representation of anomalous events such as low-level jets.

## 3 Clustering procedure

This section illustrates the clustering procedure for the offshore location. The data is filtered and normalised and its dimensions are reduced using a principal component (PC) analysis. Next, the clustering performance is analysed and the number of clusters is chosen for the wind resource representations analysed in Sec. 4.

### 3.1 Preprocessing of the wind data

The operation of an AWE system is affected by the variation of wind speed and direction with height. Therefore, wind profile shapes are studied with both these features included. Each wind profile sample consists of easterly and northerly wind speed components for multiple heights (vertical grid points) at a given time and location and is processed in two steps to obtain its shape. Firstly, similar to Kalverla et al. (2017) and Malz et al. (2020a), the wind speed components are expressed as parallel and perpendicular components relative to the wind velocity at a reference height, which we have chosen to be 100 m. As a result, the value for the perpendicular wind speed at 100 m is zero and the reformatted wind profile is independent of the wind direction at 100 m. Secondly, the wind speed components are normalised using the 90th percentile of the sample's wind velocity magnitudes. Using the percentile makes the normalisation less sensitive to outliers than using the maximum value. The normalised parallel and perpendicular wind speeds together form the *wind profile shape* of a sample. The normalisation yields a more compact wind resource representation, however, it is prone to producing irregular wind profile shapes for low winds. Therefore, the wind profiles that have a mean wind speed below $5\,\mathrm{m\,s^{-1}}$ are filtered out before clustering. Note that the low wind conditions have a small contribution to the AEP of a wind energy system and their wind profile shapes are thus of small importance for the AEP calculation. Although the results presented in Sects. 3 and 4 do not account for the the low wind samples, the AEP calculation in Sect. 5 does.

### 3.2 Principal component analysis of the wind profile shape dataset

The mean wind profile shape for the offshore location is illustrated in Fig. 2a by plotting the normalised wind speed $\tilde{v}$ against height using profiles for the parallel and perpendicular velocity components. As expected for an offshore location, the mean shape exhibits low wind shear. The hodograph in Fig. 2e shows how the normalised wind velocity changes with height by plotting the parallel and perpendicular normalised wind speed ($\tilde{v}_{\parallel}$ and $\tilde{v}_{\perp}$) for every height. In accordance with Ekman theory, the mean shape shows wind veer (wind direction turns clockwise with height). A logarithmic profile with roughness length $z_0$=0.0002 m, a representative value for open water (Wijnant et al., 2015), is fitted to the lower 200 m of the mean shape. We use 200 m as a proxy for the top of the surface layer, though in very stable situations the surface layer could be considerably smaller. Consequently, we consider applying the logarithmic profile relationship up to 200 m to be valid. Following the approach recommended by Kelly and Gryning (2010), the profile is fitted by varying the friction velocity $v_*$ and the stability function

$\Psi$, which we constrain to the functional forms given in Eqs. 3 and 4. From this, a mean value of the Obukhov length $L$ can be inferred. The best fit profile corresponds to a value $L$=-3391 m, implying a neutral logarithmic profile in the surface layer (assuming neutral conditions if $|L| > 500$). Above 200 m, the fit slightly deviates from the mean profile.

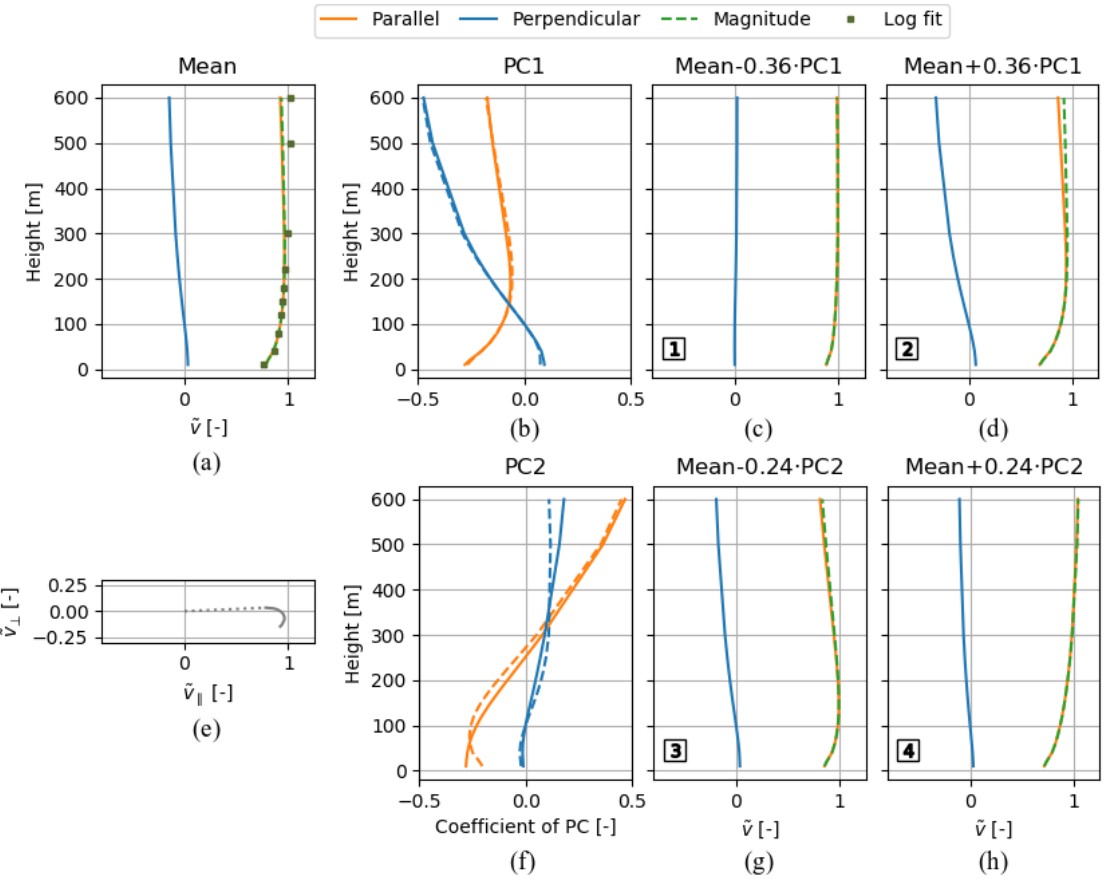

**Figure 2.** Mean wind profile shape and corresponding non-adiabatic logarithmic profile fit (a) and corresponding hodograph (e) for the filtered offshore dataset. Composition of the first and second PCs (b and f). The average of the PC1 (PC2) profiles from the two reference locations is plotted alongside the offshore PC1 (PC2) profile using the dashed line. PC multiplicands superimposed on the mean wind profile shape using minus (c and g) and plus (d and h) one standard deviation as multipliers. The wind profile shape numbers 1–4 refer to the markers in Fig. 4a.

Prior to clustering, a PC analysis is used to reduce the dimensionality of the dataset, while preserving most of the variance. This reduces the computational effort and thus speeds up the clustering. The PC analysis specifies a transformation from the original to the PC coordinate system with its origin coinciding with the mean of the dataset. The first axis is oriented such that it accounts for most of the variance in the data. Subsequent axes are perpendicular to their predecessors and oriented such that

they account for as much of the variance as possible. As a result, the last axis accounts for least of the variance. The PCs are unit vectors in the direction of the positive PC axes.

The compositions of the first two 34-dimensional PCs of the offshore dataset are illustrated in the second column of Fig. 2. The coefficients of each PC describe the relation between the PC and the parallel and perpendicular normalised wind speed components at the 17 heights. The absolute values of the PC coefficients quantify the contribution of the respective normalised wind speed components to the PC. The contributions of the perpendicular components account for most of PC1, indicating that PC1 mostly characterises wind veer. In contrast, the contributions of the parallel components account for most of PC2, indicating that PC2 mostly characterises wind shear. Both PCs show large contributions at both ends of the height range, which indicates that most variance in the dataset is found at these heights. In the PC-space, the data is expressed by multiplicands of the PCs superimposed on the mean wind profile shape. The third and fourth columns of Fig. 2 illustrate how to physically interpret the PCs by depicting two variations of the wind profile shape along PC1 and PC2 using minus and plus one standard deviation as multipliers. This means that 68 % of the PC1 (PC2) values lie between the values used for generating wind profile shape 1 and 2 (3 and 4). Indeed, the wind veer differs substantially between wind profile shape 1 and 2 and the wind shear between 3 and 4.

The percentage of variance retained after dimensionality reduction depends on how many PCs are used to express the data. The relation between the percentage of variance retained and the number of PCs follows from the PC analysis and is shown in Fig. 3. The first four PCs already account for more than 90 % of the variance in the offshore dataset. Since the wind velocities of neighbouring vertical grid points are highly correlated, most of the variance in the data is retained using a limited number of PCs. We consider retaining 90 % or more acceptable for our application. Since the variance retained still increases a few percent between four and five PCs, we opt for using five PCs. The preprocessed data is mapped onto the PC1–5-space and used as input for the clustering.

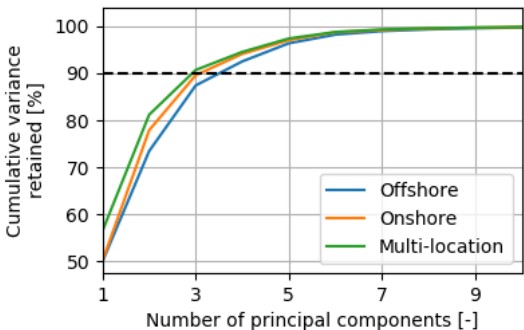

**Figure 3.** Relationship between the percentage of variance retained and the number of PCs for the filtered offshore, onshore, and multi-location datasets analysed in Sect. 4.1, 4.3, and 4.4, respectively.

Figure 4a shows the frequency distribution of the wind profile shapes in the PC1, PC2-space. The PC1, PC2-projection of the wind profile shapes in the third and fourth columns of Fig. 2 are indicated with the markers. By visual inspection, two relatively

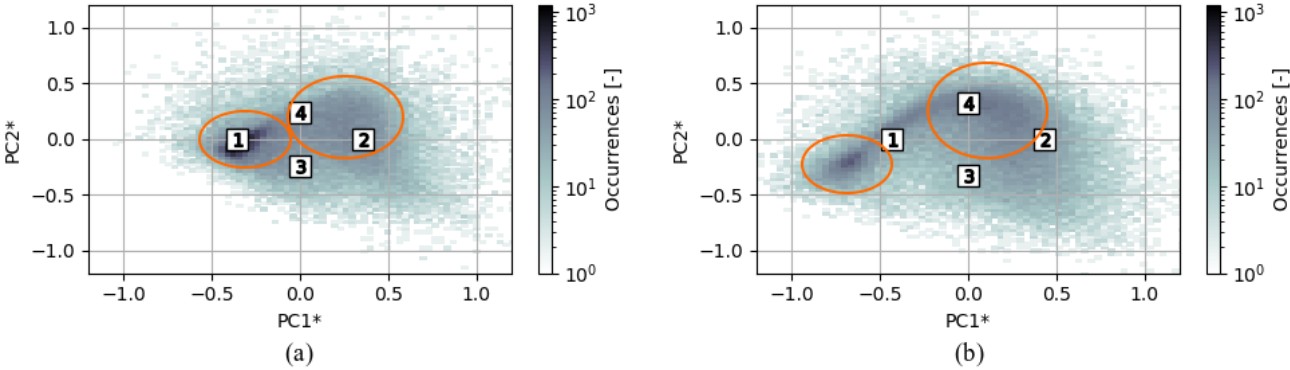

**Figure 4.** Sample frequency distributions in the PC1, PC2-space for the offshore (a) and onshore (b) locations. The origins coincide with the mean wind profile shapes and the markers with the wind profile shapes numbered 1–4 in Figs. 2 and 11. The orange ellipses indicate the visually identified clusters. The plots share the same coordinate system with the x-axis (y-axis) aligned with the average of the PC1 (PC2) unit vectors from the two reference locations. The averaged PCs are denoted by an asterisk and their profiles shown in Figs. 2 and 11 (b and f).

dense groups of data points are identified: a confined group and a less confined group which resembles a tail extending from the first group, marked with the left and right ellipses, respectively. Figure 4b shows results for the onshore location and will be discussed in Sect. 4.3.

### 3.3 Choosing the number of clusters

K-means clustering (Pedregosa et al., 2011) is applied to identify the set of wind profile shapes that are used for representing the wind resource. Each cluster is represented by its centroid and each sample is assigned to the cluster with the nearest centroid. The clustering algorithm iteratively searches for the positions of the centroids that minimise the sum of the squared Euclidean distances between the centroids and their associated samples. This cost function is also referred to as the within-cluster sum of squares (WCSS). The resulting centroids reflect the *cluster-mean wind profile shapes* in the dataset, which follow from
back-transforming the cluster-centroids from the PC to physical space.

K-means clustering is always able to produce a result, which makes it very powerful but also potentially deceptive. The algorithm tends to produce spherical clusters with equal radius and sample size, and works best on data with such a structure. The previous visual analysis of Fig. 4 revealed a different structure type for the wind profile shape datasets with two unevenly sized groups of data points. The number of clusters $k$ generated by the algorithm needs to be specified by the user and it is
15 often not evident how many clusters to choose. The elbow and silhouette method are used for finding an appropriate number for $k$. Moreover, the choice for $k$ is evaluated in the context of applying the cluster-mean wind profile shapes to represent the wind resource.

The elbow method investigates the trend of WCSS against $k$. Increasing the number of clusters is equivalent to reducing the WCSS. Kinks in the trend indicate appropriate choices for $k$. The elbow plot in Fig. 5a shows no distinct kinks for more than three clusters.

The silhouette score expresses the similarity of a sample to the other samples in its cluster relative to its similarity to the
nearest neighbouring cluster's samples. The dimensionless score ranges from -1 to 1: a negative value suggests that the sample is assigned to the wrong cluster, a value around zero indicates that the sample lies between two clusters, and a high value indicates that the sample is assigned to a distinct cluster. Figure 5b shows the mean silhouette score is highest for two clusters. The division of the dataset into two clusters thus yields the most cohesive clusters, which is in agreement with the visual inspection of Fig. 4a. The decreasing trend of silhouette score with $k$ implies that, in general, a small number of clusters should
be used to maintain cluster cohesiveness.

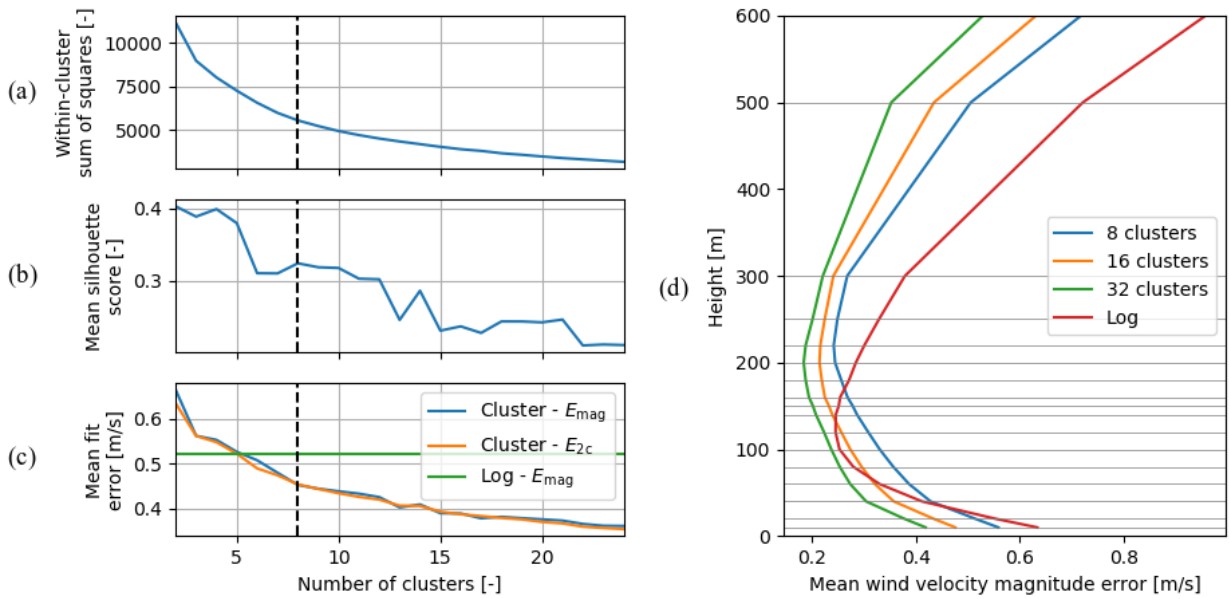

**Figure 5.** Sensitivity of the k-means clustering performance to the number of clusters over the full vertical grid (a–c) and for each height (d) for the filtered offshore dataset. Cost function of the clustering algorithm (a), cluster cohesiveness metric (b), and the mean wind speed fit error (c) against the number of clusters. The dashed vertical lines depict the final choice for eight clusters.

After obtaining the cluster-mean wind profile shapes, they are used for constructing the *cluster representation* of the wind resource. Each sample's absolute vertical wind speed profile is approximated by scaling the associated cluster-mean wind profile shape using the normalisation wind speed used in the pre-processing. We assess the accuracy of this cluster representation using the mean fit error over all filtered samples. The fit error of the $j^{\text{th}}$ sample is calculated by the root mean square of the
errors at each vertical grid point. Two different expressions are used to evaluate the error at the $i^{\text{th}}$ vertical grid point: the wind velocity magnitude error $\varepsilon_{i,j}$ and that which includes both the parallel and perpendicular wind speed errors $\varepsilon_{\parallel,i,j}$ and $\varepsilon_{\perp,i,j}$.

The resulting magnitude and two-component forms of the mean fit error, $E_{\mathrm{mag}}$ and $E_{\mathrm{2c}}$, are given by:

$$E_{\mathrm{mag}} = \frac{1}{n_{\mathrm{s}}} \sum_{j=1}^{n_{\mathrm{s}}} \left( \sqrt{\frac{1}{n_{\mathrm{h}}} \sum_{i=1}^{n_{\mathrm{h}}} \varepsilon_{i,j}^2} \right) \tag{7}$$

and

$$E_{\mathrm{2c}} = \frac{1}{n_{\mathrm{s}}} \sum_{j=1}^{n_{\mathrm{s}}} \left( \sqrt{\frac{1}{2 n_{\mathrm{h}}} \sum_{i=1}^{n_{\mathrm{h}}} \left( \varepsilon_{\parallel,i,j}{}^2 + \varepsilon_{\perp,i,j}{}^2 \right)} \right) \quad , \tag{8}$$

in which $n_h$ is the number of heights, $n_s$ is the number of samples. The relation between both mean fit errors and the number of clusters is shown in Fig. 5c.

We consider the use of the cluster representation valid when it yields a higher accuracy than a representation that uses logarithmic profiles to approximate the vertical variation of the horizontal wind speed. The logarithmic wind resource representation is obtained by fitting logarithmic profiles with roughness length $z_0$=0.0002 m to each sample. Here, the Obukhov length $L$ passed to the $\Psi$ stability function is restricted to the representative values of the five stability classes, listed in the third column of Table 1. Moreover, the fit is performed to the full height range, i.e. 10–600 m, as we aim to minimise the fit error of the wind resource representation and, therefore, allow applying the logarithmic profile relationship beyond the surface layer. As the logarithmic representation does not include information about the wind direction variation with height, its accuracy is only assessed using $E_{\mathrm{mag}}$. We evaluate the fit error of the cluster representation in relation to the number of clusters and compare it to the fit error of the logarithmic representation. Figure 5c shows that whether the fit error of the cluster representation is evaluated using $E_{\mathrm{mag}}$ or $E_{\mathrm{2c}}$ makes little difference. The cluster representation is more accurate than the logarithmic representation when using three clusters or more.

**Table 1.** Stability classes in terms of the Obukhov length, adapted from Holtslag et al. (2014).

| Class name | Class boundaries [m] | Representative $L$ [m] |
|---|---|---|
| Very unstable (VU) | $-200 \leq L < 0$ | -100 |
| Unstable (U) | $-500 \leq L < -200$ | -350 |
| Neutral (N) | $\|L\| > 500$ | $10^{10}$ |
| Stable (S) | $200 < L \leq 500$ | 350 |
| Very stable (VS) | $0 < L \leq 200$ | 100 |

The wind resource representations do not yield the same accuracy for each vertical grid point. To investigate the height dependency, the mean wind velocity magnitude error over all filtered samples is calculated for each vertical grid point. The results are shown in Fig. 5d, in which the horizontal lines depict the 17 heights of the vertical grid points of DOWA. The fits have a relatively low error around 150 m height and a higher error at the top and bottom of the vertical grid. Around 150 m height, the grid is relatively fine, which is equivalent to allocating more weight to the 100–200 m interval for the logarithmic

profile fitting procedure. As a result, the fitting favours minimising the errors in this interval over those at both ends of the height range. Note that the sensitivity of the cluster representation to the grid spacing is limited by the PC analysis prior to the fitting. As stated before, the PC1 and PC2 profiles show that most variance in the dataset is found at both ends of the height range. Due to the relatively high variance and fit model deficiencies, the fit error is also expected to be largest at these heights.

Although the error of the cluster representations at 100 m is higher than that of the logarithmic representation, on average they perform substantially better.

The choice for the number of clusters used to represent the wind resource depends on the type of analysis. Eight clusters are chosen for investigating their characteristics in Sect. 4. This choice follows from a trade-off between the mean wind profile fit error, the silhouette score, representation validity, and our aim to present a concise analysis and meaningful interpretation

of the resulting clusters. To get more insight in the structures of the eight offshore clusters (MMIJ-1–8), the mean silhouette score is calculated for each cluster. The higher the mean silhouette score, the more likely that a cluster is representing a natural structure in the data. Figure 6 shows that a large fraction of the samples have high silhouette scores for MMIJ-1–4, indicating that MMIJ-1–4 are relatively cohesive clusters. The silhouette score distributions of MMIJ-5–8 indicate less uniform sets of samples, especially that of MMIJ-8. Note that MMIJ-1 is roughly a factor 2.5 larger than the second biggest cluster despite the

tendency of k-means clustering to produce equally sized clusters.

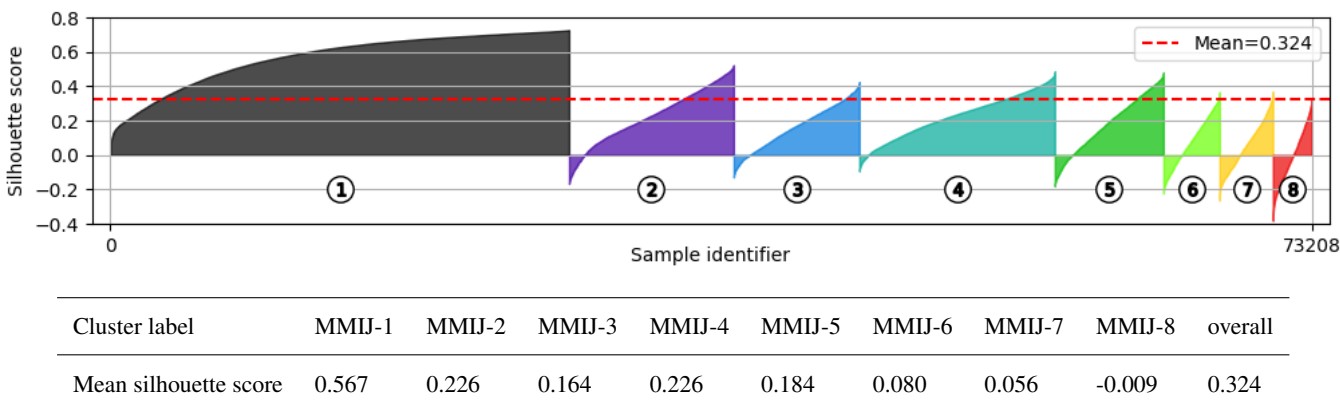

| Cluster label | MMIJ-1 | MMIJ-2 | MMIJ-3 | MMIJ-4 | MMIJ-5 | MMIJ-6 | MMIJ-7 | MMIJ-8 | overall |
|---|---|---|---|---|---|---|---|---|---|
| Mean silhouette score | 0.567 | 0.226 | 0.164 | 0.226 | 0.184 | 0.080 | 0.056 | -0.009 | 0.324 |

**Figure 6.** The silhouette scores of the individual samples grouped by cluster and in ascending order for the filtered offshore dataset. The numbered markers and filled area colours indicate to which cluster the sample belongs. The overall mean score is indicated by the dashed line and the table below the figure states the mean score for each cluster.

## 4 Cluster wind resource representation

This section discusses our physical interpretation of the cluster representations. Firstly, the clusters and their cluster-mean wind profile shapes that result from the offshore dataset are presented. For each cluster, patterns in the times of occurrences are studied together with their association to wind properties at 100 m and atmospheric stability. The analysis is then repeated for the onshore location. The cluster sets for both reference locations are compared to shed some light on the similarities and differences between them. Finally, data from 45 locations are combined to obtain a single set of clusters that is applicable for the entire DOWA domain. For each of the resulting clusters, a map is generated depicting the cluster frequency distribution over the DOWA domain.

### 4.1 Cluster representation for the offshore location

The clustering of the dataset for the offshore location at the met mast IJmuiden yields eight clusters (MMIJ-1–8), which are represented by their centroids shown in Fig. 7a. The clusters are well spread over the PC1, PC2-space, with the exception of MMIJ-5 and 7, which are relatively close to each other. Note that only two axes of the five-dimensional PC-space are shown. Table 2 lists all five PC-coordinates of the cluster-centroids and confirms that the PC1 and PC2 coordinates of MMIJ-5,7 are similar, in contrast to their PC3 coordinates: the centroids are furthest apart along PC3. The PC4 and PC5 values have a substantially smaller range than that for PC1–3 and are superfluous for distinguishing between the eight clusters.

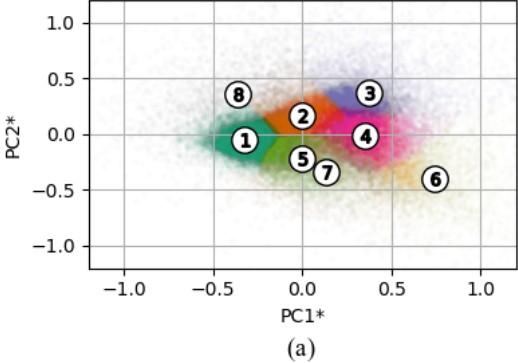
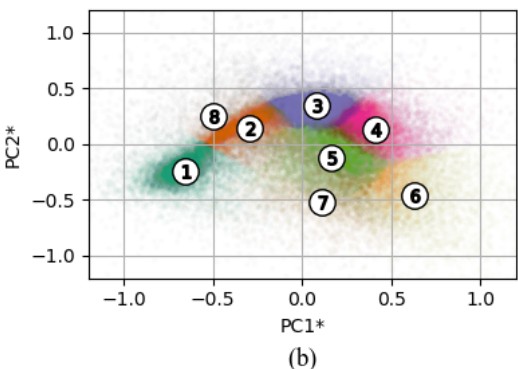

(a)  (b)

**Figure 7.** Projection of the samples onto the PC1, PC2-space for the offshore (a) and onshore (b) locations. The colour indicates to which cluster a sample belongs and the markers represent the cluster-centroids. The plots share the same coordinate system with the x-axis (y-axis) aligned with the average of the PC1 (PC2) unit vectors from the two reference locations. The averaged PCs are denoted by an asterisk and their profiles shown in Figs. 2 and 11 (b and f).

The cluster-mean wind profile shapes of the offshore clusters are shown in Fig. 8. Logarithmic profiles with roughness length $z_0$=0.0002 m are fitted to the magnitude profiles and shown for comparison. Here, the Obukhov length used in the stability function is varied freely and the fit is restricted to the lower 200 m. The values for the Obukhov lengths inferred from

**Table 2.** Principal component coordinates of the cluster-centroids for the filtered offshore dataset. The centroid positions in the PC1, PC2-space are depicted in Fig. 7a with the numbered markers.

| Cluster label | PC1 | PC2 | PC3 | PC4 | PC5 |
|---|---|---|---|---|---|
| MMIJ-1 | -0.33 | -0.05 | -0.04 | -0.02 | 0.01 |
| MMIJ-2 | 0 | 0.17 | -0.08 | 0.05 | 0.02 |
| MMIJ-3 | 0.38 | 0.38 | -0.01 | 0.05 | 0 |
| MMIJ-4 | 0.35 | -0.02 | 0.04 | -0.09 | -0.06 |
| MMIJ-5 | 0 | -0.22 | -0.16 | 0.07 | -0.04 |
| MMIJ-6 | 0.74 | -0.4 | 0.02 | 0 | 0.12 |
| MMIJ-7 | 0.14 | -0.33 | 0.44 | 0.09 | 0.03 |
| MMIJ-8 | -0.36 | 0.36 | 0.45 | 0.04 | 0.02 |

the fits are plotted as $500\,\mathrm{m}/L$ in Fig. 9 and categorised using the stability classes in Table 1. The comparison serves to show to what extent the cluster shapes deviate from non-adiabatic logarithmic profiles, particularly above the surface layer.

To investigate the characteristics of each cluster, Fig. 10a–c show how the clusters are distributed over the years, months, and hours of the day. Figure 10a shows that the inter-annual variability is limited, which asserts that the results can safely be generalised to the lifetime of a wind energy system ($\sim$20 years). The absolute frequency on the y-axis serves to show the cluster sizes. Figure 10d–f show the relative frequency of each cluster for different conditions in terms of wind speed, wind direction, and atmospheric stability. As for the logarithmic profile fits, the stability of each sample is classified using Table 1.

For generating Fig. 10f, we derive the stability class distributions using the bulk Richardson number $\mathrm{Ri_B}$ converted to the Obukhov length $L$ using Eqs. 5 and 6. The data from either ERA5 or DOWA could be used to derive $\mathrm{Ri_B}$, however, we found that using the data from the two lowest ERA5 model levels, i.e., $\sim$10–31 m yields the most realistic values. We use the arithmetic mean of the model level heights for $\bar{z}$ in order to convert $\mathrm{Ri_B}$ to $L$.

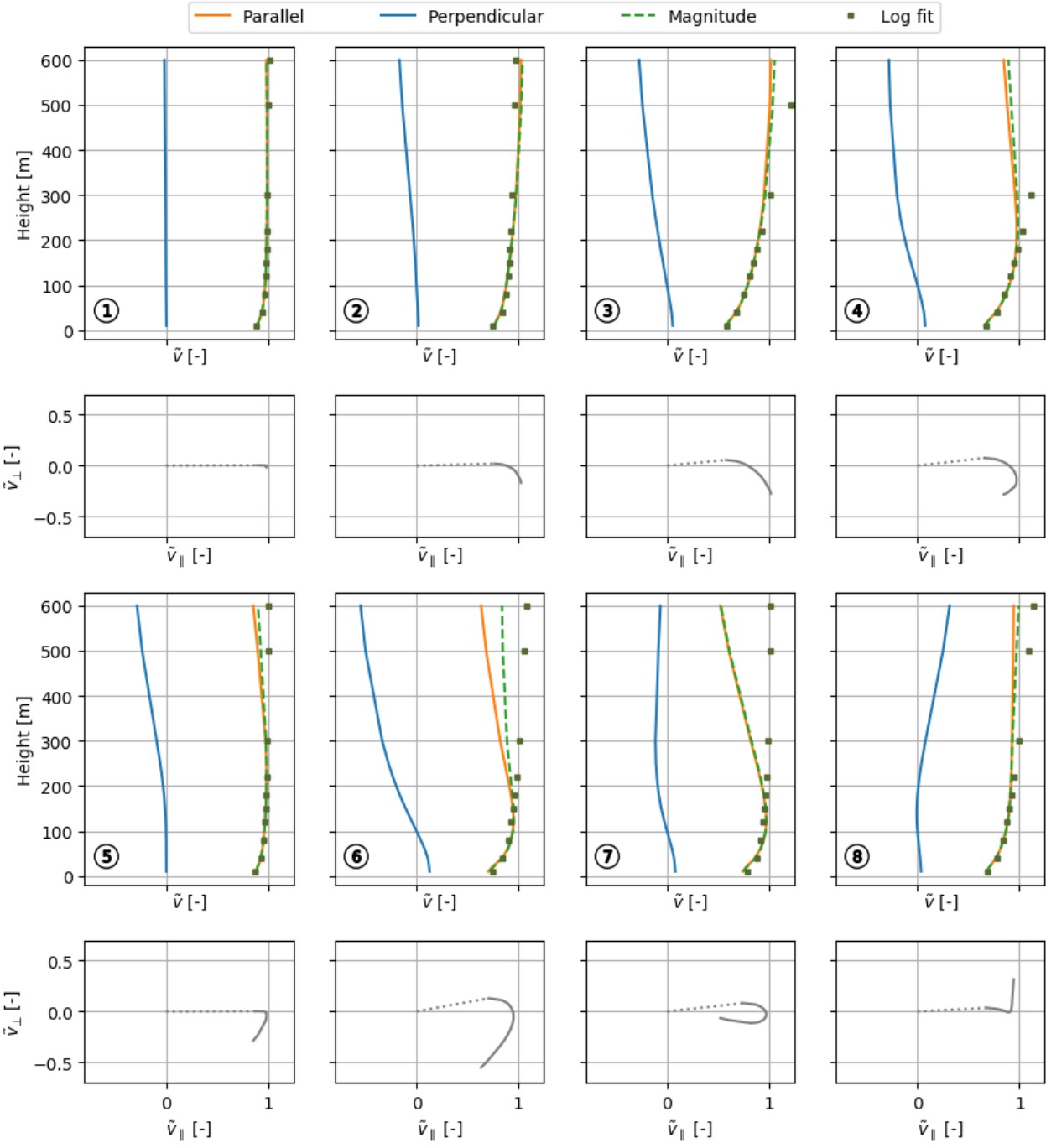

**Figure 8.** The eight cluster-mean wind profile shapes of the offshore clusters (MMIJ-1–8). Each shape is depicted by the normalised wind speed components with height (first and third rows) with the corresponding hodograph below (second and fourth rows). Non-adiabatic logarithmic profile fits are plotted alongside the shapes. In each hodograph, the lower end of the profile is indicated by the dotted line connecting the lowest height point to the origin. All plots share the same x-axis.

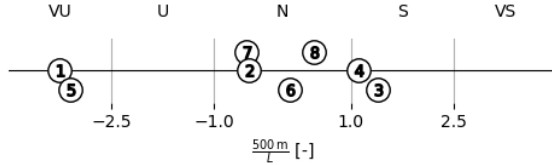

**Figure 9.** Obukhov lengths (plotted as $500\,\mathrm{m}/L$) found by fitting logarithmic profiles to the offshore cluster-mean wind profile shapes in Fig. 8. The stability classes are adopted from Table 1.

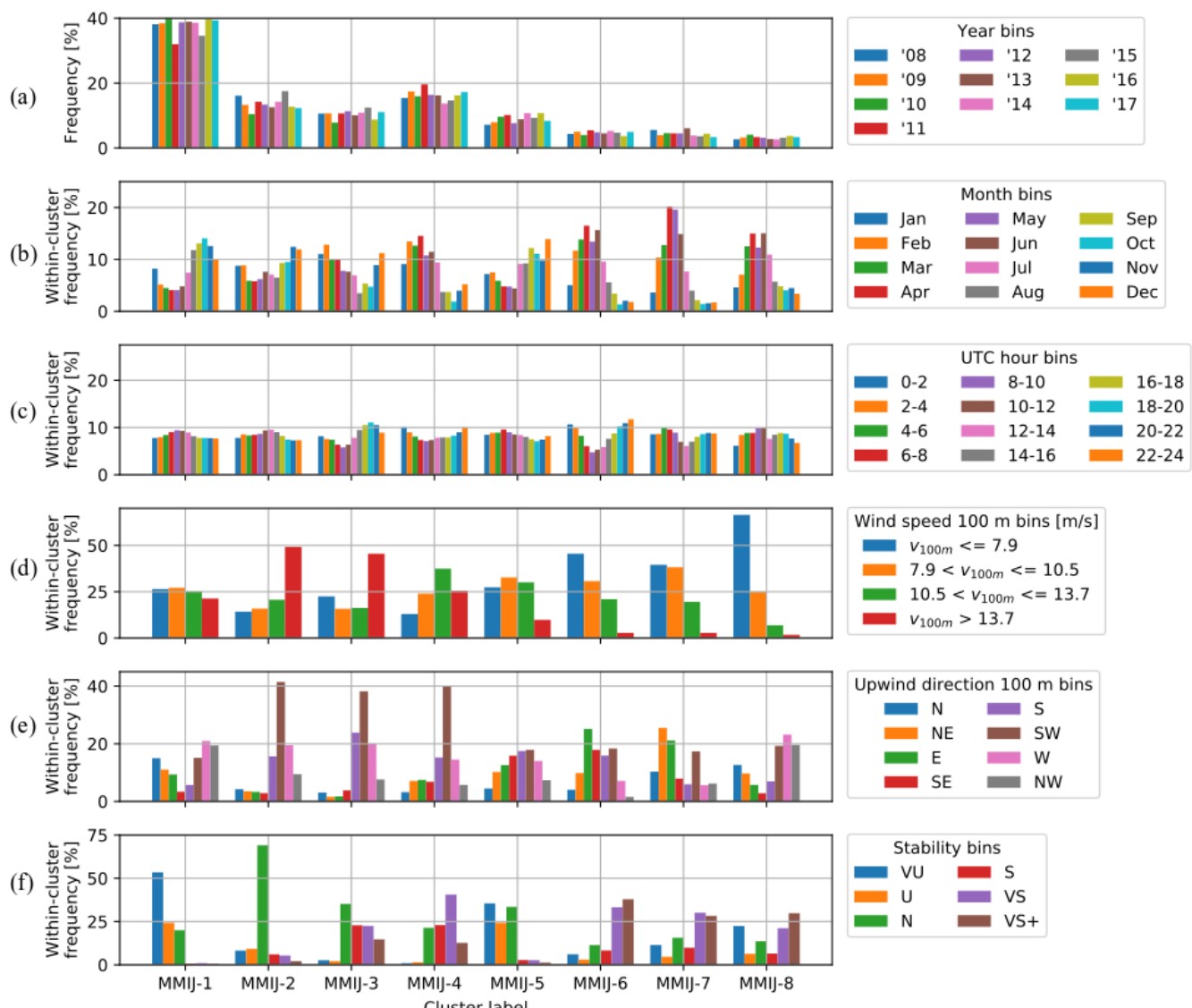

**Figure 10.** Frequency distributions broken down into bins by time of occurrence (a, b, and c), wind speed and direction at 100 m (d and e), and atmospheric stability (f) for the filtered offshore dataset. The wind speed bin limits are chosen such that the frequency over all clusters for each bin is roughly the same. The stability bins correspond to the classes in Table 1 together with the VS+ bin ($\mathrm{Ri_B} \geq 0.2$). The other distributions have equal bin widths.

## 4.2 Interpretation of the offshore cluster representation

By examining Figs. 8 and 9, we can see how the cluster-mean wind profile shapes differ from standard logarithmic profiles, particularly above the surface layer. Moreover, by referring to Fig. 10, it is possible to investigate the conditions under which each of the clusters occur, and to gain insight into their physical origins.

Figure 8 shows that the MMIJ-1 and 2 magnitude profiles are well-described with logarithmic profiles. The MMIJ-1 profile shape suggests a well-mixed convective profile with little wind shear and veer. MMIJ-1 occurs predominantly in autumn and is slightly more frequent in the morning hours. The wind is more frequently weak or moderate than strong and mostly coming from the westerly, north-westerly or northerly directions. Furthermore, this cluster occurs predominantly during unstable conditions. These observations make sense as in autumn, the relatively warm sea water favours neutral to unstable stratification; the dominant wind directions have long fetches over sea which allow the boundary layer to reach an equilibrium state due to the relatively constant surface forcing.

The MMIJ-2 and 3 profile shapes show an increase in wind shear relative to MMIJ-1. The MMIJ-2 profile shape closely resembles a neutral logarithmic profile up to $600\,\text{m}$, whereas that of MMIJ-3 only shows a good fit with a stable logarithmic profile in the surface layer. These clusters occur typically during strong winds, predominantly from the south-west. Strong south-westerly winds are characteristic of the wind climate at this mid-latitude location, which is dominated by the frequent passage of low-pressure systems. The relatively strong winds explain why we see the highest occurrence of near-neutral conditions, especially for MMIJ-2. For MMIJ-3, we also see frequent stable conditions. Simultaneously, we observe that MMIJ-2 occurs more often in the late autumn and MMIJ-3 in winter and the start of the spring. The colder sea water in spring favours the formation of stable stratification, which explains the difference in stability distribution between the two clusters. Stable stratification suppresses turbulent mixing, which helps to sustain a strong wind shear, consistent with the increasing wind shear and veer seen in Figs. 7a and 8.

The MMIJ-4–7 profile shapes are all jet-like. Because wind speed increases monotonically with height in the logarithmic wind profile relationship, it can not describe these type of profile shapes. The wind direction and stability distributions associated with the MMIJ-4 cluster are correlated with south-westerly winds and stable stratification. The seasonal cycle is very pronounced and peaks in spring, when stable stratification is frequent. The winds recorded for MMIJ-4 are mostly moderate to strong. The distributions associated with the MMIJ-5 cluster are very similar to those of MMIJ-1, with the exception of the wind direction distribution, which shows an opposite trend. The winds with a southerly component are dominant for MMIJ-5 and typically have shorter fetches over sea than the north and westerly winds seen for MMIJ-1. The hodograph of the MMIJ-5 profile shape indicates a rather abrupt kink around $140\,\text{m}$, suggesting a discontinuity such as a boundary-layer top. The MMIJ-6 profile shape shows a maximum at $120\,\text{m}$. Although the magnitude profiles of MMIJ-5 and 6 look somewhat similar, the MMIJ-6 profile shape veers more. The MMIJ-7 profile shape shows the most pronounced jet-like shape, also peaking around $120\,\text{m}$. MMIJ-6 and 7 occur almost exclusively for very stable conditions in spring and for weak wind situations. Both clusters occur predominantly for winds with an easterly component and show a diurnal cycle with fewer occurrences around noon. Such a diurnal cycle is in agreement with various studies that have linked low-level jets and the diurnal variation of both the

land-sea temperature difference and the intensity of turbulent mixing (e.g., Burk and Thompson, 1996; Parish, 2000; Mahrt et al., 2014; Shapiro et al., 2016).

The hodographs of MMIJ-5 and 8 both show a sharp bend around 140 m. However, the wind direction turns anticlockwise with height above the bend for MMIJ-8, which is opposite to the veering of the other profile shapes. Despite the peculiarities of the wind direction profiles, the magnitude profiles of MMIJ-5 and 8 are described reasonably well below 200 m with very unstable and neutral logarithmic profiles, respectively. MMIJ-8 occurs mostly in spring, under stable conditions, and more often for winds with a westerly rather than a southerly component. Note that this shape belongs to an incohesive cluster and, therefore, gives a relatively poor representation of the cluster samples.

## 4.3  Comparing the on- and offshore cluster representations

The dataset for the onshore location at the met mast Cabauw is clustered using the same approach. The eight resulting clusters are referred to as MMC-1–8. The results of the PC analysis of the onshore dataset are shown in Fig. 11, which we will compare to those of the offshore dataset, shown in Fig. 2. A logarithmic profile with roughness length $z_0$=0.1 m, a representative value for the area surrounding the mast (Verkaik, 2006), is fitted to the mean wind profile shape as before. With a stability function value corresponding to $L$=476 m, the mean profile shape below 200 m is in accordance with a stable logarithmic profile. Above that, the fitted logarithmic profile rapidly diverges from the mean shape. A higher wind shear is observed than for the offshore location due to the higher surface roughness. The hodograph in Fig. 11e shows that also the wind veer is substantially increased. Despite the apparent differences in mean shape, the PC1 and PC2 profiles are very similar for both reference locations. The average of the PC1 (PC2) profiles from the two reference locations is plotted alongside the onshore PC1 (PC2) profile using the dashed line. To enable a direct comparison between results, the same coordinate system is used for Figs. 4a and b. The x-axis (y-axis) is aligned with the average of the PC1 (PC2) unit vectors from the two reference locations.

The distribution in Fig. 4b shows a similar pattern to Fig. 4a: a dense, confined group of samples, marked with the left ellipse, with a tail of samples extending from this group at around 45° towards the right ellipse. In general, the samples of the onshore dataset are more spread out than the offshore samples, particularly along the PC1 axis. Also the confined group is less dense for the onshore location. Figure 7 shows that, for both locations, the samples of these confined groups belong to the on- and offshore clusters with number 1. The remaining onshore clusters with monotonic wind speed and veering profiles, MMC-2–4, account for most of the tail, see Fig. 12. Note that the clustering algorithm produces arbitrary labels for each cluster. We have manually renumbered them such that the onshore cluster numbers align with the offshore cluster numbers. This allows us to draw parallels between them and show that the resulting profiles are very similar between both locations, e.g., the first offshore clusters (MMIJ-1–3) also have monotonic profiles.

Logarithmic profiles with roughness length $z_0$=0.1 m are fitted to the cluster-mean wind profile shapes, and plotted alongside them in Fig. 12. The values for the Obukhov lengths inferred from the fits are shown in Fig. 13 and categorised by stability class. For the offshore location, Fig. 9 shows that six out of eight logarithmic profiles found by fitting are neutral or stable and those for MMIJ-1 and MMIJ-5 are more unstable. Figure 13 shows that only one unstable logarithmic profile is found for the onshore location, next to six stable and one neutral logarithmic profile. Since there is little diversity in the shape of the unstable

profiles, all the associated samples are grouped together by the clustering. The fact that this type of profile is well-mixed with little shear and a relatively high boundary layer height explains why the diversity is small. By contrast, the neutral and stable profiles can have a wide range of shear and in addition, particularly in stable conditions, the boundary layer height can be quite low which will have a strong influence on wind shear. This means that a greater diversity of profile shapes is to be expected under neutral or stable conditions.

The profile shapes for MMC-1 to MMC-3 show an increase in wind shear. Between MMC-3 and MMC-4, we see an increased wind veer, though reduced wind shear. The profile shapes for MMC-5–7 are jet-like, as is the case for the offshore clusters MMIJ-4–7. MMC-5 and MMC-6 have similar wind velocity magnitude profiles with a relatively weak wind speed maximum around $200\,\mathrm{m}$, but MMC-6 shows a much stronger wind veer. MMC-7 shows the strongest fall-off above $200\,\mathrm{m}$. Like its offshore counterpart, MMC-8 is characterised by an anticlockwise-turning profile with a sharp bend, which is most clearly visible in the hodograph. Recall that the offshore wind profile shape for MMIJ-5 also showed a sharp bend, albeit, in combination with clockwise turning. We do not see these features for any of the MMC profile shapes.

Figure 14 shows that MMC-3 is the most frequent cluster in the filtered onshore dataset, with a frequency of $20.6\,\%$. The first five onshore clusters have similar total frequencies, whereas MMIJ-1 dominates for the offshore location. As for the offshore clusters MMIJ-6–8, the onshore clusters MMC-6–8 are less frequent.

Figure 14 shows clusters that typically occur during spring/summer (MMC-1, MMC-7 and MMC-8) or autumn/winter (MMC-2–6). The diurnal cycles of the onshore location are highly pronounced in contrast to those for the offshore location. This effect is caused by the lower heat capacity of the land surface which promotes a more immediate heat transfer to or from the atmosphere. Convection created by solar irradiation leads to more turbulent mixing during the day than at night. Indeed, MMC-1 and MMC-2 show mixed profiles and predominantly occur during the day, whereas MMC-3–8 show profiles with less mixing and predominantly occur during the night. Note that low-level jets, and stable conditions in general, occur almost exclusively at night. Figure 14c indicates a pronounced diurnal cycle in atmospheric stability for the onshore location, whereas for the offshore location the seasonal cycle, shown in Fig. 10b, is more pronounced. Figures 10d and 14d display almost identical frequency distributions over the bins, however, the actual wind speed distributions differ due to the different bin limit values. Note that the chosen limits give the same total frequency for each bin, thereby the distributions of the individual clusters are easily related to the uniform general distribution and compared with one another. Also the wind direction distributions show similar patterns for both locations. In the case of the stability distributions, the onshore location shows a tendency to more stable conditions for all clusters.

In conclusion, we see that very similar cluster-mean wind profile shapes have been identified for the on- and offshore reference locations. Moreover, similar profiles seem to be related to similar conditions in terms of wind speed, wind direction, and atmospheric stability. The strongest winds typically act to neutralise the stratification, leading to monotonic profiles with relatively little veer. These profiles are relatively well captured by logarithmic wind profiles. For weaker winds, atmospheric stability acts to enhance wind shear and veer, up to the point where low-level jets are observed. However, whereas stability at the offshore location is governed by a clear seasonal cycle in the underlying sea surface, stability over land is regulated by the

relatively rapid diurnal heating cycle of the land surface. Over sea, the wind direction also seems to play a more pronounced role, since it controls the characteristics of the prevailing fetch.

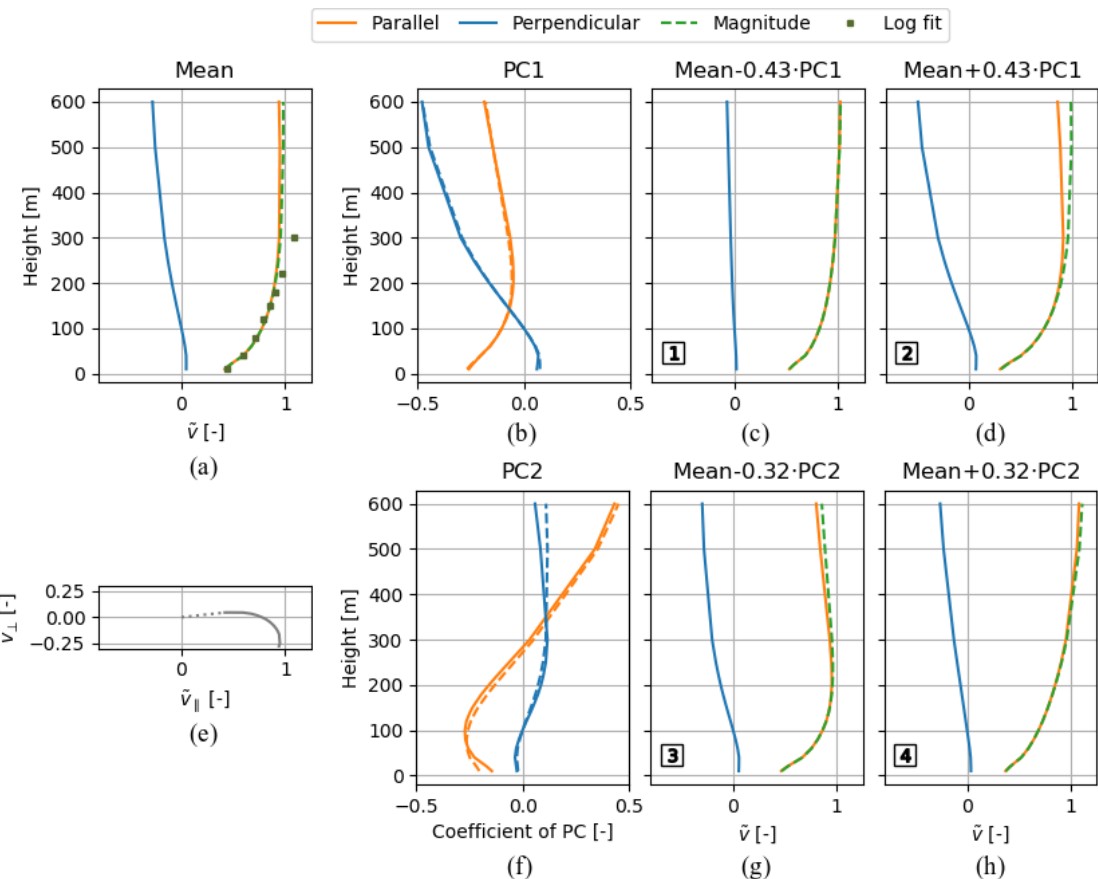

**Figure 11.** Mean wind profile shape and corresponding non-adiabatic logarithmic profile fit (a) and corresponding hodograph (e) for the filtered onshore dataset. Composition of the first and second PCs (b and f). The average of the PC1 (PC2) profiles from the two reference locations is plotted alongside the onshore PC1 (PC2) profile using the dashed line. PC multiplicands superimposed on the mean wind profile shape using minus (c and g) and plus (d and h) one standard deviation as multipliers. The wind profile shape numbers 1–4 refer to the markers in Fig. 4b.

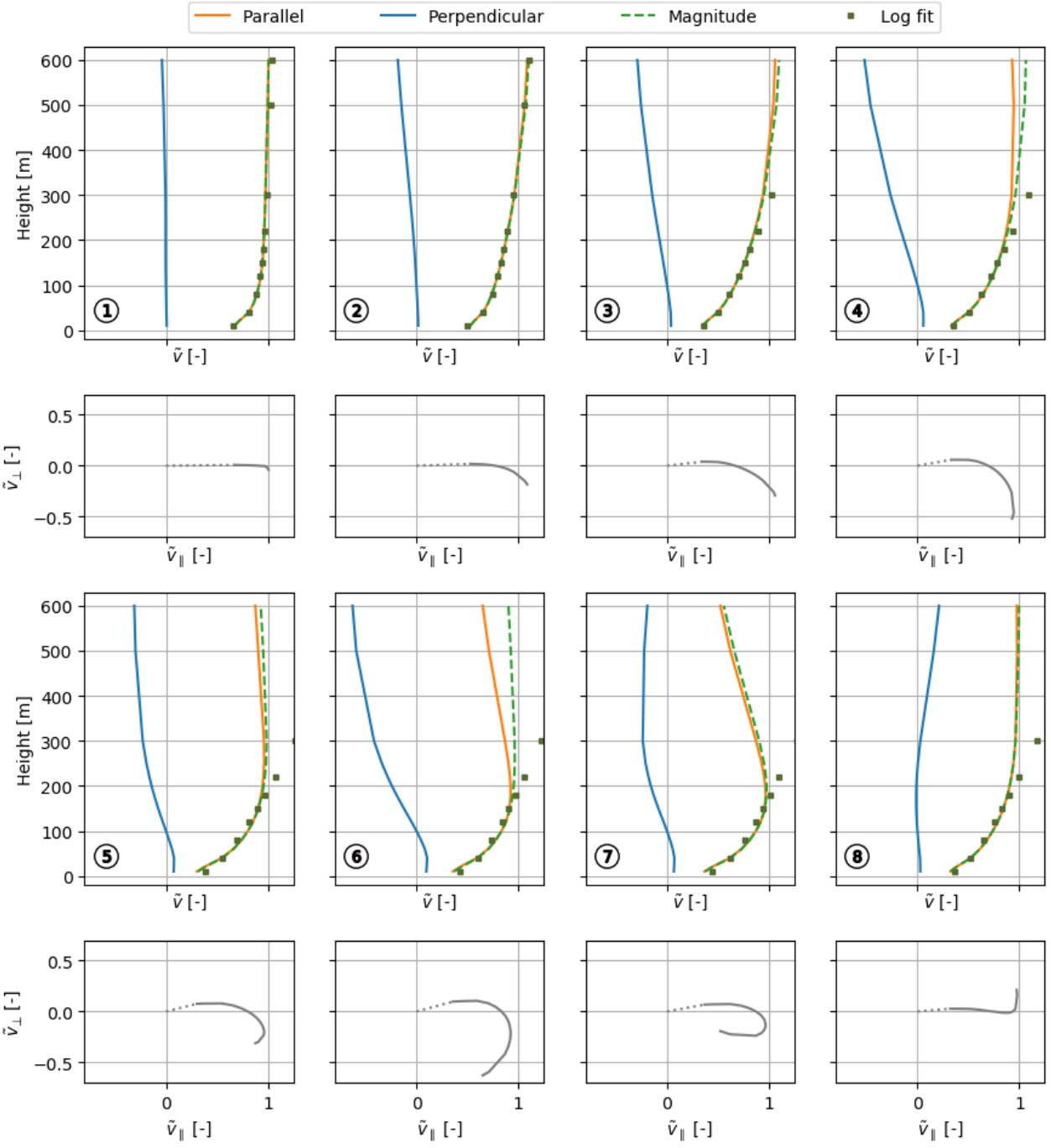

**Figure 12.** The eight cluster-mean wind profile shapes of the onshore clusters (MMC-1–8). Each shape is depicted by the normalised wind speed components with height (first and third rows) with the corresponding hodograph below (second and fourth rows). Non-adiabatic logarithmic profile fits are plotted alongside the shapes. In each hodograph, the lower end of the profile is indicated by the dotted line connecting the lowest height point to the origin. All plots share the same x-axis.

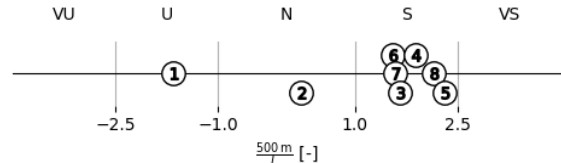

**Figure 13.** Obukhov lengths (plotted as $500\,\mathrm{m}/L$) found by fitting logarithmic profiles to the onshore cluster-mean wind profile shapes in Fig. 12. The stability classes are adopted from Table 1.

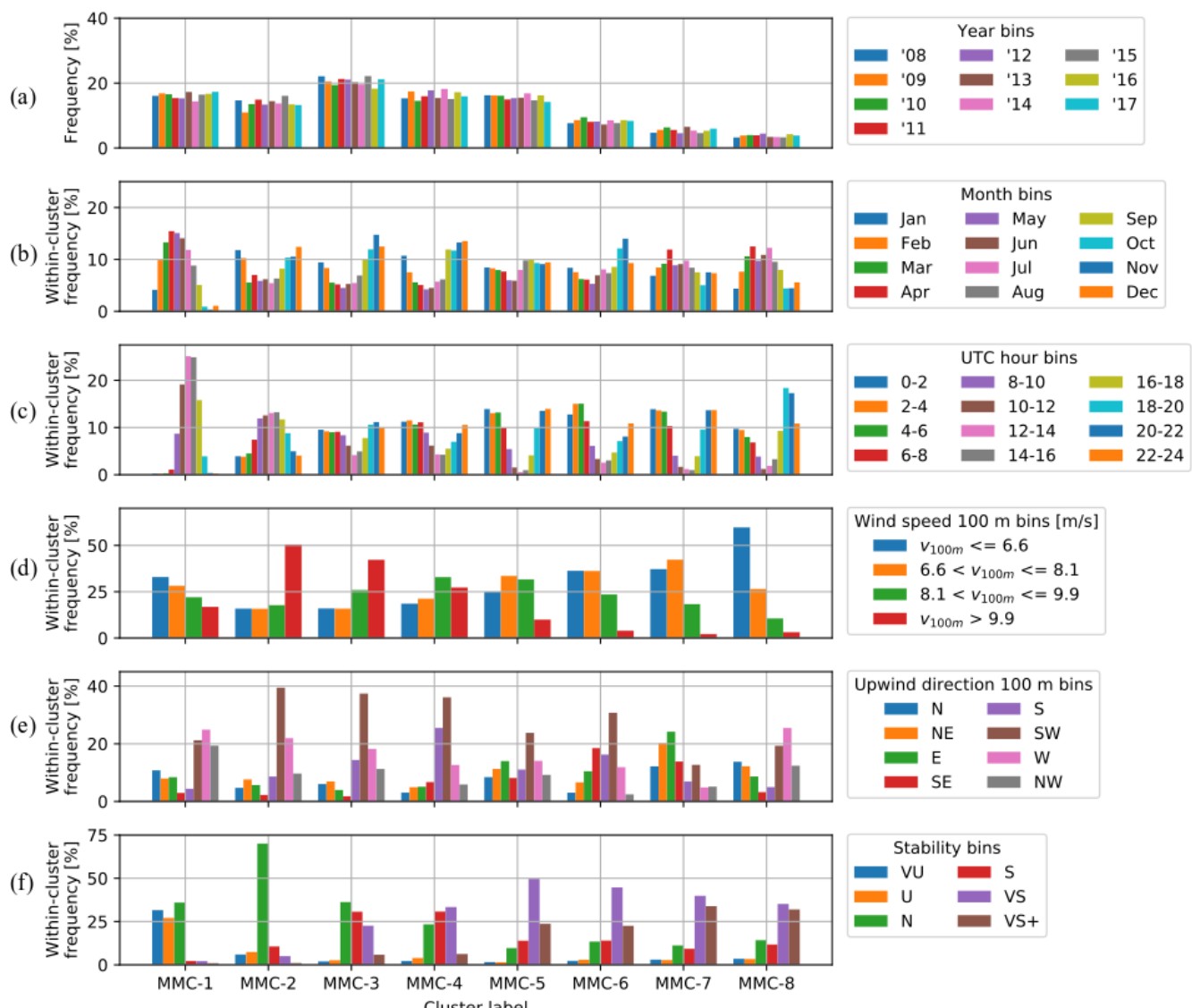

**Figure 14.** Frequency distributions broken down into bins by time of occurrence (a, b, and c), wind speed and direction at $100\,\mathrm{m}$ (d and e), and atmospheric stability (f) for the filtered onshore dataset. The wind speed bin limits are chosen such that the frequency over all clusters for each bin is roughly the same. The stability bins correspond to the classes in Table 1 together with the VS+ bin ($\mathrm{Ri_B} \geq 0.2$). The other distributions have equal bin widths.

## 4.4 Spatial frequency distribution of wind profile shape clusters

So far, we have shown that similar clusters and, consequently, similar wind profile shapes were identified for both onshore and offshore locations. Here, we apply our clustering algorithm on a dataset that includes wind data from a variety of locations. The multi-location dataset (filtered to exclude low wind samples) includes wind data from 45 DOWA grid points that are selected such that onshore, coastal, and offshore locations are equally represented. For each location type, 15 grid points are chosen pseudo-randomly to yield a good coverage of the full DOWA domain (50778 grid points in total). The sampled grid points are marked on the map in Fig. 1. Our aim is to give some insight into the spatial variability of wind profile characteristics, in particular to see how the clustering approach highlights profile characteristics of the on- and offshore environments. In principle, the multi-location approach gives a set of profile shapes that could be used for an AEP assessment, though a site-specific set would be better suited if a more accurate assessment is required. Whilst we increase the applicability of the cluster representation to a larger area by keeping the number of clusters the same, we compromise on the accuracy. As mentioned before, increasing the number of clusters reduces the error. The number of clusters can be increased until a suitable accuracy is attained. Here, we still use eight clusters, as it suffices to give an impression of the spatial variability of the wind profile shapes. The eight resulting multi-location clusters are referred to as ML-1–8.

Figure 15 shows the cluster-mean wind profile shapes for each of the multi-location clusters. Each sample of every grid point in the DOWA domain is assigned to the cluster with the closest centroid. For each cluster, a map is generated showing the spatial distribution of its frequency of occurrence, see Fig. 16. Note that the colour scale is different for each map such that spatial patterns are easier to observe. Table 3 lists the frequency of each cluster at the on- and offshore reference locations. It is interesting to compare the multi-location clusters with the site-specific clusters identified earlier. With a frequency of 48.5 %, ML-1 is dominant at the met mast IJmuiden. Therefore, we expect it to be similar to MMIJ-1, the dominant cluster resulting from the offshore analysis. Comparing Figs. 8 and 15 indeed shows that the cluster-mean wind profile shapes of ML-1 and MMIJ-1 look alike. Similarly, ML-7 has the highest frequency at the met mast Cabauw, i.e. 21.7 %, and has a profile shape somewhere in between those of MMC-3 and 4, the most frequent clusters resulting from the onshore analysis. Every multi-location cluster is manually linked to the single location clusters based on resemblance of their cluster-mean wind profile shapes, see Table 3.

The maps in Fig. 16 show a distinct division between clusters that mostly occur over sea (ML-1–3) and over land (ML-4–8). The latter group is sub-divided into coastal and onshore clusters, see Table 3. The sharply defined patterns in the frequency maps of ML-5–8 coincide with orographic features and thus suggest a strong relationship between the clusters and orography. Other site characteristics such as recurring weather systems and land cover also affect the clusters and thus the frequency maps. Over land the frequency maps of ML-5 and ML-6 suggest an inverse relationship: the frequency of ML-5 peaks at high elevations, whereas that of ML-6 is highest in the river valley in the lower right corner of the DOWA domain. A similar inverse relationship is observed between ML-7 and ML-8. Also, the frequency maps of ML-7 and ML-8 show contours coinciding with the elevation map, though the relationship between the frequency and elevation is not as direct as for ML-5 and ML-6.

**Table 3.** Classification of the multiple-location clusters and frequencies of occurrence of the clusters at the on- and offshore reference locations (met masts Cabauw and IJmuiden).

| Cluster label | Class | Similar single location cluster(s) | Frequency at offshore location | Frequency at onshore location |
|---|---|---|---|---|
| ML-1 | offshore | MMIJ-1 | 48.5 % | 5.8 % |
| ML-2 | offshore | MMIJ-2, 3 | 22.0 % | 4.2 % |
| ML-3 | offshore | MMIJ-4, 6 | 14.1 % | 4.2 % |
| ML-4 | coastal | MMC-1 | 8.6 % | 16.1 % |
| ML-5 | onshore/coastal | MMC-2 | 3.0 % | 17.4 % |
| ML-6 | onshore | MMC-6 | 2.0 % | 13.3 % |
| ML-7 | onshore | MMC-3, 4 | 1.2 % | 21.7 % |
| ML-8 | onshore | MMC-5 | 0.6 % | 17.3 % |

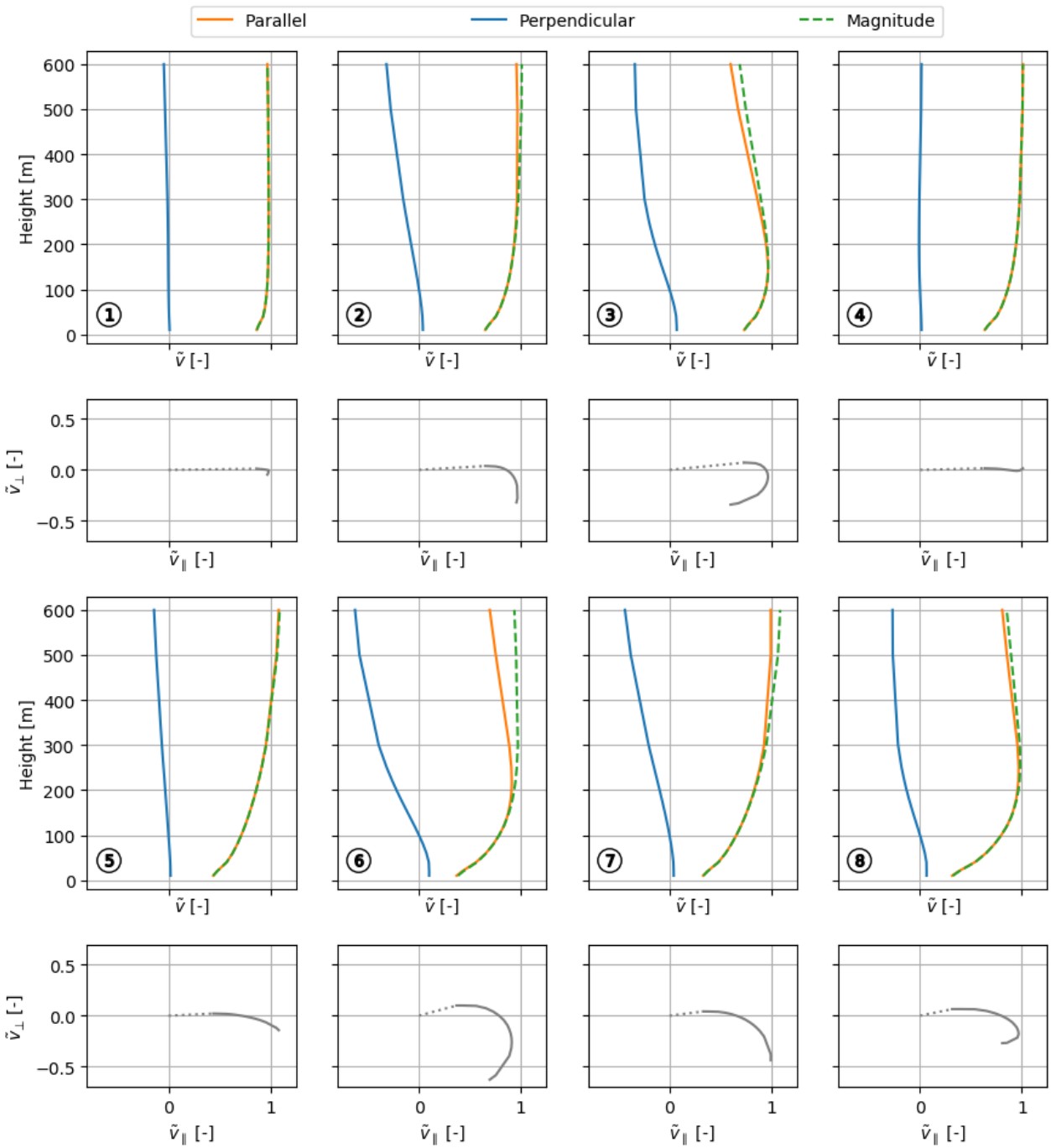

**Figure 15.** The eight cluster-mean wind profile shapes of the multi-location clusters (ML-1–8). Each shape is depicted by the normalised wind speed components with height (first and third rows) with the corresponding hodograph below (second and fourth rows). In each hodograph, the lower end of the profile is indicated by the dotted line connecting the lowest height point to the origin. All plots share the same x-axis.

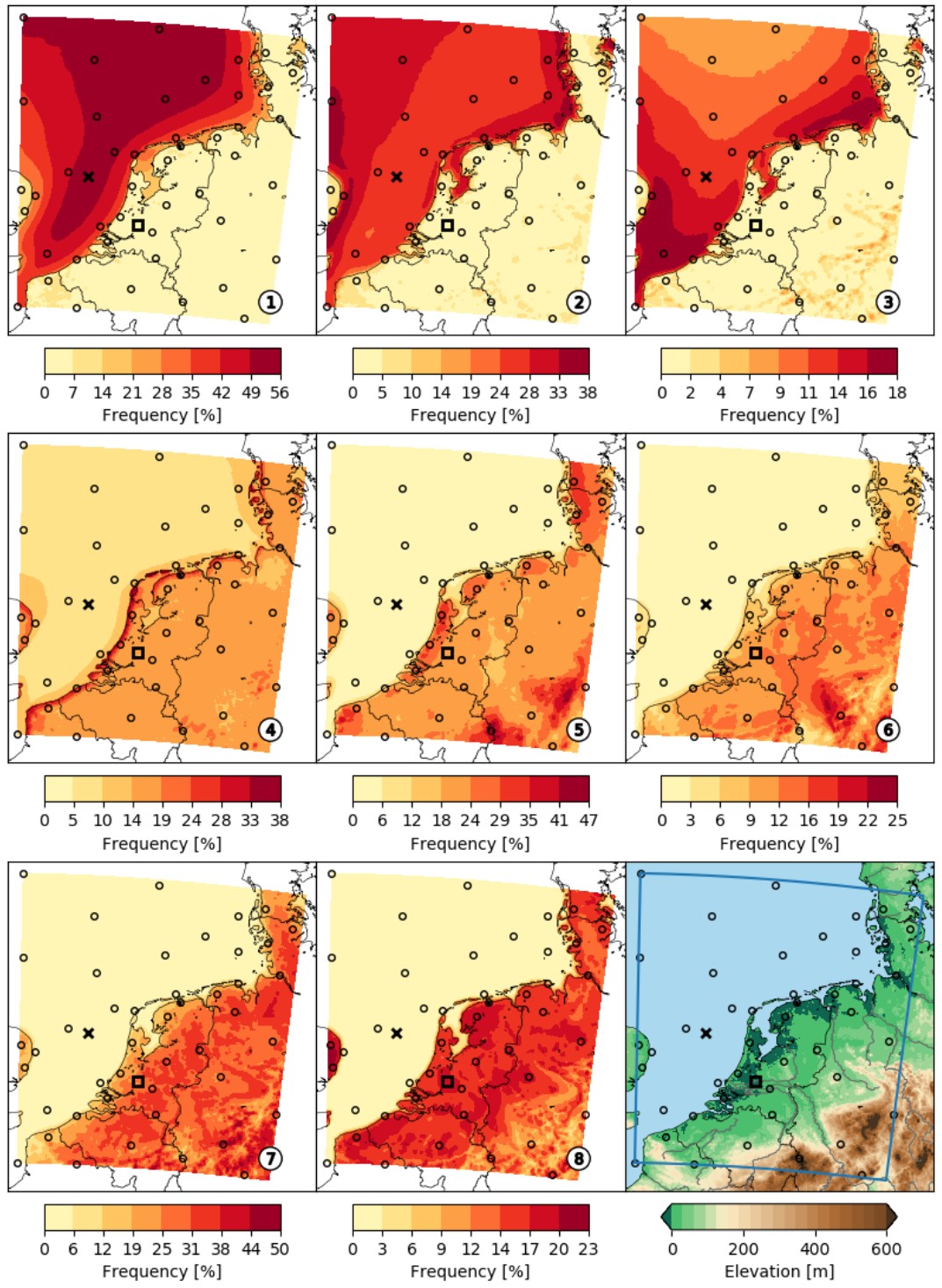

**Figure 16.** Frequency of occurrence of each multiple-location cluster (ML-1–8) mapped over the DOWA domain. The × and □ markers depict the reference locations of the met masts IJmuiden and Cabauw. The ○ markers show the sampled grid points. The lower right plot is a repetition of Fig. 1.

## 5 Efficient AWE production estimation using the cluster representation

With AWE technology maturing and approaching the deployment stage, the community is debating how to uniformly define the performance of AWE systems (Van Hussen et al., 2018). A generally applicable set of wind profile shapes is considered to be an important step to facilitate the standardisation of wind conditions for which AWE systems are rated in terms of power production. In this section, we demonstrate how the wind resource representations obtained using the clustering procedure can be used for estimating the AEP of a pumping AWE system. An advantage of AWE systems over tower-based wind turbines is that they have access to winds at higher altitudes. This advantage is limited when low-shear wind profiles are frequent at the installation site, as is the case offshore, but unusual for onshore locations. Deploying an AWE system at an onshore location thus requires a more variable operational approach. For this reason, we demonstrate the AEP estimation for the met mast Cabauw location using the eight clusters from the single location analysis (Sect. 4.3). A separate power curve is generated for each cluster using its cluster-mean wind profile shape. Each power curve together with the corresponding cluster-specific wind speed distribution yields the AEP contribution of the respective cluster. Finally, the sensitivity of the total AEP to the number of clusters is evaluated.

### 5.1 Deriving power curves for a pumping AWE system

A pumping AWE system is alternating between reeling the tether in and out and thereby consuming and producing power, respectively. The relatively high lift force generated by the kite during reel-out and long reel-out phase yields a positive net energy output. The specific operational approach differs between AWE concepts and may require different performance models for calculating the generated power. We evaluate a flexible-kite system using the quasi-steady model (QSM) developed by Van der Vlugt et al. (2019) specifically for this concept. The cluster representation can in principle be used together with any performance model for estimating the AEP.

Flexible-kite systems typically sweep a large height range during the pumping cycle, which requires pronounced transitions between the reel-out and reel-in phases (Salma et al., 2019). Figure 17 shows the distinct phases of a pumping flexible-kite system. During the reel-out phase, the kite flies figure-of-eight manoeuvres in a fast cross-wind motion. After the reel-out phase, the kite stops flying cross-wind, de-powers, and flies towards zenith. Once reeled back in, the kite steers down, flies towards the starting position of the reel-out phase, and starts a next cycle. The QSM idealises and represents the pumping cycle using three phases: the reel-in, transition, and reel-out phase. The transition between the reel-out and reel-in phases is not modelled separately, but is included in the reel-in phase. The model does not resolve the cross-wind flight manoeuvres during the reel-out phase but represents them by an average cross-wind flight state with constant values for the elevation, azimuth, and course angle. The motion of the kite is approximated by moving it along the idealised flight path according to the computed steady-state kite speed.

The QSM assumes a steady wind field with a constant wind direction and only a vertical variation in the wind speed. Therefore, we consider only the magnitude profiles of the cluster-mean wind profile shapes in the calculations. The unidirectional wind profile approximation is equivalent to hypothetically knowing the wind direction profile and steering the kite to correct

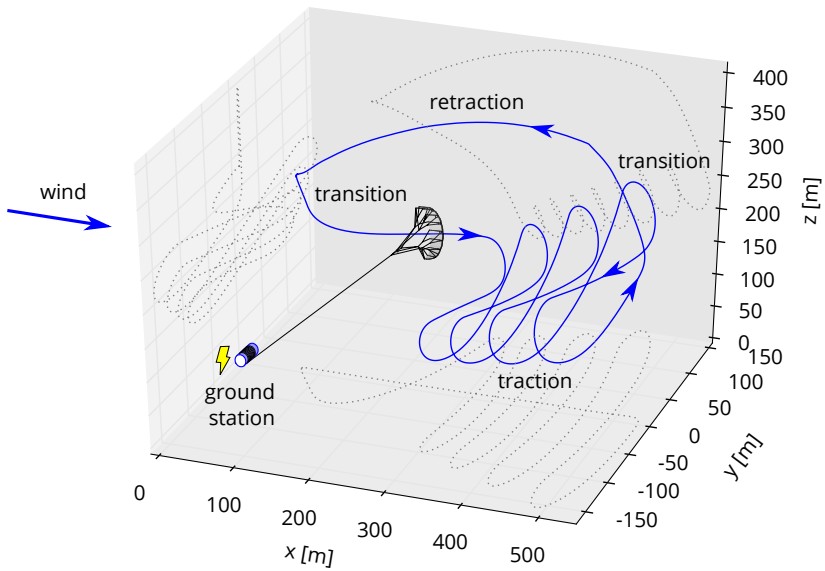

**Figure 17.** Flight path of the flexible-kite, pumping AWE system (kite & drum not to scale) adapted from Fechner (2016).

for direction changes. We use the system properties of the 20 kW technology demonstrator of Delft University of Technology given in Table 4. The QSM uses constant values for the lift and drag coefficients of the powered and de-powered kite. In reality, the coefficients vary and representative values of the leading edge inflatable kite are selected based on the experiment of Oehler and Schmehl (2019).

**Table 4.** Constant system properties that are required as model input for the QSM. $C_L$ and $C_D$ stand for lift and drag coefficients, respectively. The kite and tether properties follow from the work of Oehler and Schmehl (2019). The other properties are chosen by judgment of the authors for being representative for the analysed system.

| Kite properties | | Tether properties | | Operational limits | | Representative reel-out state | |
|---|---|---|---|---|---|---|---|
| Projected area | $19.75\,\mathrm{m}^2$ | Density | $724\,\mathrm{kg\,m^{-3}}$ | Min. reeling speed | $2\,\mathrm{m\,s^{-1}}$ | Azimuth angle | $13°$ |
| Mass | $22.8\,\mathrm{kg}$ | Diameter | $4\,\mathrm{mm}$ | Max. reeling speed | $10\,\mathrm{m\,s^{-1}}$ | Course angle | $100°$ |
| $C_{L,\mathrm{powered}}$ | $0.9$ | $C_{D,\mathrm{tether}}$ | $1.1$ | Min. tether force | $300\,\mathrm{N}$ | | |
| $C_{D,\mathrm{powered}}$ | $0.2$ | | | Max. tether force | $5000\,\mathrm{N}$ | | |
| $C_{L,\mathrm{depowered}}$ | $0.2$ | | | | | | |
| $C_{D,\mathrm{depowered}}$ | $0.1$ | | | | | | |

5    The proposed AEP estimation requires the characterisation of the maximal mean cycle power for a large variety of wind conditions. The mean cycle power depends on the operational settings that control the cycle trajectory and phase durations. These cycle settings include the forces applied to the tether during reel-in and reel-out. The values of the cycle settings are chosen such that they yield maximal mean cycle power. The reel-in tether force should allow a fast retraction of the kite,

while limiting the energy consumption. During the transition phase, the reeling speed is kept zero as long as the tether force does not exceed its limit. During reel-out, the tether force should yield a high power, while increasing the fraction of time spent producing energy in the cycle. For high wind speeds, the system runs into its maximum tether force and reeling speed limits. Increasing the elevation angle of the reel-out path generally indirectly de-powers the kite and alleviates the tether force.

Controlling the elevation angle can thereby expand the wind speed range that allows safe operations. Although not considered here, the kite could also be de-powered directly by controlling $C_{L,\mathrm{powered}}$. The effective pumping length of the trajectory is the difference between the minimum and maximum tether length during reel-out. The minimum tether length is fixed at 200 m.

We use numerical optimisation to determine the cycle settings that maximises the mean cycle power. Table 5 lists the cycle setting parameters, which are used as optimisation variables, together with their respective limits. Imposing a lower

bound on the tether force ensures that the kite stays tensioned, as required for a flexible-kite. The upper bound corresponds to the maximum allowed tether force. The remaining limits are chosen by judgment of the authors. The optimisation uses the sequential quadratic programming algorithm (SLSQP) that is part of pyOpt (Perez et al., 2012). This class of algorithms is generally seen as a good general-purpose method for differentiable constrained non-linear problems. The power curves required for the AEP estimation relate the mean cycle power to the scaling parameter used for de-normalising the cluster-mean

wind profile shapes of MMC-1–8. Given the profile shape, the wind speed at any height can be used as a scaling parameter. We use the wind speed at 100 m as scaling parameter. A power curve is derived for each of the clusters, by determining the maximal mean cycle power for a range of wind speeds at 100 m between cut-in and cut-out. In each step, the profile shape is de-normalised, followed by an optimisation using the resulting wind profile as input.

**Table 5.** Cycle setting parameters which are varied for maximising the mean cycle power and their corresponding limits defining the search space. The limits are chosen by judgment of the authors for being representative for the analysed system.

| Parameter | Lower bound | Upper bound |
| --- | --- | --- |
| Reel-out force | 300 N | 5000 N |
| Reel-in force | 300 N | 5000 N |
| Reel-out elevation angle | 25° | 60° |
| Pumping length tether | 150 m | 250 m |

Prior to performing the optimisations, we determine the cut-in and cut-out wind speeds at 100 m for each wind profile shape.

The cut-in limit is assumed to be the lowest wind speed for which, along the entire reel-out path, feasible steady flight states are found with the QSM. The cut-out limit is determined by the criterion that the pumping cycle should complete at least one figure-of-eight manoeuvre (at an elevation angle of 60°). This criterion becomes more critical at high wind speeds, as the reel-out phase gets shorter. The QSM as presented by Van der Vlugt et al. (2019) does not resolve the cross-wind flight motion. However, this motion can also be approximated as a transition through steady flight states, yielding an approximate duration

of the figure-of-eight manoeuvre. Dividing the total duration of the reel-out phase by the average duration of a figure-of-eight manoeuvre yields the number of cross-wind manoeuvres flown.

Scaling each wind profile shape such that the wind speed at 100 m equals the previously determined cut-in and cut-out wind speeds yields the respective absolute wind profiles, shown in Fig. 18. The cut-in profiles have the same wind speed at roughly 80 m, which is the kite height at the start of the reel-out phase for the minimum elevation angle employed at low winds. This indicates that, for every wind profile, the cut-in criterion is critical at the start of the reel-out phase rather than at the end.

The cut-out profiles exhibit roughly the same wind speed at 300 m, which is the kite height at the end of the reel-out phase for the maximum elevation angle and tether length employed at high winds. The cut-out wind conditions for an AWE system are ambiguous when defined by wind speeds at a certain height without defining the profile shape. However, since the cut-out profiles all intersect at roughly 300 m, characterising the cut-out wind speed at this height yields a reasonably precise definition for all profile shapes. Similarly, the cut-in wind speed is well-defined at 80 m. At 100 m, MMC-3 and MMC-7 show the lowest

and highest cut-out wind speed, respectively.

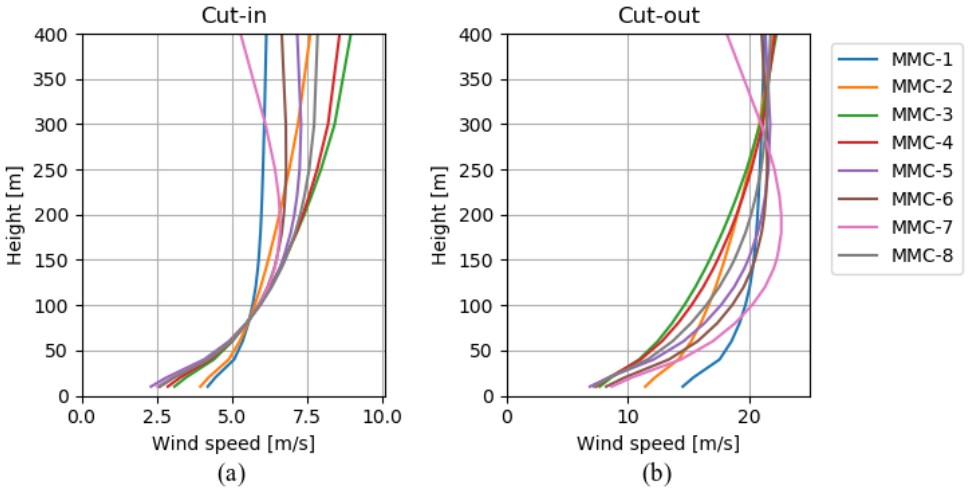

**Figure 18.** The cut-in (a) and cut-out (b) wind profiles that follow from scaling the onshore profile shapes in Fig. 12 (Sect. 4.3) using the calculated cut-in and cut-out wind speeds at 100 m, respectively.

Figure 19 shows the idealised cycle trajectories that follow from the optimisations for the cluster-mean wind profile shape of MMC-1. The depicted trajectories highlight changes in the operational approach at wind speeds at 100 m between cut-in and cut-out, which are discussed next. The optimal pumping tether length coincides with its upper bound for all wind speeds. The reel-out elevation angle of the flight trajectory below $v_{100m}=10.5\,\mathrm{m\,s^{-1}}$ coincides with its lower bound. For higher wind

speeds, an increased inclination of the reel-out path yields a higher mean cycle power. At roughly $v_{100m}=16\,\mathrm{m\,s^{-1}}$, the maximal mean cycle power is reached with the kite completing only one cross-wind pattern. Above $16\,\mathrm{m\,s^{-1}}$ wind speed, the constraint that requires completing at least one cross-wind pattern is driving the elevation angle to higher values until reaching its upper bound for $v_{100m}=19.7\,\mathrm{m\,s^{-1}}$, above which no feasible solution exists.

The calculated power curves are shown in Fig. 20. Note that plotting the mean cycle power against the wind speed at 300 m

would yield curves that end at roughly the same wind speed. Up to roughly $v_{100m}=8.5\,\mathrm{m\,s^{-1}}$, all the power curves are similar.

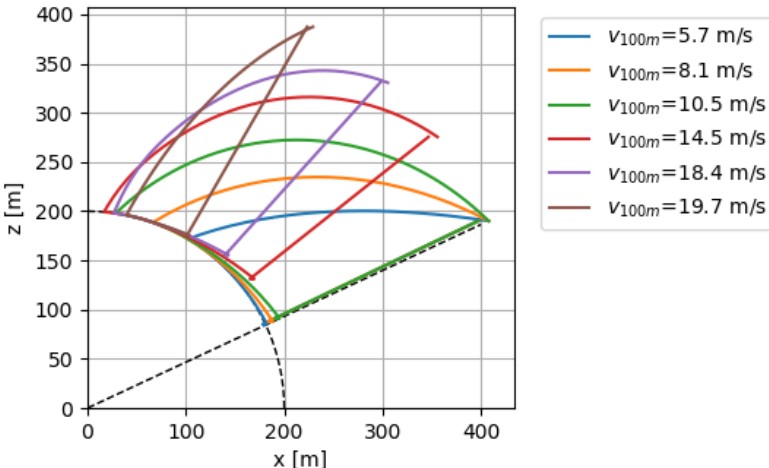

**Figure 19.** The optimal idealised cycle trajectories for six wind profiles with the same cluster-mean wind profile shape, i.e., that of cluster MMC-1, but different scaling. The wind speeds for which the trajectories are depicted highlight changes in the operational approach. The angle of inclination of the straight dotted line is the minimum elevation angle. The radius of the dotted quarter circle shows the fixed minimum tether length.

Above this wind speed, the curves flatten off and become different from one another. The MMC-3 and MMC-7 curves show the lowest and highest maximal mean cycle power, respectively. In conclusion, a pronounced low-level jet is favoured over a high-shear wind profile shape in terms of the power production of an AWE system.

## 5.2 Estimating the annual energy production

5  The previously derived power curves are used to calculate the average generated power of the AWE system:

$$\bar{P} = \sum_{i=1}^{n_\mathrm{c}} \int_0^\infty p_i(v_\mathrm{norm}) \cdot P_i(v_\mathrm{100m}) \, \mathrm{d}v_\mathrm{norm} \approx \sum_{i=1}^{n_\mathrm{c}} \sum_{j=1}^{n_\mathrm{b}} \frac{f_{i,j}}{n_\mathrm{s}} \cdot P_i(v_{j,\mathrm{100m}}) \quad , \tag{9}$$

in which the wind speed probability $p_i$ is a function of the normalisation wind speed $v_\mathrm{norm}$ used in the pre-processing, the maximal mean cycle power $P_i$ is a function of the wind speed at $100\,\mathrm{m}$ height $v_\mathrm{100m}$, both functions apply to the $i^\mathrm{th}$ cluster, and $n_\mathrm{c}$ is the number of clusters. The integral in the expression is solved numerically using $n_\mathrm{b} = 100$ wind speed bins between

10  the cut-in and cut-out of equal width. A large number of bins is used to mitigate numerical errors. In the resulting right hand side expression, the number of samples $n_\mathrm{s}$ is used to normalise the frequency $f_{i,j}$ of the $i^\mathrm{th}$ cluster and $j^\mathrm{th}$ bin, which is determined on the basis of the normalisation wind speeds. Consequently, the normalisation wind speed is used to express the bin limits. The argument of $P_i$ is the equivalent wind speed at $100\,\mathrm{m}$ height at the center of the $j^\mathrm{th}$ bin, which is derived using: $v_{j,\mathrm{100m}} = v_{j,\mathrm{norm}} \cdot \tilde{v}_{i,\mathrm{100m}}$, in which $\tilde{v}_{i,\mathrm{100m}}$ is the normalised wind speed at $100\,\mathrm{m}$ height of the $i^\mathrm{th}$ mean-cluster wind profile

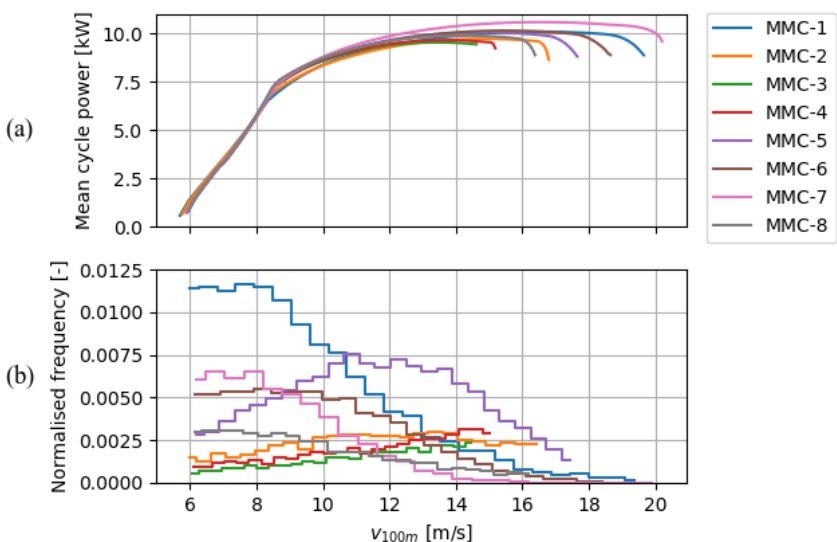

**Figure 20.** Power curves obtained by performance optimisations using the scaled cluster-mean wind profile shapes of MMC-1–8 (a). Wind speed distribution of the samples within each cluster using the full onshore dataset (b). Only the frequencies between cut-in and cut-out are depicted and every four wind speed bins are aggregated purely for illustrative purposes and shown as a single bin.

shape. The resulting wind speed distributions based on the normalised bin frequencies are shown in Fig. 20b. Multiplying the average generated power by the hours in a year gives the AEP estimate.

The AEP at the onshore location is evaluated for the MMC and ML cluster representations from Sects. 4.3 and 4.4, respectively. Moreover, the number of clusters used for the representations is varied to assess how many clusters are needed for the AEP to converge to a steady value, see Fig. 21. The trend for the MMC representation converges to around 36 MWh for a large number of clusters. In the following, we refer to the difference relative to the AEP at 32 clusters calculated using the MMC representation as the AEP error. For four or more clusters, the AEP error is within three percent and for 14 or more clusters, there is virtually no more variation in the AEP and the steady solution is reached. The error can be mostly attributed to the wind resource representation, but also the numerically obtained power curves and numerical integration introduce errors. The

AEP trend for the MMC representation converges faster than that for the ML representation, since the former is generated specifically for the evaluated location. The AEP error at 16 clusters for the MMC representation is similar to the AEP error at 32 clusters for the ML representation, which suggests that the ML representation needs twice the number clusters to yield the same accuracy as the MMC representation. Note that assumptions in the performance model also affect the convergence, e.g., neglecting the change of wind direction with height is expected to increase the convergence rate. How many clusters to

use depends on the application of the AEP calculation. In a preliminary design optimisation, where the computational cost is critical, four MMC-clusters may be a sensible choice. For more detailed design studies, 14 MMC-clusters would be more suitable.

Previously, at more than 50 wind speeds between cut-in and cut-out performance optimisations were performed to obtain each of the highly detailed power curves in Fig. 20. Half the number of optimisations yield similar level of detail with half the computational cost. Assuming that a four-cluster representation provides sufficient accuracy and 25 optimisations are used to generate a single power curve, 100 performance optimisations are needed for the AEP calculation. In comparison, an hourly brute force calculation needs 8760 optimisations per year. Already for a one-year calculation the number of optimisations required by the presented methodology is two orders of magnitude lower. The calculation for a longer period does not require more optimisations, however, it does increase the computational effort for the clustering.

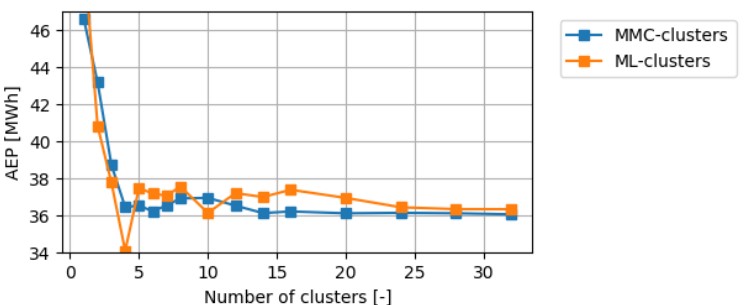

**Figure 21.** Comparison of the AEP convergence with increasing number of clusters at the onshore reference location for the MMC and ML cluster wind resource representations.

## 6 Conclusions

We have presented a methodology for including multiple wind profile shapes in a wind resource description. A data-driven approach is used to identify a set of wind profile shapes that characterises the wind resource. These shapes go beyond the height range for which conventional wind profile relationships are developed, such as the logarithmic profile. Moreover, they include non-monotonic wind profile shapes such as low-level jets. We demonstrated this methodology for an on- and offshore reference location using DOWA data. Subsequently, the resulting cluster wind resource representation for the onshore location has been used to estimate the AEP of a pumping AWE system.

To obtain the wind profile shapes of the DOWA samples, the wind profile of each sample is expressed relative to its wind velocity at the $100\,\mathrm{m}$ reference height and normalised. A PC analysis shows that three PCs already account for about $90\,\%$ of the variance in the dataset. The first and second PCs are very similar for the datasets of the onshore and offshore locations. The first PC mostly characterises wind veer, whereas the second PC mostly characterises wind shear. Moreover, the analysis reveals a natural structure of the data in the principal component space with two relatively dense groups of data points. The data points for the onshore location are more spread out, indicating a larger variety of wind profile shapes.

The dataset is partitioned using k-means clustering. The resulting cluster-mean wind profile shapes are used to approximate the vertical variation of the wind, yielding the cluster wind resource representation. This representation reduces the wide

variety of wind conditions in the DOWA dataset to a reasonable number of wind profile shapes. The accuracy of the representation using three or more clusters is already higher than that of a representation using logarithmic wind profiles. The eight cluster-mean wind profile shapes of the offshore representation include three monotonic profiles, four jet-like profiles, and an anticlockwise-turning, sharply-bent profile. Very similar cluster-mean wind profile shapes have been identified for the onshore location occurring under similar conditions. A single set of clusters is generated that is representative for the entire DOWA domain and used to analyse the spatial variability of the frequency of occurrence of the clusters. The cluster frequency maps indicate a clear distinction between onshore and offshore clusters. The sharply defined patterns in the frequency maps of the onshore clusters coincide with orographic features and thus suggest a strong relationship between the wind profile shape and orography.

The AEP of a flexible-kite, pumping AWE system is estimated using the onshore cluster representation. For each cluster-mean wind profile shape, a power curve is derived by using a quasi-steady model in power production optimisations. The highest power is found for the shape with a pronounced low-level jet. Together with the respective wind speed distributions, the power curves yield the AEP contributions of the clusters. The relationship between the estimated AEP and the number of site-specific clusters shows that the difference in AEP relative to the converged value is less than three percent for four or more clusters. For 14 or more clusters, there is virtually no more variation in the AEP estimation. For a four-cluster representation and using 25 optimisations for deriving the power curve of a single cluster, 100 optimisations are required for the AEP estimation against 8760 for an hourly brute-force calculation. The AEP estimation using clusters is thereby roughly two orders of magnitude faster.

The presented methodology has the capability to produce a single set of wind profile shapes that is valid for a large area. Such a set can facilitate the standardisation of wind conditions for which AWE systems are rated in terms of power production. Moreover, the multi-location cluster representation enables an assessment of which installation site is best for an AWE system in terms of its AEP, which makes this methodology a very powerful tool for project developers. In future work, the role of the performance model in estimating the AEP is further investigated.

*Author contributions.* Analysis and preparation of the paper were performed by MS, under the supervision of RS and SJW. PCK provided input on meteorology and data analysis and contributed to writing the result section.

*Competing interests.* The authors declare that they have no conflict of interest.

*Acknowledgements.* Without the publicly available Dutch Offshore Wind Atlas by the KNMI and the ERA5 dataset by the ECMWF, this work would not have been possible. Mark Schelbergen and Roland Schmehl have received financial support by the project REACH (H2020-FTIPilot-691173), funded by the European Union's Horizon 2020 research and innovation programme under grant agreement No. 691173,

and AWESCO (H2020-ITN-642682) funded by the European Union's Horizon 2020 research and innovation programme under the Marie Skłodowska-Curie grant agreement No. 642682.

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
