# Peer review of "Clustering wind profile shapes to estimate airborne wind energy production"

_Wind Energy Science, 2019_

## Referee Comment (RC1) · Markus Sommerfeld (Referee) · 6 Feb 2020

**1  General comments**

This paper is a useful contribution to better understand the wind energy potential of airborne wind energy systems (AWESs). The investigated onshore and offshore wind regimes make it especially interesting for regions close to the shore such as the Netherlands which this papers wind data is based on. Simulated Dutch Offshore Wind Atlas (DOWA) data is normalized, transformed using principal component analysis (PCA) and clustered to generate generalized wind profiles which are then scaled and fed into a quasi-steady AWES model to estimate power curves and annual energy production (AEP). However, this paper would benefit from a more detailed explanation, validation and justification of the described process. Following are some of the general questions and comments that require further explanation.

Language: Please revise the writing of this paper with regards to the usage of active voice to form shorter more concise sentences. Avoid sentences such as: " Finally, it is demonstrated how a set of wind profile shapes and their statistics can be used to estimate the AEP of a pumping AWE system" (page 1, line 9). Section 2 highlights some but not all text passages with passive voice. Remove repetitive sentences, combine sentences where possible. Avoid filler words and obvious wording such as "vertical wind profile". Some line breaks seem unnecessary as both the preceding and following paragraph are related or continue the same topic.

Figures: Many figures are very similar. The paper might benefit from focusing on one location and moving the figures showing the other locations to the appendix. More detailed captions would improve the understanding of your paper, especially for people who are skimming through the text or are reading the paper for the first time. Try to be consistent with the labels within figures as some figures use circles and others use squares.
Wind data: You are using several different data sets in your analysis which occasionally confuses the reader which could be because of the naming convention you chose: MMIJ, MMC and ML. A simpler naming such as "onshore" and "offshore" could help, especially when skimming through the paper.

I am sceptical about combining offshore and onshore wind data into one data set. It is my understanding that you want to simplify the energy estimation of AWES by creating a general purpose set of wind profile clusters. This could lead to results that are so generalized that their application is not valid in either situation. This averaging effect probably aggravates due to the small number of clusters and the temporal resolution of the data set. Comparing the mean normalized wind speed profiles in Fig. 2 and 10 (as well as Fig. 8 and 11) shows that these profiles are in deed fairly different especially at altitudes up to 300m which is well within the operating range shown in Fig. 18. This difference is further supported by the map in Fig. 15. Please show a validation of your approach (e.g. compare power outputs or wind profile shapes reconstructed against actual simulation) or explain in the text how you validated it.

The usage of data sets in your analysis is as follows:

- Section 3: Normalization, transformation and clustering process with **2D offshore** wind profiles

- Section 4: Compare normalized clustered **2D offshore, onshore, lidar** (7 lines of text) and entire domain

- Section 5: Calculate power curve for **1D scaled onshore** wind profiles

- Section 5: Compare AEP for **1D onshore** and entire domain

This leads to the repetition of similar plots, e.g. Fig.8, 11, 13, 14 which take up a lot of space and hardly additional information content. Therefore, my recommendation is to focus on either the onshore or offshore location, and compare final results such as

power curve and AEP with all data sets. You can choose to move the other figures to the appendix or remove them entirely. Similarly, I recommend removing the lidar section from this paper, because you do not use it in any analysis other than "... cluster profile shapes for both datasets are very similar." A deeper analysis of the differences and commonalities between the data sets would justify keeping all these figures if you choose to keep them.

While you go into great detail explaining the process of normalizing, applying PCA and clustering your wind data, very little detail is given on the denormalization of the wind profiles so they can be used for power estimation. Did you use the cluster centroids i.e. the mean of all the profiles which is not an actual profile that occurs in your data set? Similarly one can argue that the profiles derived from the $mean \pm PC * std$ shown in Fig. 2 and 10 are not actual profiles in the data set. Which range and step size did you use to denormalize the wind speed profiles? Is it possible to determine the wind speed range of each cluster or is this information lost due to normalization? If so it would be interesting to see this range in power curve plot. Validating your approach against actual simulated profiles by comparing energy or power curves would add to the credibility of this paper. This could include a comparison against non-normalized, clustered wind profiles or not PCA transformed profiles or standard log profiles with assumed Weibull distribution etc... Please justify why you go through all this process for 2D wind profiles if you just use the wind speed as an input into your AWES model.

PCA: Which PC and std do you use to derive the profiles column 3 and for of Fig. 2 and 10? Maybe a better representation would be to plot the mean, $mean \pm PC * std$ in one plot with a shaded area in between to highlight the range of possible speeds / velocities? It is my understanding that the justification for using PCA is that it accelerates the clustering process. Is clustering the bottle neck of your analysis or is it the optimization? How does the end result vary comparing transformed and not transformed profiles?

Clustering: Please justfy in the text why you only use the cluster centroids for your analysis and how which uncertainties / inaccuracy this causes? How much does the profile shape within one cluster vary and is this variation reduces by normalizing the data? Clustering labels are not in order. How did you sort them or determine analogies between clusters of different data sets (see table 2)? Comparing clusters like this gives the impression that they have the same profile shape. Cluster centroids are the arithmetic mean of all the data points within this cluster. Therefore, adding or removing a data point changes the outcome of the entire process. How can you compare these clusters in table 2? If only 8 clusters are enough, then why not use existing stability classification (like table 1 in https://www.adv-sci-res.net/6/155/2011/asr-6-155-2011.pdf or https://www.wind-energ-sci.net/4/563/2019/wes-4-563-2019.pdf) based on Obukhov length or Richardson number which are widely accepted and a common meteorological classification?

**2 Specific comments including technical corrections**

Title: I recommend a more specific title including information such as: ground-gen / pumping mode, mesoscale wind data, but understand that it is personal preference.

2.1 Abstract

page 1

> line 1: Why not use AWES abbreviating for Airborne Wind Energy System as commonly used in the community and literature
> line 6: Introduce abbreviation DOWA here.
> line 10: Mention the derivation of power curves.

line 10: Mention the location where you estimate AEP or add AEP analysis for all sites since it sounds like multiple AEP estimates are compared.

line 11: Define or rewrite the sentence so that it is clear that you compare the amount of clusters necessary to estimate AEP and use concrete results rather than "within a few percent".

**2.2 Introduction**

page 1

line 13: AWE already defined in Abstract in line 1.

line 14: Many different concepts exist with varying (anticipated) operating altitudes. Either remove subordinate clause "typically in a range up to 500 m" or explain / reference what this assumption is based on.

line 14: Add reference for stronger and more persistent winds.

line 18: This is not the definition of surface layer. I would stay away from concrete number as the height of the surface layer varies a lot depending on atmospheric condition. Remove or replace with: in the order of tens of meters depending on atmospheric stability (http://glossary.ametsoc.org/wiki/Surface_boundary_layer)

page 2

line 1: Low-level jets are a known phenomena and have been discovered a while ago. Rewrite: "Recent studies have identified numerous low-level jets"

line 4: Remove: "more"

line 9: What kind of "performance calculation"?

line 9: Add reference to COSMO-DE

line 10 & 11: Add: "shape" after wind profile

line 14: Add reference to MERRA-2

line 14 & 16: Malz et. al b) referenced before a).

line 15: Sentence on the reduction of computational cost and choice of initialization seem unnecessary for this paper.

line 20: Add reference and expansion of WRF

line 26: Remove "such a historical dataset". Does not contribute to the understanding or quality of this paper. What makes this data set so special?

line 34: Add reference to DOWA, ERA5 and expansion of LiDAR (also, check WES house standards on capitalization of abbreviations: https://www.wind-energy-science.net/for_authors/manuscript_preparation.html. One feedback I got was: "WRF is a defined model name (the same applies to the ARW), but lidar is a general term and is therefore not capitalized according to our house standards."

page 3

line 1: Is the DOWA data set comprised of met mast data? I thought it is derived from reanalysis models which had various measurements assimilated.

line 2: Sentence is a little confusing as it suggests that met mast data is somehow involved. Maybe chose a different naming convention e.g. onshore, offshore.

line 2: "The procedure" is confusing as the previous sentence describes comparing DOWA and LiDAR and not generating a set of representative wind speed profiles from clusters

line 5: More accurate than log profiles or uniform wind? Is it actually more accurate and did you validate the improvement?

line 7: Active voice

**2.3 Wind data sets**

page 3

- line 10: Combine first 2 sentences.
- line 16: Repetitive sentence: "Both long-term ..."
- line 17: In what way "sparser" lidar? spatial, temporal?
- line 21: Repetitive sentence: "... coast in the North Sea". Also shorten the sentence e.g. "The selected offshore location, met mast IJmuiden, is located 85 km off the Dutch coast."
- line 21: Write out the wind direction as it doesn't safe lots of space but adds clarity
- line 21: Why not use the Cabauw or IJmuiden met masts for validation as well? I thought that was one of the reasons why you chose this location.

**2.3.1 ERA5**

page 3

- line 26: Define and add reference to ERA5 and ECMWF.
- line 27: I don't think the year in brackets as well as the information that more data from years back will be available soon is necessary.
- line 31: Replace outputs with a different verb: produces, uses, calculates and rewrite sentence to be more concise

page 4

- figure 1: More explanation in caption

figure 1: How and why did you choose the sample location? Are these representative locations?

figure 1: Is it important that you use met mast locations as you don't you met mast data at all.

line 1: Remove sentence: The long-term wind climate is not important for this study as no long-term predictions were made.

line 2: You use ERA5 to determine atmospheric stability. Does DOWA not provide the necessary data?

line 2: Stability is only used for cluster statistics in Fig. 9 and similar. Please expand this analysis and the relationship between clusters and stability.

line 3: DOWA defined on page 2

**2.3.2 Dutch offshore wind atlas**

**page 4**

title : Capitalization different from previous DOWA

line 6: How downscaled? Is it not more the extraction of specific values from a data set?

line 6: Add reference to model and / or move link to footnotes

line 10: Add reference

line 11: Remove 2nd "than" and "alone"

line 12: Define or paraphrase "ASCAT" and "mode-s ehs". Add reference

line 13: Sentence about website unnecessary

line 14: Remove "additionally" and conjugate "show" or add et. al

line 15: LLJs are not anomalous

**2.3.3 lidar observations**

page 5

- title : Check capitalization and abbreviation rules for lidar
- line 1: Rewrite. I don't know what you want to say with this sentence.
- line 4: ECN?
- line 5: Add link as reference and / or footnotes
- line 6: Where is difference between "clock hour" and hourly?
- line 7: Active voice
- line 8: Only time you use "data set" instead of "dataset"

**2.4 Clustering procedure**

page 5

- line 12: Mention that you do the same for MMC
- line 13: How normalized? Add PCA abbreviation
- line 14: rewrite "for choosing", active voice e.g. the number of clusters is chosen based on the clustering performance...

**2.4.1 Preprocessing of the wind data**

page 5

- line 16: Shorten sentence, mention use of all time steps, entirety sounds like more than it is.

line 20: Is that 90th percentile of each time step or of all data points of this altitude or of all data points at 100 m?

line 20: Line break not necessary

line 21: Why did you choose this normalization and why normalize in general if you expect that it will not lead to good results for low winds?

line 21: Active voice. Do you only expect eccentric profiles or did you actually observe them?

line 24: How do you implement them if they are omitted?

2.4.2   Principal component analysis of the wind profile shape dataset

page 5

line 27: Defined above

page 6

line 4: "for every wind direction" = omnidirectional

line 4: The difference to ...

line 5: "logarithmic profile representation of the wind environment" = logarithmic wind profile. Remove one "logarithmic profile". Introduce $z_0$ here. Why is $z_0 = 0.0002$?

line 6: "vertical" is understood, wind speed is always magnitude

line 6: Add variable name $u(z)$ and reference to equation

line 8: Add reference or explanation to Obukhov length

line 13: Explain relationship of $L$ and atmospheric stability. Does this sentence mean that the average of multiple years of wind data fits an unstable log profile? Are you fitting wind speed or just parallel component?

line 13: Add reference to "Theory". Add "mean wind profile...", remove "in the" & "direction".

line 13: Add "(top-view in the bottom left panel)"

line 14: Remove "of the wind profile shape"

line 22 & 23: Remove text in parentheses

line 24: Add reference

line 28: Active voice

line 31: Isn't this a general feature of PC? Remove "Note that"?

page 7

figure 2: What do the number 1-4 in bottom boxes mean? Add normalized to vertical wind profile. Does second column show the mean or is it the orientation of PC axis with height? What does it mean to multiply PC with std? Which std do you multiply with? It might make sense to show the std of each PC in a table. Plots in column 2 are not on the same x-axis. Please explain how PC1 and 2 rotate over altitude.

**2.4.3   Number of clusters**

page 7

line 6: Either active voice or general statement in which case you should use "a" instead of "the"

line 8: Remove parenthesis. Replace "all" with "each"

line 10: Add reference

page 8

figure 3: Explain legend abbreviations and subset in caption. What is the MMIJ subset?

figure 4: why $*$ on x- & y-axis label? Are the markers the orange circles? Do the numbers 1-4 correspond to the clusters shown in Fig. 2? If so mention this in the caption.

figure 4: Maybe replace "wind profile shape" with wind velocity profile.

line 1: What is "itk" ? Number of clusters is usually the $k$ in k-means clustering

line 5: Delete if not shown. If you keep it in explain "over-fitting" in this context. Probably not happening because of (relatively long) temporal averaging.

line 7: How is the silhouette score calculated? Adding silhouette score and WCSS equation would make it easier to understand. Is a score of 1, 0 or -1 better?

line 9: Remove repeated subject "mean silhouette score". How do you interpret the decreasing trend? Explain if you mention it, the fact that it is decreasing is obvious.

page 9

figure 5: Which data set do these results belong to: MMIJ, MMC, ML? Are these results the same for all data sets? Expand on the captions.

figure 5 a): y-axis label: Isn't it the "sum of squared distances" ?

figure 5 a): Add a vertical line at $k = 8$ to highlight your choice.

figure 5 b): Explain non-linear grid lines in caption. Why no vertical grid lines? "...error over height for logarithmic profiles...(...and four different stability...)", "...three exemplary clusters". How is the error defined?

line 1: So it would be best to use only 2 clusters to represent the many different wind conditions even though the sum of square distance is way higher and 4

log-fits out perform it? What defines a good silhouette score? Wouldn't it be a fair comparison to also vary the number of log-fits over the x-axis? Why 4 log-fits?

line 2: Why do you show the different approaches to get best k and then use AEP without showing it here?

line 5: Shorten to"filtered wind profile data". Parentheses are repeating what has been said before, remove.

line 5: Remove: "Next, it is..."

line 8: Equation would make it easier to understand.

line 11: More combined than individually, otherwise 2 lines.

line 16: How did you choose these $L$ values? What are the assumed ranges of $L$ associated with certain atmospheric stability?

page 10

figure 6: Why vertical grid line in center of plot? Showing the x-axis as percentage of total data would be better. Briefly explain what positive and negative values mean?

line 5: Is this weighting intended? Why not linear interpolation in z?

line 6: It seems like this is not the only reason. Resolution is high up to 200 m.

line 9: I assume that a low number of clusters is enough to capture variations within the hourly data set which is why I would recommend mentioning that this choice is specific for your temporal resolution.

line 14: Move the last sentence up to the section where you first introduce clusters.

**2.5 Wind resource representation based on clusters**

**2.5.1 Interpretation of prevailing wind profile shapes at MMIJ**

page 11

- line 10: Are these profiles the centroids of the clusters you calculated before?
- line 13: Are these the Obukhov lengths? Which ranges do you associate with each stability condition?
- line 12: Replace "...moving between..."
- line 13: What are the number in parentheses? Add variable.
- line 13: Move line break as veering clearly refers to the previous paragraph.
- line 21: Fig 7 is mentioned after Fig 8. Active voice
- line 22: Also active voice: "Examining the five PC coordinates in table 1 ...."
- line 26: How can that be deduced? Define filter e.g. "low-speed filtered..." as readers might have forgotten or skipped the previous sections.
- line 28: Define "calm wind"
- line 29: These 2 sentences are more general and introduce the data set. Move up before previous sentences.

page 12

- equation 2: Why are you using different $\Delta z$ for $u$. $v$, $\theta$ within this equation? Are $u$. $v$, $\theta$ on different altitudes?
- line 6: Why not rewrite Eq. 2 to include absolute temperature instead? Explain how it is used. I guess instead of virtual potential temperature.

line  6: Isn't humidity high close to the shore / offshore and would the effect not be considerable?

line  8: Add "Positive $Ri_B$ values"

table  1: Do these values relate to Figure 2?

page  13

table  2:The cluster centroid shape will change based on the underlying data, i.e. filtered. It seems that you assume that clusters between data sets are same, e.g. cluster 1 is same through out all data sets. Did you sort or determine similarity between cluster centroids of different data sets? How similar are PC transformed and normalized wind data? Can you quantify how similar / different they are?

figure  7: Nice visualization. Why $*$ on x and y label?

page  14

figure  8: Is this the filtered or unfiltered data set? Do all x-axis use the same range, because different label? Explain more in caption, i.e. why $\hat{v}$ over variable, where is the origin of the hodograph / increasing $z$ direction? Better spread out all the graphs so that they appear after mentioned in text. Do you need all these graphs here? Maybe move some to the appendix.

page  15

figure  9: Nice way of comparing the data set. Might be a bit overwhelming at first though. Is this the entire data set or filtered? Where are these $Ri$ bins coming from? Add reference. What do you mean by "Bins have the same overall frequency" ? Wouldn't that mean that all the bars have same height? Do you mean bin width? Use the same comprehensible bin width for all subfigures (especially wind speed and $Ri$ seem arbitrarily chosen) in

figures 9 and 12. Maybe add an "offshore" and "onshore" to data set labels for clarity. A better and easier understandable metric would be atmospheric stability (i.e stable, unstable) instead of using $Ri$ number.

page 16

line 1: Change figure order or reference which figure you mean after 3 pages of figures. "... over the 10 year timeframe".

line 2: Why is this a prerequisite? You have temporal variations on all time scales. But I guess this way you make sure that your data is not based on outliers.

line 3: Remove sentence and line break. Maybe change order. Why first talk about 9 a) and then all the other figures separately ?

line 5: Rewrite: "... not so frequently strong". According to figure 9, cluster 1 has an almost even occurrence through all wind speed ranges.

line 5: "are more frequent".

line 7: Add reference to figure which shows cluster shape.

line 9: "well mixed". How about shear?

line 26: Remove line break as both paragraphs are about cluster 6,7.

2.5.2   Comparison with an onshore location

page 17

line 5: Vertical is understood. Better write: "The mean normalized wind speed profile...".

line 8: Same as line 5

line 11: Removing "of the two locations ... " makes the sentence easier to understand.

line 19: Add the variable to values in parentheses

line 20: So is there no cluster that corresponds to stable stratification without LLJ? If so wouldn't that be unusual?

line 22: "however" and "just" seem like filler words. You could shorten the sentence

line 24: Active voice.

line 26: Sentence hard to read. Shorten i.e. "This affect is caused by the lower heat capacity of the land surface which promotes a more immediate heat transfer to or from the atmosphere."

line 30: What is the point of comparing diurnal to seasonal cycles? They are caused by entirely different effects and play out over vastly different timescales.

line 32: What does overall bin frequency mean and why does it have to be equal? It sounds like you are varying your classification to get a certain result rather than try to characterize actualphysical effects.

page 18

figure 10,11,12: More explanation for readers who just skim through the text or just look at figures, i.e. onshore, normalised wind speed, numbers and explain PC*std. Consider moving some figures to the appendix.

2.5.3  Validation with LiDAR observations

page 21

line 1: This paragraph seems rather unnecessary. Very short, no mayor inside, only that results are similar.

line 2: replace "investigate" with "show"..

**2.5.4 Spatial distribution of wind profile shapes**

page 21

line 10: How did you chose these sample locations in Fig.1? Are 45 grid points the entire domain? Does your selection affect / bias the results? Line break not needed.

line 12: Explain cluster mapping. Why only 8 clusters again? reference previous chapter.

line 13: How can you apply the same mapping to a new data set?

line 16: Remove "be" and shorten / rewrite sentence.

line 18: Remove sentence "Since cluster 1..."

page 22

table 3: Is this a necessary table? offshore / onshore is not a sufficient classification of wind. How did you match clusters from different data sets? Are they the exact same clusters and if not how similar are they?

page 23

figure 13: Is this a necessary figure? Dashed lines missing in legend. Is this based on hourly average lidar data? Do these numbers in circles have any meaning?

page 24

figure 14: Rewrite: many "the" in caption. Maybe a bit more explanation, i.e. which locations are represented.

page 25

 figure 15: Add details to captions. Add ML abbreviation. Add reference to fig. 14 for info on clusters. How did you "map" to a new data set? Meaning of numbers? Use consistent frequency ranges for comparison or justify why you did not.

 figure 15: This clear division between on- and off-shore profiles would justify separating the analysis. Wouldn't this lead to better more detailed results and insights? Please validate and quantify how much information you lose by clustering everything this way rather than off- and onshore individually.

**2.6 Fast AWE production estimation based on historical wind data**

page 26

 title : What is "fast" about this analysis?

 line 1: Rather "estimating" than "calculating"

 line 2: You went through all the process of explaining the clustering process using MMIJ data set than introduce MMC, lidar and ML to now only use MMC? Why not focus on MMC entirely or apply your power estimation to all data sets?

**2.6.1 Determining power curves for AWE systems operated in pumping mode**

page 26

line 10: "... differs between ...". Add reference to flexible-kite sentence.

line 15: Rewrite sentence: "...is ended ... is depowered... is steered ....". Remove: "the" in front of zenith. Add: "... to the starting position of the traction phase"

line 17: Check capitalization rules for abbreviations.

line 18: Rewrite sentence: "... moves the kite along an idealised flight path conform a series..." ?

line 19: Doesn't the limitation to lightweight membrane kites mean that the approach is only applicable to soft kites and not "any kind of pumping AWE system" (line 12)?

line 24: I agree that it is justified to use only magnitude wind speed profiles, but why go through all the process of clustering 2D profiles in the first place?

line 26: Explain in more detail how do you derive the power curve and how you scale the normalized profiles.

line 28: Active voice. Mention that aero coefficients are assumed to be constant.

line 31: What means "sufficiently high"? How is the tether reeling speed during reel-in and -out? Is it a constraint or output of the optimization?

line 31: Rewrite to include tether force constraint and that it corresponds to setting a fixed max tether diameter.

page 27

table 4: More descriptive caption.

line 1: What are your control variables? Table 5 contains constraints that you are keeping constant I assume. Maybe write out the optimization formulation?

line 2 & 3: What is the message of these sentences? Optimizations basically always have active constraints. Why do you have to lower the tether force?

What if an increase in elevation angle leads to an increase in wind speed and therefore force?

line 3: Define effective pumping length.

line 8: Rewrite so that you describe where cut-in and cut-out comes from in this sentence or same paragraph.

page 28

table 5: What are these constraints based on? Is actual tether length $l_{min} + l_{pumping}$ ? Seems like a list of constraints. What are all the constraints? Are these realistic values (add reference)?

line 3: How do you define "steady flight states"? What are the states? Active voice.

line 4: What about reducing lift / flight speed to achieve one figure-of-eight? To which ground station reeling speeds do your constraints correspond? Explain why you chose this constraint.

line 6: Explain the developed module and assumptions etc.

line 9: This sentence explains cut-out limit again, same as line 4.

line 11: Rewrite sentence. "The corresponding cut-in and cut-out wind profiles are shown in figure 17" or so. How did you scale the normalized wind speeds?

line 12: Is the critical height of 80 m related to the minimal tether length, size of device or other parameters?

line 19 & figure 24: If wind speeds at 80 and 300m are a sensible choice why do you use $v_{100}$ ?

line 25: add "... power curves..."

line 26: Check comma placement

page 29

figure 17: You sometimes use left and right and other times a and b for sub-figures, choose one. More descriptive caption. How did you scale the profiles?

figure 18: Change to "Height" instead of "z" as you always used height before. Add black dashed lines in legend. Why disconnected lines at end of traction phase? Why did you choose this strange $v_{100}$ values? Why not $v_{80}$ instead? Remove: "traction" and "constant" from captions.

line 1: It would be interesting to see reeling speeds, tether force and other variables during one production cycle.

line 4: Why is a profile with LLJ the last to reach cut-out speeds? I would have expected it to cut-out earlier. Is it because of the height of the LLJ?

2.6.2 Estimating the Annual Energy Production

page 30

figure 19: Seems like the profile shape has almost no impact especially at lower wind speeds where the power ramps up. Remove: "... that are ...". What is the actually wind speed range of each non-normalized cluster? How come the power curve does not plateau and bend down before cut-out?

line 4: $f$ for frequency of occurrence rather than $p$ for probability as I assume it is based on the data you used and not a model like Weibull. It would be great if you could show the distribution of wind speed frequency. Remove apostrophe: "... is the systems power curve...."

line 7: No line break needed.

line 8: How constructed? Equation?

page 31

figure 20: How about MMMIJ? How does the AEP and power curve compare to log profiles with Weibull distribution?

line 3: Did you use 50 calculations to get the power curves of 8 clusters, i.e. more than 6 wind speeds per cluster?

**2.7 Conclusions**

page 31

line 7: What is "fast" about the calculation? Do you mean simplified? Rewrite e.g.: "... used to estimate AEP for a simplified pumping-mode AWES ..."

line 11: Shorten to: "...simple logarithmic profiles...". Would be good to compare power curve and AEP against these log profiles.

line 12: For hourly average profiles. What could be the impact of higher resolution data?

line 13 & 14: Shorten: Both locations show similar results.

line 14: Which samples do you refer to, all, MMC, MMIJ?

line 21: I am not convinced by this conclusion. Why does profile shape similarity proof that clustering is able to differentiate between atmospheric conditions? Also which conditions? If only stable and unstable two clusters might be enough.

line 21: Is your process able to determine atmospheric stability (with a certain confidence) solely based on wind profile shape? If so that would be a great addition to your analysis.

page 32

line 3: Which wind resource presentation?

line 4: How did you get a distribution from profiles? Would be interesting to see which cluster/ time of year or day contributes how much to AEP.

line 7: How do 25 optimizations relate to 4 clusters or wind speeds?

line 11: How high is the error in comparison to single location clustering?

line 12: Add line break in front of "In the future..."

---

## Referee Comment (RC2) · Anonymous Referee #2 · 17 Feb 2020

**1   General comments**

Paper "Clustering wind profile shapes to estimate airborne wind energy production" describes statistical analysis of wind profiles in order to identify typical shapes of wind profiles that are later used to optimize Airborne Wind Energy systems. The paper is interesting, has scientific novelty and describes some interesting, meaningful and useful results. However, I would suggest that the paper would benefit from some careful editing. The overall quality of English is good, but the presentation of information could use some improvement. I had to apply some effort to follow the argument set out by the authors. Some important procedures (normalization of the wind profiles, fitting of the Obukhov length) are not explained sufficiently, while other aspects of the work that

do not lead to any significant conclusions are explained at length.

**2 Specific comments**

- P2L6: "Computationally expensive, brute force calculations". How expensive are the calculations? Can they be done in a few hours on a desktop computer, or is HPC required? An approximate estimation could be added here for the general reader.

- P3L10-L25. Here the word "reanalysis" is used to refer both to ERA5 and DOWA. Although formally DOWA fits the definition of reanalysis, in practice, at least in my experience, the word "reanalysis" is used exclusively for global reanalysis, such as ERA5, ERA-Interim, MERRA, etc. I would suggest using term "modelled datasets" or something similar, to refer to ERA5 and DOWA at the same time, to avoid confusing the reader.

- P5L6: Hourly averages are calculated for the LiDAR data. Why? For the rest of the paper modelling results with hourly resolution are used, which are usually interpreted as the "instantaneous" wind speed, of course taking into account the fact that the mesoscale model cannot represent the turbulent fluctuations. Are the hourly averages comparable with model data?

- P6L9: In the paper vertical wind speed profiles are fitted using logarithmic profile that has been corrected for stability. From the context I understand that Obukhov length L is the parameter that is being changed during the fitting procedure, but a more precise description of the methodology how the Obukhov length is acquired would be beneficial. If I understand correctly, the same procedure is applied for fitting the instantaneous and long-term mean profiles in Fig. 2. I would like to see some arguments why long-term means can be treated in the way authors do it

here. I refer the authors to the paper "Kelly, Mark, and Sven-Erik Gryning. "Long-term mean wind profiles based on similarity theory." Boundary-layer meteorology 136.3 (2010): 377-390." for a discussion about long-term stability correction of wind profiles.

- P5L18: Do I understand correctly that the parallel and perpendicular components are calculated for a different reference wind direction in each sample, namely, in each sample the wind velocity at 100 m will have only the parallel component? If so, please expand the description of this procedure in the text.

- P5L19-20 "each sample is normalized by the 90th percentile of its wind speeds at each height". This sentence is highly confusing, because one interpretation could be that at each height the distribution of wind speeds is constructed, and each height has its own normalization speed calculated as the 90th percentile of all wind speeds in given height. I suggest expanding the description of normalization to remove ambiguity.

- I think that Figure 2 is not successful in conveying the important information in an effective manner. The horizontal axis for PC1 and PC2 are not the same. The hodograph is small and the markings (dotted line vs uninterrupted line) are not explained. However, my biggest problem is the fact that the x-axis is chosen in such a way that the plots for parallel component and magnitude of the wind, which arguable are the most important features of the profile, are very small. In order to understand the results and conclusions I had to look at profiles that had the width of less than a centimeter (when printed out). I would like the most important features of profile to be plotted large and easy to understand. I would recommend rearranging Figure 2, dividing it into separate figures. To keep paper at a reasonable length, I suggest editing and shortening further sections of the manuscript. These issues continue in further Figures, such as Fig. 8. etc. I recommend focusing on the important parts of the graphs, making them large

and omitting less important information for brevity.

- Section 3.3. requires some editing. Many complicated metrics are introduced, but only briefly, which does not allow the reader to follow the discussion easily. For instance, the fact that higher values of silhouette score mean better similarity within cluster, should be mentioned when the metric is first introduced (P8L10), or at the start of the discussion paragraph (e.g. P10L12)

- P8L1: Why is "itk" used as a parameter name for the number of clusters? It is used twice and then never used again, for instance, in Fig. 5.a. Maybe it is better not to introduce such variable at all.

- Fig.5.(a). If the "cluster mag" and "cluster 2d" lines are identical, then maybe one of them can be omitted?

- Figure 9 and Figure 12: (a) panel could be omitted without loss of important information – all years are the same. Please make the rest of the plots larger – again I was forced to concentrate on very small portions of plots to arrive at conclusions. Maybe Figures 9 and 12 can be combined to save space for legends. I would also suggest using the same Richardson number bins for both figures to make interpretation easier.

- The discussion of filtered vs full dataset, e.g. Table 2, could be omitted to shorten the paper.

- Two different measures of stability are used in the paper – L (Obukhov length) and bulk Richardson number. For clusters (2) and (3) in Fig.8. L values that correspond to neutral stability conditions are reported (P11L13). From Fig.9. and the discussion (P16L11-L17) they are associated with stable conditions. How do authors explain this discrepancy? Additionally, I would like to point out that clusters

(2) and (3) are associated with higher windspeeds, and higher windspeeds typically are associated with higher frequency of neutral conditions, see for instance, Holtslag et al. 2014.

- P11L13: Fitted Obukhov lengths are only reported for clusters (1)- (3). I understand why they should not be reported for cases where the fit cannot reproduce the shape of the profile, e.g. cluster (7). But what are the fitted Obukhov lengths for other clusters, such as (5) and (8)?

- The authors claim that clusters 4-7 indicate potential low-level jets. I do not disagree, but in my opinion, the evidence presented is not strong enough and is somewhat circumstantial. Mostly, because the amplitude of wind speed maxima in Figures 8 and 11 is quite small (except for cluster (7) where I agree that the jet-like profile is quite distinct). Sometimes in literature it is required that wind speed in jets is at least 3 m/s higher than the surrounding flow. Due to the normalization procedure it is hard to estimate how pronounced are these wind speed maxima. The authors' position could be strengthened if more robust evidence to associate the clusters with jets could be provided – a case study or some wind speed profiles that are not normalized and show how distinct the jet actually is.

- If the conditions for cluster (5) MMIJ DOWA are analyzed (Fig. 9), than one could conclude that they are very similar to cluster (1) for MMIJ DOWA, with the exception of wind direction, so all the interpretation for conditions for cluster (1) MMIJ DOWA could apply to cluster (5) as well, with the exception that the upwind flow is advected over the water for shorter time, as the prevailing directions for cluster (5) are from South. In fact, in PC1-PC2 axis (Fig. 7) the cluster (5) is next to cluster (1). The problem is that for MMC such conditions are impossible and therefore cluster (5) for MMC is not the same as cluster (5) for MMIJ, nor in the shape of the profile and nor in the location of cluster in the PC1-PC2 axis. Therefore, I cannot agree with the statement that cluster (5) for MMC is similar to

MMIJ (P17L20).

- Table 3, column "Most frequent area". What is this classification based on? Are the orography and land use data used in DOWA available? If so, then adding this information to Figure 15 as a plot would strengthen the authors' argument, because currently, although I agree that features seen in Figure 15 are probably related to orography, again I would argue that more evidence is needed to support the authors' claims. I would not expect the general reader to be familiar with the topography of the Netherlands and Germany.

**3 Technical comments**

- Paper referenced as "Sommerfeld et al.": "Improving mid-altitude mesoscale wind speed forecasts using LiDAR based observation nudging for Airborne Wind Energy Systems" does not seem to contain anything related to k-means clustering. Maybe a different paper with the same first author is meant instead: Sommerfeld, Markus, et al. "LiDAR based characterization of mid altitude wind conditions for airborne wind energy systems." Wind Energy 22.8 (2019): 1101-1120.

- Order of figures. Figure 8 is referenced in text before Figure 7. Figure 8 is referenced in P11L10. Figure 7 is referenced in P11L21.

- P1L19: "Deviating profiles are likely to occur". I suggest "deviations from the expected profile shape are likely to occur".

- P4L11: "Better representation of the coastal morphology". "Coastal morphology" is the study of natural processes that change the shape of coastline, e.g., erosion or sediment transport. I suggest using "coastline" or "better resolution of coastline" instead.

- P16L9: "the wind profile is typically will mixed". Probably, "well mixed".

---

## Author Comment (AC1) · 20 Apr 2020

Thank you for the comprehensive comments. We feel that they were very helpful for increasing the quality of the paper to the current level. Your comments, together with those of referee #2, led to a thorough revision of the paper. The most important changes to the paper include:

1. Including information on orography

2. Discarding the lidar discussion

3. Using one stability metric: the Obukhov length, and corresponding classification for identifying stability trends within the clusters

4. Fitting logarithmic profiles only to the lower part, i.e. <200 m, of the mean profile shapes

We respond to the referee comments by including our answers below the original comments. Our answers are preceded by one or both of the following labels:

AR = author's response

AC = author's changes in manuscript

**1 General comments**

This paper is a useful contribution to better understand the wind energy potential of airborne wind energy systems (AWESs). The investigated onshore and offshore wind regimes make it especially interesting for regions close to the shore such as the Netherlands which this papers wind data is based on. Simulated Dutch Offshore Wind Atlas (DOWA) data is normalized, transformed using principal component analysis (PCA) and clustered to generate generalized wind profiles which are then scaled and fed into a quasi-steady AWES model to estimate power curves and annual energy production (AEP). However, this paper would benefit from a more detailed explanation, validation and justification of the described process. Following are some of the general questions and comments that require further explanation.

Language: Please revise the writing of this paper with regards to the usage of active voice to form shorter more concise sentences. Avoid sentences such as: " Finally, it is demonstrated how a set of wind profile shapes and their statistics can be used to estimate the AEP of a pumping AWE system" (page 1, line 9). Section

2 highlights some but not all text passages with passive voice. Remove repetitive sentences, combine sentences where possible. Avoid filler words and obvious wording such as "vertical wind profile". Some line breaks seem unnecessary as both the preceding and following paragraph are related or continue the same topic.

[AR] We have tried to incorporate the active voice as much as possible based on your comments. It is however not always used, as sometimes it feels less appropriate. We feel that the language has improved substantially, e.g., repetitive sentences and obvious words are omitted and the phrasing has become more precise.

Figures: Many figures are very similar. The paper might benefit from focusing on one location and moving the figures showing the other locations to the appendix. More detailed captions would improve the understanding of your paper, especially for people who are skimming through the text or are reading the paper for the first time. Try to be consistent with the labels within figures as some figures use circles and others use squares.

[AR] The clusters of the offshore, onshore and multi-location analyses are considered to be the core of the paper, therefore, putting them in the appendix would not be appropriate. Please find more explanation later on in this document.

[AC] A more consistent labelling is introduced. Now, only the clusters are referred to using the acronyms: MMIJ, MMC, and ML. We have made the captions a lot more informative. Furthermore, we have discarded the lidar discussion as it was not a strong validation and introduced an additional 1-page plot with 8 profile shapes.

Wind data: You are using several different data sets in your analysis which occasionally confuses the reader which could be because of the naming convention you chose: MMIJ, MMC and ML. A simpler naming such as "onshore" and "offshore" could help, especially when skimming through the paper.

[AC] Now, the locations of the met masts IJmuiden and Cabauw are referred to as the off- and onshore reference locations.

I am sceptical about combining offshore and onshore wind data into one data set. It is my understanding that you want to simplify the energy estimation of AWES by creating a general purpose set of wind profile clusters. This could lead to results that are so generalized that their application is not valid in either situation. This averaging effect probably aggravates due to the small number of clusters and the temporal resolution of the data set. Comparing the mean normalized wind speed profiles in Fig. 2 and 10 (as well as Fig. 8 and 11) shows that these profiles are in deed fairly different especially at altitudes up to 300m which is well within the operating range shown in Fig. 18. This difference is further supported by the map in Fig. 15. Please show a validation of your approach (e.g. compare power outputs or wind profile shapes reconstructed against actual simulation) or explain in the text how you validated it.

[AR] The aim with using the larger combined onshore and offshore dataset was to highlight how the prevailing profiles do vary depending on the terrain. We have tried to show this by relating the cluster profiles so obtained with those of the individual onshore and offshore analyses. For a detailed performance assessment, indeed we would suggest a more tailored (offshore or onshore) clustering approach which is why we used the conditions specifically at Cabauw for illustrating the AWE AEP assessment.

[AC] We have made some changes in the text at the beginning of Section 4.4 to try and clarify the rationale for the large area clustering analysis.

The usage of data sets in your analysis is as follows:

- Section 3: Normalization, transformation and clustering process with **2D offshore** wind profiles

- Section 4: Compare normalized clustered **2D offshore, onshore, lidar** (7 lines

of text) and entire domain

- Section 5: Calculate power curve for **1D scaled onshore** wind profiles

- Section 5: Compare AEP for **1D onshore** and entire domain

This leads to the repetition of similar plots, e.g. Fig.8, 11, 13, 14 which take up a lot of space and hardly additional information content. Therefore, my recommendation is to focus on either the onshore or offshore location, and compare final results such as power curve and AEP with all data sets. You can choose to move the other figures to the appendix or remove them entirely. Similarly, I recommend removing the lidar section from this paper, because you do not use it in any analysis other than "... cluster profile shapes for both datasets are very similar." A deeper analysis of the differences and commonalities between the data sets would justify keeping all these figures if you choose to keep them.

[AR] Although, the methodology is developed with the aim of using the cluster wind resource representation for AWE production estimations, we consider obtaining and analysing the cluster representation the most important contribution of this paper. The last section is merely to show an application of the cluster representation and thereby presenting a complete story line: from data to application.

[AC] Lidar discussion is discarded. Onshore and multi-location analyses are expanded.

While you go into great detail explaining the process of normalizing, applying PCA and clustering your wind data, very little detail is given on the denormalization of the wind profiles so they can be used for power estimation. Did you use the cluster centroids i.e. the mean of all the profiles which is not an actual profile that occurs in your data set? Similarly one can argue that the profiles derived from the $mean \pm PC * std$ shown in Fig. 2 and 10 are not actual profiles in the data set. Which range and step size did you use to denormalize the wind speed profiles? Is

it possible to determine the wind speed range of each cluster or is this information lost due to normalization? If so it would be interesting to see this range in power curve plot. Validating your approach against actual simulated profiles by comparing energy or power curves would add to the credibility of this paper. This could include a comparison against non-normalized, clustered wind profiles or not PCA transformed profiles or standard log profiles with assumed Weibull distribution etc... Please justify why you go through all this process for 2D wind profiles if you just use the wind speed as an input into your AWES model.

[AR] Indeed they are not actual profiles, but averages. 100 bins are used between the cut-in and cut-out speed prescribed at 100 m. Basically, we only use the magnitude profiles because the performance model is not compatible with a profile that is veering. The methodology is however not developed specifically for coupling to this performance model. For different applications, the two-component profiles may be useful. We believe that the two-component analysis does provide useful insights in terms of profile veering which relate to stability.

[AC] We have included a more precise description of the wind profile shapes reflecting the centroids: "the cluster-mean wind profile shapes". Also, we introduced a more precise description of obtaining the cluster representation: "After obtaining the cluster-mean wind profile shapes, they are used for constructing the *cluster representation* of the wind resource. Each sample's vertical wind variation is approximated by de-normalising [scaling] the cluster-mean wind profile shape of the cluster to which it is assigned using the normalisation wind speed of the pre-processing." Furthermore, the power curves are complemented by the wind speed distributions.

PCA: Which PC and std do you use to derive the profiles column 3 and for of Fig. 2 and 10? Maybe a better representation would be to plot the mean, $mean \pm PC * std$ in one plot with a shaded area in between to highlight the range of possible speeds / velocities? It is my understanding that the justification for using PCA is that it

accelerates the clustering process. Is clustering the bottle neck of your analysis or is it the optimization? How does the end result vary comparing transformed and not transformed profiles?

[AR] For each PC a row is reserved, the std is an output of the PC analysis. In our case, the clustering is relatively cheap. The PC analysis however reveals some interesting features of the datasets. Clustering the data in the 'physical' space would probably yield similar results.

[AC] More precise panel titles are added.

Clustering: Please justfy in the text why you only use the cluster centroids for your analysis and how which uncertainties / inaccuracy this causes? How much does the profile shape within one cluster vary and is this variation reduces by normalizing the data? Clustering labels are not in order. How did you sort them or determine analogies between clusters of different data sets (see table 2)? Comparing clusters like this gives the impression that they have the same profile shape. Cluster centroids are the arithmetic mean of all the data points within this cluster. Therefore, adding or removing a data point changes the outcome of the entire process. How can you compare these clusters in table 2? If only 8 clusters are enough, then why not use existing stability classification (like table 1 in https://www.adv-sci-res.net/6/155/2011/asr-6-155-2011.pdf or https://www.wind-energ-sci.net/4/563/2019/wes-4-563-2019.pdf) based on Obukhov length or Richardson number which are widely accepted and a common meteorological classification?

[AR] We aimed in this work for a compact representation and therefore use normalisation, which inevitably comes at the cost of precision. Standard wind resource assessment relies on the use of a monotonic logarithmic profile which at best incorporates the effect of stability. We believe that our compact cluster mean profile methodology represents a significant advance on this approach particularly where the wind resource

extends well beyond any surface layer. One could look at variability of profiles in the cluster as a measure of uncertainty, but this is beyond the scope of this work and would cloud the central aim of the work. Indeed, we now do try and relate the properties of the surface layer profile to the stability conditions and use a standard stability classification (Table 1).

[AC] We have expanded the text on the labelling of the clusters. While discussing Figure 7, we state: "Note that the cluster algorithm produces arbitrary labels for each class. We have manually renumbered them such that the numbering is more or less aligned between onshore and offshore clusters." Furthermore, the original Table 2 is omitted. We added some text in the conclusions to acknowledge variability in the profile shape in each cluster. We added a table of fairly standard stability classes.

**2 Specific comments including technical corrections**

Title: I recommend a more specific title including information such as: ground-gen / pumping mode, mesoscale wind data, but understand that it is personal preference.

[AR] This would not really reflect what we see as our main contribution: the data-driven methodology.

**2.1 Abstract**

page 1

    line 1: Why not use AWES abbreviating for Airborne Wind Energy System as commonly used in the community and literature
    [AR] We prefer to stick to using solely AWE as a convention.
    line 6: Introduce abbreviation DOWA here.

[AR] We consider it as good practice to avoid acronyms in abstract.

line 10: Mention the derivation of power curves.

[AR] Added

line 10: Mention the location where you estimate AEP or add AEP analysis for all sites since it sounds like multiple AEP estimates are compared.

[AR] Added

line 11: Define or rewrite the sentence so that it is clear that you compare the amount of clusters necessary to estimate AEP and use concrete results rather than "within a few percent".

[AC] Replaced by "within three percent"

**2.2 Introduction**

page 1

line 13: AWE already defined in Abstract in line 1.

[AR] We consider it as good practice to reintroduce acronyms in the body of the paper.

line 14: Many different concepts exist with varying (anticipated) operating altitudes. Either remove subordinate clause "typically in a range up to 500 m" or explain / reference what this assumption is based on.

[AC] Rephrased to "above 150 m" and references added.

line 14: Add reference for stronger and more persistent winds.

[AR] Considered superfluous as this is described by the relationships in the next paragraph.

line 18: This is not the definition of surface layer. I would stay away from concrete number as the height of the surface layer varies a lot depending on atmospheric condition. Remove or replace with: in the order of tens of meters depending on atmospheric stability (http://glossary.ametsoc.org/wiki/Surface_boundary_layer)

[AC] Left out the 200 m

page 2

line 1: Low-level jets are a known phenomena and have been discovered a while ago. Rewrite: "Recent studies have identified numerous low-level jets"

[AC] Rephrased

line 4: Remove: "more"

[AR] Done

line 9: What kind of "performance calculation"?

[AC] replaced by "power production"

line 9: Add reference to COSMO-DE

[AR] Only listed because used in the reference. Since the indirect reference, we do not deem it necessary to include a separate reference to the dataset.

line 10 & 11: Add: "shape" after wind profile

[AR] Done

line 14: Add reference to MERRA-2

[AR] Only listed because used in the reference. Since the indirect reference, we do not deem it necessary to include a separate reference to the dataset.

line 14 & 16: Malz et. al b) referenced before a).

[AR] Modified

line 15: Sentence on the reduction of computational cost and choice of initialization seem unnecessary for this paper.

[AR] We find it useful as it emphasises that there is a demand for cheaper AEP calculations.

line 20: Add reference and expansion of WRF

[AC] Expanded WRF

line 26: Remove "such a historical dataset". Does not contribute to the understanding or quality of this paper. What makes this data set so special?

[AC] Rephrased

line 34: Add reference to DOWA, ERA5 and expansion of LiDAR (also, check WES house standards on capitalization of abbreviations: https://www. wind-energy-science.net/for_authors/manuscript_preparation.html. One feedback I got was: "WRF is a defined model name (the same applies to the ARW), but lidar is a general term and is therefore not capitalized according to our house standards."

[AC] Reference included in the data section

page 3

line 1: Is the DOWA data set comprised of met mast data? I thought it is derived from reanalysis models which had various measurements assimilated.

[AC] Rephrased

line 2: Sentence is a little confusing as it suggests that met mast data is somehow involved. Maybe chose a different naming convention e.g. onshore, offshore.

[AC] Rephrased

line 2: "The procedure" is confusing as the previous sentence describes comparing DOWA and LiDAR and not generating a set of representative wind speed profiles from clusters

[AC] Rephrased

line 5: More accurate than log profiles or uniform wind? Is it actually more accurate and did you validate the improvement?

[AC] Rephrased

line 7: Active voice

[AC] Rephrased

**2.3  Wind data sets**

page 3

line 10: Combine first 2 sentences.

[AC] Rephrased

line 16: Repetitive sentence: "Both long-term ..."

[AC] Rephrased

line 17: In what way "sparser" lidar? spatial, temporal?

[AC] Rephrased

line 21: Repetitive sentence: "...  coast in the North Sea".  Also shorten the sentence e.g. "The selected offshore location, met mast IJmuiden, is located 85 km off the Dutch coast."

[AC] Rephrased

line 21: Write out the wind direction as it doesn't safe lots of space but adds clarity

[AC] Rephrased

line 21: Why not use the Cabauw or IJmuiden met masts for validation as well? I thought that was one of the reasons why you chose this location.

[AC] Rephrased — stressed that we don't use met mast data (anymore)

**2.3.1 ERA5**

page 3

line 26: Define and add reference to ERA5 and ECMWF.
[AC] Reference added

line 27: I don't think the year in brackets as well as the information that more data from years back will be available soon is necessary.
[AC] Rephrased

line 31: Replace outputs with a different verb: produces, uses, calculates and rewrite sentence to be more concise
[AC] Rephrased

page 4

figure 1: More explanation in caption
[AC] Caption expanded

figure 1: How and why did you choose the sample location? Are these representative locations?
[AC] Explained in text

figure 1: Is it important that you use met mast locations as you don't you met mast data at all.
[AR] No, the reason for doing so is that they are well covered in literature.

line 1: Remove sentence: The long-term wind climate is not important for this study as no long-term predictions were made.
[AC] Removed

line 2: You use ERA5 to determine atmospheric stability. Does DOWA not provide the necessary data?

[AR] It does, however DOWA did not give satisfactory results for calculating the Richardsons number.

[AC] We clarify that we used ERA5 rather than DOWA as it gave more satisfactory stability values/

line 2: Stability is only used for cluster statistics in Fig. 9 and similar. Please expand this analysis and the relationship between clusters and stability.

[AR] Parallels have been drawn between the clusters and stability in Sect. 4.2, however, because we only feed in the normalised wind profile data, the relationship will be indirect.

[AC] We have added more analysis of relationship between stability and cluster mean profiles.

line 3: DOWA defined on page 2

[AC] Rephrased

2.3.2   Dutch offshore wind atlas

page 4

title : Capitalization different from previous DOWA

[AC] Capitalized

line 6: How downscaled? Is it not more the extraction of specific values from a data set?

[AR] Mesoscale model downscaling is a widely accepted way to increase the resolution of a coarser scale meteorological dataset. It is not just the extraction of values - the mesoscale model captures the physics of finer scale processes.

line 6: Add reference to model and / or move link to footnotes

 [AC] Reference added

line 10: Add reference

 [AR] Same ref as previous

line 11: Remove 2nd "than" and "alone"

 [AC] Removed

line 12: Define or paraphrase "ASCAT" and "mode-s ehs". Add reference

 [AC] Put between parentheses

line 13: Sentence about website unnecessary

 [AC] Removed

line 14: Remove "additionally" and conjugate "show" or add et. al

 [AC] Removed

line 15: LLJs are not anomalous

 [AR] It is defined as such in https://www-sciencedirect-com.tudelft.idm.oclc.org/science/article/pii/S0167610516307061 .

**2.3.3 lidar observations**

[AC] Subsection removed

page 5

 title : Check capitalization and abbreviation rules for lidar

 line 1: Rewrite. I don't know what you want to say with this sentence.

 line 4: ECN?

 line 5: Add link as reference and / or footnotes

line 6: Where is difference between "clock hour" and hourly?

line 7: Active voice

line 8: Only time you use "data set" instead of "dataset"

**2.4 Clustering procedure**

page 5

line 12: Mention that you do the same for MMC
[AR] The onshore analysis is part of Sect. 4 and therefore only introduced there.

line 13: How normalized? Add PCA abbreviation
[AR] Described in following subsection.

line 14: rewrite "for choosing", active voice e.g. the number of clusters is chosen based on the clustering performance...
[AC] Rephrased

**2.4.1 Preprocessing of the wind data**

page 5

line 16: Shorten sentence, mention use of all time steps, entirety sounds like more than it is.
[AC] Rephrased

line 20: Is that 90th percentile of each time step or of all data points of this altitude or of all data points at 100 m?
[AR] Each time step
[AC] Rephrased

line 20: Line break not necessary

[AC] Removed

line 21: Why did you choose this normalization and why normalize in general if you expect that it will not lead to good results for low winds?

[AR] Normalisation is used as we aim for a compact representation.

[AC] Rephrased: "Fewer outlying wind profile shapes result when the 90th percentile instead of, e.g., the maximum value is taken as normalisation value. The normalisation yields a more compact wind resource representation."

line 21: Active voice. Do you only expect eccentric profiles or did you actually observe them?

[AR] We observed them mostly in the lidar dataset due to outliers. For the modelled data it is less urgent to use a percentile instead of the max value.

line 24: How do you implement them if they are omitted? [AC] Rephrased

2.4.2   Principal component analysis of the wind profile shape dataset

page 5

line 27: Defined above

[AC] Rephrased

page 6

line 4: "for every wind direction" = omnidirectional

[AC] Rephrased

line 4: The difference to ...

[AC] Rephrased

line 5:"logarithmic profile representation of the wind environment" = logarithmic wind profile. Remove one "logarithmic profile". Introduce $z_0$ here. Why is $z_0 = 0.0002$?

[AC] Rephrased, added reference for roughness length.

line 6: "vertical" is understood, wind speed is always magnitude

[AC] Rephrased

line 6: Add variable name $u(z)$ and reference to equation

[AC] Moved to introduction, and introduced before stating the equation.

line 8: Add reference or explanation to Obukhov length

[AC] Explained in introduction.

line 13: Explain relationship of $L$ and atmospheric stability. Does this sentence mean that the average of multiple years of wind data fits an unstable log profile? Are you fitting wind speed or just parallel component?

[AR] Fitting to the magnitude profiles, not the parallel component profile.

[AC] Explained in introduction.

line 13: Add reference to "Theory". Add "mean wind profile...", remove "in the" & "direction".

[AR] Ekman theory is considered text-book knowledge and therefore we chose not to add a reference here.

[AC] Rephrased to "In accordance with Ekman theory, the mean wind profile shape veers .."

line 13: Add "(top-view in the bottom left panel)"

[AC] Referred to lower left panel in text.

line 14: Remove "of the wind profile shape"

[AC] Removed

line 22 & 23: Remove text in parentheses

[AC] Removed and rephrased

line 24: Add reference

[AC] Rephrased: "We consider retaining 90 % or more acceptable for our application."

line 28: Active voice

[AC] Rephrased

line 31: Isn't this a general feature of PC? Remove "Note that"?

[AC] Removed

page 7

figure 2: What do the number 1-4 in bottom boxes mean? Add normalized to vertical wind profile. Does second column show the mean or is it the orientation of PC axis with height? What does it mean to multiply PC with std? Which std do you multiply with? It might make sense to show the std of each PC in a table. Plots in column 2 are not on the same x-axis. Please explain how PC1 and 2 rotate over altitude.

[AR] Second column is not the mean: it illustrates the unit vector defining the direction of the PCs. Not completely clear what you mean with: "Please explain how PC1 and 2 rotate over altitude." Refrained from adding another table, the panel titles should be more self explaining now.

[AC] Added: "The wind profile shape numbers 1–4 refer to the markers in Fig. 4a." Changed x-axes of column 2.

2.4.3   Number of clusters

page 7

line 6: Either active voice or general statement in which case you should use "a" instead of "the"

[AC] Reordered/rephrased, reference shouldn't have been at the end of the sentence.

line 8: Remove parenthesis. Replace "all" with "each"

[AC] Rephrased

line 10: Add reference

[AR] Rephrased a little, such that is more generic and does not need a reference.

page 8

figure 3: Explain legend abbreviations and subset in caption. What is the MMIJ subset?

[AC] Changed legend

figure 4: why $*$ on x- & y-axis label? Are the markers the orange circles? Do the numbers 1-4 correspond to the clusters shown in Fig. 2? If so mention this in the caption.

[AC] Expanded caption. "The coordinate system represents the average PC profiles of the two reference locations, denoted by an asterisk." Also reference to Fig. 2's profiles explained.

figure 4: Maybe replace "wind profile shape" with wind velocity profile.

[AC] Shape is used consistently throughout paper, so also preferred here.

line 1: What is "itk" ? Number of clusters is usually the $k$ in k-means clustering

[AR] Typo, should be italic k.

[AC] Corrected

Interactive
comment

line 5: Delete if not shown. If you keep it in explain "over-fitting" in this context. Probably not happening because of (relatively long) temporal averaging.

[AC] Removed

line 7: How is the silhouette score calculated? Adding silhouette score and WCSS equation would make it easier to understand. Is a score of 1, 0 or -1 better?

[AR] The silhouette score calculation is quite complex and not considered crucial for understanding the assessment and therefore left out.

[AC] Added explanation on score values.

line 9: Remove repeated subject "mean silhouette score". How do you interpret the decreasing trend? Explain if you mention it, the fact that it is decreasing is obvious.

[AC] Rephrased

page 9

figure 5: Which data set do these results belong to: MMIJ, MMC, ML? Are these results the same for all data sets? Expand on the captions.

[AC] Added ".. for filtered offshore dataset"

figure 5 a): y-axis label: Isn't it the "sum of squared distances" ?

[AC] Changed to WCSS, is considered a more precise description.

figure 5 a): Add a vertical line at $k = 8$ to highlight your choice.

[AC] Added vertical line

figure 5 b): Explain non-linear grid lines in caption. Why no vertical grid lines? "...error over height for logarithmic profiles...(...and four different stability...)", "...three exemplary clusters". How is the error defined?

[AC] Expanded in text.

line 1: So it would be best to use only 2 clusters to represent the many different wind conditions even though the sum of square distance is way higher and 4 log-fits out perform it? What defines a good silhouette score? Wouldn't it be a fair comparison to also vary the number of log-fits over the x-axis? Why 4 log-fits?

[AR] Purely based on the silhouette score: yes, but overall: no. Score close to 1 is desirable. 5 log shapes are used now: 1 for each stability class. Is considered a reasonable comparison when considering 5 stability corrections - often the wind resource representation in AEP estimation is not corrected for stability at all.

[AC] Rephrased

line 2: Why do you show the different approaches to get best k and then use AEP without showing it here?

[AC] Rephrased

line 5: Shorten to"filtered wind profile data". Parentheses are repeating what has been said before, remove.

[AC] Rephrased

line 5: Remove: "Next, it is..."

[AC] Removed

line 8: Equation would make it easier to understand.

[AC] Equations added

line 11: More combined than individually, otherwise 2 lines.

[AC] Rephrased

line 16: How did you choose these $L$ values? What are the assumed ranges of $L$ associated with certain atmospheric stability?

[AC] Table added with stability classes, including reference.

figure 6: Why vertical grid line in center of plot? Showing the x-axis as percentage of total data would be better. Briefly explain what positive and negative values mean?

[AR] We feel it is justified to use "identifiers" on the x-axis, as they point to specific samples and thereby emphasize that the order of the samples is not random.

[AC] No specific reason for the middle grid line, therefore discarded.

line 5: Is this weighting intended? Why not linear interpolation in z?

[AR] We did not tailor the weighting, but it could be justified: the weight is higher around 100–200 m, in which the reel-out phase is mostly taking place.

line 6: It seems like this is not the only reason. Resolution is high up to 200 m.

[AR] Agreed, we had a closer look and found out that model deficiencies also contribute to this effect.

[AC] Added: ".. the PC1 and PC2 profiles show that most variance in the dataset is found at both ends of the height range. Due to the relatively high variance and fit model deficiencies, the fit error is also expected to be largest at these heights."

line 9: I assume that a low number of clusters is enough to capture variations within the hourly data set which is why I would recommend mentioning that this choice is specific for your temporal resolution.

[AR] For this type of analysis we are not interested in atmospheric phenomena with a time scale lower than an hourly one.

[AC] Added to start of Sect. 2 to address the latter: "An hourly temporal resolution of the datasets suffices for capturing the diurnal cycle of the wind profile. While smaller scale atmospheric phenomena might have an

(adverse) effect on the power production, these effects are typically super-imposed on a steady-state wind profile using separate models for assessing, e.g., the associated loss in power production (Fechner, 2016)."

line 14: Move the last sentence up to the section where you first introduce clusters.
[AC] Sentence removed

**2.5 Wind resource representation based on clusters**

**2.5.1 Interpretation of prevailing wind profile shapes at MMIJ**

page 11

line 10: Are these profiles the centroids of the clusters you calculated before?
[AC] Rephrased this sentence. Moreover, added "The resulting centroids reflect the cluster-mean wind profile shapes in the dataset, which follow from back-transforming the cluster centroids from the PC to physical space." to Sect. 3.3 to be more clear about how the centroids relate to the shapes.

line 13: Are these the Obukhov lengths? Which ranges do you associate with each stability condition?
[AC] Stability classes given in Table 1.

line 12: Replace "...moving between..."
[AC] Rephrased

line 13: What are the number in parentheses? Add variable.
[AC] Obukhov lengths found for the mean-cluster shapes are now presented in Figure 9.

line 13: Move line break as veering clearly refers to the previous paragraph.
[AC] Text is rearranged

line 21: Fig 7 is mentioned after Fig 8. Active voice

[AC] Figure ordering corrected

line 22: Also active voice: "Examining the five PC coordinates in table 1 ...."

[AC] Rephrased

line 26: How can that be deduced? Define filter e.g. "low-speed filtered..." as readers might have forgotten or skipped the previous sections.

[AC] Discussion on filtered vs full dataset removed as requested by other referee.

line 28: Define "calm wind"

[AR] See previous bullet

line 29: These 2 sentences are more general and introduce the data set. Move up before previous sentences.

[AR] See previous bullet

page 12

equation 2: Why are you using different $\Delta z$ for $u$. $v$, $\theta$ within this equation? Are $u$. $v$, $\theta$ on different altitudes?

[AR] No, on the same altitudes.

[AC] Equation moved to introduction and reformulated. Now, we only use a single $\Delta z$.

line 6: Why not rewrite Eq. 2 to include absolute temperature instead? Explain how it is used. I guess instead of virtual potential temperature.

[AR] We changed the stability metric. Currently, we now determine $Ri_B$ between 10–31 m.

[AC] Added to Sect. 4.1: "Here, we derive the stability class distributions using the bulk Richardson number, $Ri_B$, converted to the Obukhov length, $L$,

using Eqs. **??** and **??**. The data from either ERA5 or DOWA could be used to derive $Ri_B$, however, we found that using the data from the two lowest ERA5 model levels, i.e., $\sim$10–31 m yields the most realistic values. We use the arithmetic mean of the model level heights for $z$ in order to convert $Ri_B$ to $L$."

line 6: Isn't humidity high close to the shore / offshore and would the effect not be considerable?

[AR] The newly used expression used for $Ri_B$ includes humidity.

line 8: Add "Positive $Ri_B$ values"

[AC] Added

table 1: Do these values relate to Figure 2?

[AC] Added: "The centroids are depicted in Fig. 7a at their PC1, PC2-coordinates with the numbered markers."

page 13

table 2:The cluster centroid shape will change based on the underlying data, i.e. filtered. It seems that you assume that clusters between data sets are same, e.g. cluster 1 is same through out all data sets. Did you sort or determine similarity between cluster centroids of different data sets? How similar are PC transformed and normalized wind data? Can you quantify how similar / different they are?

[AR] The clusters are not the same, only the number assigned to them is. Labelling of the clusters is done manually. The implications of PC transformation is explained at length in Sect. 3.2, but I don't think this is what you're asking about. If I'm correct, you are asking about the sensitivity of the clusters to the data pre-processing? Such a sensitivity analysis could be done, however, we feel this is out of the scope of this paper. At an early stage, we found that the PC transformation does not have a large effect on

the resulting clusters. The PCA however helps us to understand the data structure and therefore is a helpful contribution to to the paper.

[AC] Added to Sect. 4.3: " Note that the cluster algorithm produces arbitrary labels for each class. We have manually renumbered them such that the numbering is more or less aligned between onshore and offshore clusters. This allows us to draw parallels between them .."

figure 7: Nice visualization. Why $*$ on x and y label?

[AR] The location-average of the PCs is used for the coordinate system, enabling a direct comparison between the two subfigures.

[AC] Added: The coordinate system represents the average PC profiles of the two reference locations, denoted by an asterisk and shown in Figs. 2 and 11.

page 14

figure 8: Is this the filtered or unfiltered data set? Do all x-axis use the same range, because different label? Explain more in caption, i.e. why $\hat{v}$ over variable, where is the origin of the hodograph / increasing $z$ direction? Better spread out all the graphs so that they appear after mentioned in text. Do you need all these graphs here? Maybe move some to the appendix.

[AR] Filtered. Yes, same x-axis. Yes, we discuss all shapes in the body of the paper, so therefore it would also be best to present them here.

[AC] Caption expanded, added: "The eight cluster-mean wind profile shapes of the offshore clusters (MMIJ-1–8). Each shape is depicted by the normalised wind speed components with height (first and third rows) with the corresponding hodograph below (second and fourth rows). Logarithmic profile fits are plotted alongside the shapes. In the hodographs, the lowest points are connected to the origins with dotted lines and the highest points are the loose ends. All plots share the same x-axis."

page 15

figure 9: Nice way of comparing the data set. Might be a bit overwhelming at first though. Is this the entire data set or filtered? Where are these $Ri$ bins coming from? Add reference. What do you mean by "Bins have the same overall frequency" ? Wouldn't that mean that all the bars have same height? Do you mean bin width? Use the same comprehensible bin width for all subfigures (especially wind speed and $Ri$ seem arbitrarily chosen) in figures 9 and 12. Maybe add an "offshore" and "onshore" to data set labels for clarity. A better and easier understandable metric would be atmospheric stability (i.e stable, unstable) instead of using $Ri$ number.

[AR] Filtered data set. Bin sizes are the same over all clusters, but not within one cluster.

[AC] Rephrased and expanded caption, on/offshore added. Stability distribution changed corresponding to earlier introduced classes.

page 16

line 1: Change figure order or reference which figure you mean after 3 pages of figures. "... over the 10 year timeframe".

[AC] Reference added

line 2: Why is this a prerequisite? You have temporal variations on all time scales. But I guess this way you make sure that your data is not based on outliers.

[AC] Text expanded: "The upper panel shows that the inter-annual variability is limited, which asserts that the results can safely be generalised to the lifetime of a wind energy system (âĹij20 years)."

line 3: Remove sentence and line break. Maybe change order. Why first talk about 9 a) and then all the other figures separately ?

[AC] New subsection introduced specifically for the interpretation of the results.

line 5: Rewrite: "... not so frequently strong". According to figure 9, cluster 1 has an almost even occurrence through all wind speed ranges.
[AC] Rephrased

line 5: "are more frequent".
[AC] Rephrased

line 7: Add reference to figure which shows cluster shape.
[AC] Reference added

line 9: "well mixed". How about shear?
[AC] Corrected

line 26: Remove line break as both paragraphs are about cluster 6,7.
[AC] Removed

2.5.2   Comparison with an onshore location

page 17

line 5: Vertical is understood. Better write: "The mean normalized wind speed profile...".
[AR] Earlier in the paper we define the shape as being a normalised profile. We feel that this definition should be clear at this stage of the paper.
[AC] "Vertical" removed

line 8: Same as line 5
[AR] See above

line 11: Removing "of the two locations ... " makes the sentence easier to understand.

[AC] Rephrased

line 19: Add the variable to values in parentheses

[AC] Results now presented in Figure.

line 20: So is there no cluster that corresponds to stable stratification without LLJ? If so wouldn't that be unusual?

[AC] Stable stratification is recorded for MMC-3/4. MMC4 cluster-mean shape only shows a weak jet shape.

line 22: "however" and "just" seem like filler words. You could shorten the sentence

[AC] Rephrased

line 24: Active voice.

[AC] Rephrased

line 26: Sentence hard to read. Shorten i.e. "This affect is caused by the lower heat capacity of the land surface which promotes a more immediate heat transfer to or from the atmosphere."

[AC] Rephrased as suggested

line 30: What is the point of comparing diurnal to seasonal cycles? They are caused by entirely different effects and play out over vastly different timescales.

[AC] Rephrased: "The patterns in the times of occurrences indicate a pronounced diurnal cycle in atmospheric stability for the onshore location, whereas for the offshore location the seasonal cycle is more pronounced."

line 32: What does overall bin frequency mean and why does it have to be equal? It sounds like you are varying your classification to get a certain result rather than try to characterize actual physical effects.

[AR] Not applicable anymore for stability distributions. Bin size is the same over all clusters, not within one cluster.

[AC] Rephrased: "Note that the wind speed bin limits are chosen such that the frequency over all clusters for each bin is roughly the same, yielding different bin widths for the two reference locations."

page 18

figure 10,11,12: More explanation for readers who just skim through the text or just look at figures, i.e. onshore, normalised wind speed, numbers and explain PC*std. Consider moving some figures to the appendix.

[AC] Captions expanded

**2.5.3 Validation with LiDAR observations**

page 21

line 1: This paragraph seems rather unnecessary. Very short, no mayor inside, only that results are similar.

[AR] Agreed

[AC] Subsection removed

line 2: replace "investigate" with "show"..

**2.5.4 Spatial distribution of wind profile shapes**

page 21

line 10: How did you chose these sample locations in Fig.1? Are 45 grid points the entire domain? Does your selection affect / bias the results? Line break not needed.

[AC] Text expanded: "The multi-location dataset (filtered to exclude low wind samples) includes wind data from 45 DOWA grid points that are selected such that onshore, coastal, and offshore locations are equally represented. For each location type, 15 grid points are chosen (pseudo-randomly) to yield a good coverage of the full DOWA domain (50778 grid points in total)."

line 12: Explain cluster mapping. Why only 8 clusters again? reference previous chapter.
[AC] Added: "Each sample of every grid point in the DOWA domain is assigned to the cluster with the closest centroid."

line 13: How can you apply the same mapping to a new data set?
[AR] With mapping, we ment the assignment of samples to a cluster, see upper bullet.
[AC] "Mapping" is a bit confusing, therefore, rephrased.

line 16: Remove "be" and shorten / rewrite sentence.
[AC] Rephrased

line 18: Remove sentence "Since cluster 1..."
[AC] Rephrased

page 22

table 3: Is this a necessary table? offshore / onshore is not a sufficient classification of wind. How did you match clusters from different data sets? Are they the exact same clusters and if not how similar are they?
[AR] We expanded the analysis in this subsection and feel that it is now justified to leave the table in. We agree that it is only a very crude classification scheme, however, e.g. IEC standards also differentiate between on- and offshore classes, so we feel that justifies using these classes. Matching is done manually. In practice, clusters are never the same for different datasets.

[AC] Added: "Every multi-location cluster is manually linked to the single location clusters based on resemblance of their cluster-mean wind profile shapes, see Table 3."

page 23

figure 13: Is this a necessary figure? Dashed lines missing in legend. Is this based on hourly average lidar data? Do these numbers in circles have any meaning?

[AC] Figure removed

page 24

figure 14: Rewrite: many "the" in caption. Maybe a bit more explanation, i.e. which locations are represented.

[AC] Caption expanded

page 25

figure 15: Add details to captions. Add ML abbreviation. Add reference to fig. 14 for info on clusters. How did you "map" to a new data set? Meaning of numbers? Use consistent frequency ranges for comparison or justify why you did not.

[AR] Mapping is explained in text.

[AC] Caption expanded. Text added: "Note that the colour scale is different for each map so that spatial patterns are easier to observe."

figure 15: This clear division between on- and off-shore profiles would justify separating the analysis. Wouldn't this lead to better more detailed results and insights? Please validate and quantify how much information you lose by clustering everything this way rather than off- and onshore individually.

[AR] The aim with using the larger combined onshore and offshore dataset was to highlight how the prevailing profiles do vary depending on the terrain. We have tried to show this by relating the cluster profiles so obtained with those of the individual onshore and offshore analyses. For a detailed performance assessment, indeed we would suggest a more tailored (offshore or onshore) clustering approach which is why we used the conditions specifically at Cabauw for illustrating the AWE AEP assessment.

[AC] We have made some changes in the text at the beginning of Section 4.4 to try and clarify the rationale for the large area clustering analysis.

2.6 Fast AWE production estimation based on historical wind data

page 26

title : What is "fast" about this analysis?

[AR] The number of optimisations needed is reduced substantially w.r.t. 'brute-force' calculations.

line 1: Rather "estimating" than "calculating"

[AC] Replaced

line 2: You went through all the process of explaining the clustering process using MMIJ data set than introduce MMC, lidar and ML to now only use MMC? Why not focus on MMC entirely or apply your power estimation to all data sets?

[AC] Expanded in text: "An advantage of AWE systems over tower-based wind turbines is that they have access to winds higher up. This advantage is limited when low-shear wind profiles are frequent at the installation site, as is the case offshore, but this is not usual for onshore locations. Employing an AWE system at an onshore location thus requires a more variable operational approach. For this reason, we demonstrate the AEP estimation for the

met mast Cabauw location using the eight clusters from the single location analysis (Sect. 4.3)."

2.6.1 Determining power curves for AWE systems operated in pumping mode

page 26

line 10: "... differs between ...". Add reference to flexible-kite sentence.
[AC] Reference added

line 15: Rewrite sentence: "...is ended ... is depowered... is steered ....". Remove: "the" in front of zenith. Add: "... to the starting position of the traction phase"
[AC] Rephrased to active voice and added suggestions.

line 17: Check capitalization rules for abbreviations.
[AC] Uncapitalized

line 18: Rewrite sentence: "... moves the kite along an idealised flight path conform a series..." ?
[AC] Rephrased: "The motion of the kite is approximated by moving it along the idealised flight path according to the computed steady-state kite speed."

line 19: Doesn't the limitation to lightweight membrane kites mean that the approach is only applicable to soft kites and not "any kind of pumping AWE system" (line 12)?
[AC] Rephrased: "The specific operational approach differs between concepts and may require different performance models for calculating the generated power. We evaluate a flexible-kite system using the quasi-steady model (QSM) .."

line 24: I agree that it is justified to use only magnitude wind speed profiles, but why go through all the process of clustering 2D profiles in the first place?

[AR] The methodology is not specifically developed to be coupled to this performance model. Other applications might require the two-component profiles.

line 26: Explain in more detail how do you derive the power curve and how you scale the normalized profiles.

[AC] Expanded: "The power curves required for the AEP estimation relate the mean cycle power to the scaling parameter used for de-normalising the cluster-mean wind profile shapes of MMC-1–8. Given the profile shape, this scaling parameter can be prescribed as a wind speed at any height. We use the wind speed at 100 m. By stepping through a range of wind speeds between cut-in and cut-out, a power curve is constructed for each of the clusters. At each step, the profile shape is scaled using the respective wind speed to yield the absolute wind profile. An optimisation is then performed using this wind profile as input."

line 28: Active voice. Mention that aero coefficients are assumed to be constant.

[AC] Expanded: "The QSM uses constant values for the lift and drag coefficients of the powered and de-powered kite. In reality, the coefficients vary and representative values of the leading edge inflatable V3 kite are selected based on the experiment of Oehler and Schmehl (2019)."

line 31: What means "sufficiently high"? How is the tether reeling speed during reel-in and -out? Is it a constraint or output of the optimization?

[AR] Speed follows from the steady state calculation, given the tether force.

[AC] Rephrased: "The values of the cycle settings are chosen such that they yield maximum mean cycle power. The reel-in tether force should allow a fast retraction of the kite, while limiting the energy consumption. During the transition phase, the reeling speed is kept zero unless tether force limits

are exceeded. During reel-out, the tether force should yield a high energy production, while letting the reel-out phase comprise most of the cycle duration."

line 31: Rewrite to include tether force constraint and that it corresponds to setting a fixed max tether diameter.

[AR] I'm not considering any design variations here. Tether diameter is constant as given in Table 4.

page 27

table 4: More descriptive caption.

[AC] Caption expanded

line 1: What are your control variables? Table 5 contains constraints that you are keeping constant I assume. Maybe write out the optimization formulation?

[AC] Table 5 lists the optimisation variables and their limits. Note that we are not dealing with an optimal control problem here. We are maximising the mean cycle power by varying this confined set of variables. The stated algorithm implementation uses these limits as input. On the background the limits are converted to constraints, however, as a user of the algorithm you don't have to deal with this. Therefore, we don't think this information is needed for either the user of the reader of this paper.

line 2 & 3: What is the message of these sentences? Optimizations basically always have active constraints. Why do you have to lower the tether force? What if an increase in elevation angle leads to an increase in wind speed and therefore force?

[AC] Expanded text: "For high wind speeds, the system runs into its maximum tether force and reeling speed limits. Increasing the elevation angle of the reel-out path generally alleviates the tether force and expands the wind speed range that allows safe operations."

line 3: Define effective pumping length.

[AC] Expanded text: ".. the effective pumping length of the trajectory is the difference between the minimum and maximum tether length during reel-out and is included as a cycle setting."

line 8: Rewrite so that you describe where cut-in and cut-out comes from in this sentence or same paragraph.

[AR] The procedure for determining the cut-in and -out conditions are quite elaborate and therefore a separate paragraph is reserved for that.

page 28

table 5: What are these constraints based on? Is actual tether length $l_{min} + l_{pumping}$ ? Seems like a list of constraints. What are all the constraints? Are these realistic values (add reference)?

[AR] Yes it is. See reply to [page] 27/[line] 1.

[AC] Added to caption: "The limits are chosen by judgment of the authors."

line 3: How do you define "steady flight states"? What are the states? Active voice.

[AR] Flight states without acceleration - these follow from the QSM. Recall earlier sentence: "The motion of the kite is approximated by moving it along the idealised flight path according to the computed steady-state kite speed."

line 4: What about reducing lift / flight speed to achieve one figure-of-eight? To which ground station reeling speeds do your constraints correspond? Explain why you chose this constraint.

[AR] Lift is reduced indirectly, we don't allow $C_{L,\text{powered}}$ to change. In practice however this would be feasible. Flight speeds follow from, amongst others, kite position and $C_L$. The ground station used is not representing a real one.

[AC] - Added: "Increasing the elevation angle of the reel-out path generally indirectly de-powers the kite and alleviates the tether force. Controlling the

elevation angle can thereby expand the wind speed range that allows safe operations. Although not considered here, the kite could also be de-powered directly by controlling $C_{L,\text{powered}}$."

- Added to caption of table 5.: "The limits are chosen by judgment of the authors."

line 6: Explain the developed module and assumptions etc.

[AR] We try to refrain from going into to many technicalities as it will be to distracting here.

[AC] Expanded text "However, this motion can also be approximated as a transition through steady flight states, yielding an approximate duration of the figure-of-eight manoeuvre. Dividing the total duration of the reel-out phase by the average duration of a figure-of-eight manoeuvre yields the number of cross-wind manoeuvres flown."

line 9: This sentence explains cut-out limit again, same as line 4.

[AC] Removed last sentence

line 11: Rewrite sentence. "The corresponding cut-in and cut-out wind profiles are shown in figure 17" or so. How did you scale the normalized wind speeds?

[AC] Rewritten: Scaling each wind profile shape such that the wind speed at 100 m equals the previously determined cut-in and cut-out wind speeds yields the respective absolute wind profiles, shown in Fig. 18.

line 12: Is the critical height of 80 m related to the minimal tether length, size of device or other parameters?

[AC] Expanded: "The cut-in profiles have the same wind speed at roughly 80 m, which is the kite height at the start of the reel-out phase for the minimum elevation angle employed at low winds. This indicates that, for every wind profile, the cut-in criterion is critical at the start of the reel-out phase rather than at the end. The cut-out profiles exhibit roughly the same wind speed

at 300 m, which is the kite height at the end of the reel-out phase for the maximum elevation angle and tether length employed at high winds.

line 19 & figure 24: If wind speeds at 80 and 300m are a sensible choice why do you use $v_{100}$ ?

[AC] Rephrased to clarify the point we want to make: "The cut-out wind conditions for an AWE system are ambiguous when defined by wind speeds at a certain height without defining the profile shape. However, since the cut-out profiles all intersect at roughly 300 m, characterising the cut-out wind speed at this height yields a reasonably precise definition for all profile shapes. Similarly, the cut-in wind speed is well defined at 80 m."

line 25: add "... power curves..."

[AC] Rephrased: "Note that plotting the mean cycle power against the wind speed at 300 m would yield curves that end at roughly the same wind speed."

line 26: Check comma placement

[AC] Rephrased, see above

page 29

figure 17: You sometimes use left and right and other times a and b for sub-figures, choose one. More descriptive caption. How did you scale the profiles?

[AR] We use the letter sub-labels only when we need to explicitly refer to subfigures.

[AC] Scaling better explained

figure 18: Change to "Height" instead of "z" as you always used height before. Add black dashed lines in legend. Why disconnected lines at end of traction phase? Why did you choose this strange $v_{100}$ values? Why not $v_{80}$ instead? Remove: "traction" and "constant" from captions.

[AR] We use x, y, z if the plots comprises multiple spatial dimenstions. Jump in the lines is an artifact of the QSM. We use $v_{100}$ for the power curve, so also here.

[AC] Removed suggested words. Added to caption: "The wind speeds for which the trajectories are depicted highlight changes in the operational approach."

line 1: It would be interesting to see reeling speeds, tether force and other variables during one production cycle.

[AR] We chose to leave this out, since we already have a lot of figures as it is and we value the existing figures more than the suggested figure.

line 4: Why is a profile with LLJ the last to reach cut-out speeds? I would have expected it to cut-out earlier. Is it because of the height of the LLJ?

[AR] Because of its shape and because the cut-out speed is prescribed at 100 m. If prescribed at 300 m, all shapes would roughly reach cut-out at the same time.

2.6.2   Estimating the Annual Energy Production

page 30

figure 19: Seems like the profile shape has almost no impact especially at lower wind speeds where the power ramps up. Remove: "... that are ...". What is the actually wind speed range of each non-normalized cluster? How come the power curve does not plateau and bend down before cut-out?

[AR] What do you mean with non-normalized cluster? The curves do show such a trend.

[AC] Removed suggested. Wind speed distributions added for each cluster.

line 4: $f$ for frequency of occurrence rather than $p$ for probability as I assume it is based on the data you used and not a model like Weibull. It would be great if you could show the distribution of wind speed frequency. Remove apostrophe: "... is the systems power curve...."

[AC] Equation 9 expanded with numerical approximation. Wind speed distributions added to figure 20.

line 7: No line break needed.

[AC] Removed

line 8: How constructed? Equation?

[AC] Expanded in text: "The probability of each cluster is characterised using the normalisation wind speed of the pre-processing. The equivalent speed at 100 m height is calculated to determine the frequency in the wind speed bin, using:"

page 31

figure 20: How about MMMIJ? How does the AEP and power curve compare to log profiles with Weibull distribution?

[AR] We consider such comparison out of the scope of this paper. Remember that the aim of this section is illustrating the AWE AEP assessment.

line 3: Did you use 50 calculations to get the power curves of 8 clusters, i.e. more than 6 wind speeds per cluster?

[AR] No, 50 per curve.

[AC] Clarified in text.

2.7 Conclusions

page 31

line 7: What is "fast" about the calculation? Do you mean simplified? Rewrite e.g.: "... used to estimate AEP for a simplified pumping-mode AWES ..."

[AR] The number of optimisations needed is reduced substantially w.r.t. 'brute-force' calculations: two orders of magnitude faster.

line 11: Shorten to: "...simple logarithmic profiles...". Would be good to compare power curve and AEP against these log profiles.

[AC] Rephrased the conclusions

line 12: For hourly average profiles. What could be the impact of higher resolution data?

[AR] Such data would capture smaller than wind profile scale phenemona in which we are not interested. I would expect these to be filtered out by the PC analysis.

line 13 & 14: Shorten: Both locations show similar results.

[AC] Rephrased, drawing PCA conclusions for both locations at the same time.

line 14: Which samples do you refer to, all, MMC, MMIJ?

[AC] Rewritten: "The data points for the onshore location are more spread out, indicating a larger variety of wind profile shapes."

line 21: I am not convinced by this conclusion. Why does profile shape similarity proof that clustering is able to differentiate between atmospheric conditions? Also which conditions? If only stable and unstable two clusters might be enough.

[AC] Removed

line 21: Is your process able to determine atmospheric stability (with a certain confidence) solely based on wind profile shape? If so that would be a great addition to your analysis.

[AR] Conclusion was phrased bluntly. There's some sort of relation, but it won't be very strong or useable as suggested.

page 32

line 3: Which wind resource presentation?

[AC] Added "onshore"

line 4: How did you get a distribution from profiles? Would be interesting to see which cluster/ time of year or day contributes how much to AEP.

[AR] Question not completely clear. However the constructing the wind speed distribution is now better explained in the previous section. We don't think the specifics about that are needed here in the conclusions.

line 7: How do 25 optimizations relate to 4 clusters or wind speeds?

[AR] 25 per cluster, so 4 x 25

[AC] Expanded text: "25 optimisations for constructing the power curve of a single cluster"

line 11: How high is the error in comparison to single location clustering?

[AR] Looking at Figure 21, the ML-line at 28 clusters has a similar error as the MMC-line for 14. So you would roughly need twice the clusters to get a similar accuracy.

line 12: Add line break in front of "In the future..."

[AR] Currently this would leave a single sentence paragraph and therefore we choose not to do so.

---

## Author Comment (AC2) · 20 Apr 2020

Thank you for the comprehensive comments. We feel that they were very helpful for increasing the quality of the paper to the current level. Your comments, together with those of referee #1, led to a thorough revision of the paper. The most important changes to the paper include:

1. Including information on orography

2. Discarding the lidar discussion

3. Using one stability metric: the Obukhov length, and corresponding classification for identifying stability trends within the clusters

4. Fitting logarithmic profiles only to the lower part, i.e. <200 m, of the mean profile shapes

We respond to the referee comments by including our answers below the original comments. Our answers are preceded by one or both of the following labels:

AR = author's response

AC = author's changes in manuscript

**1  General comments**

Paper "Clustering wind profile shapes to estimate airborne wind energy production" describes statistical analysis of wind profiles in order to identify typical shapes of wind profiles that are later used to optimize Airborne Wind Energy systems. The paper is interesting, has scientific novelty and describes some interesting, meaningful and useful results. However, I would suggest that the paper would benefit from some careful editing. The overall quality of English is good, but the presentation of information could use some improvement. I had to apply some effort to follow the argument set out by the authors. Some important procedures (normalization of the wind profiles, fitting of the Obukhov length) are not explained sufficiently, while other aspects of the work that do not lead to any significant conclusions are explained at length.

[AR] Thank you for your comments. We have thoroughly revised the paper using your feedback and that of referee #1 (which had quite some comments). The most important changes include:

1. Including information on orography

2. Discarding the lidar discussion

3. Using one stability metric: the obukhov length, and using classes in terms of the Obukhov length

4. Doing the log fit only for the lower 200 m

**2 Specific comments**

- P2L6: "Computationally expensive, brute force calculations". How expensive are the calculations? Can they be done in a few hours on a desktop computer, or is HPC required? An approximate estimation could be added here for the general reader.

  [AR] For the study of Malz, three months of three-hourly MERRA-2 reanalysis data was used which originally took ten days. We decided not to put any concrete numbers here as it is very much dependent on the machine.

- P3L10-L25: Here the word "reanalysis" is used to refer both to ERA5 and DOWA. Although formally DOWA fits the definition of reanalysis, in practice, at least in my experience, the word "reanalysis" is used exclusively for global reanalysis, such as ERA5, ERA-Interim, MERRA, etc. I would suggest using term "modelled datasets" or something similar, to refer to ERA5 and DOWA at the same time, to avoid confusing the reader.

  [AC] Changed "reanalyis" in "modelled" as suggested.

- P5L6: Hourly averages are calculated for the LiDAR data. Why? For the rest of the paper modelling results with hourly resolution are used, which are usually interpreted as the "instantaneous" wind speed, of course taking into account the

fact that the mesoscale model cannot represent the turbulent fluctuations. Are the hourly averages comparable with model data?

[AR] In "Low-level jets over the North Sea based on ERA5 and observations: together they do better", P.C. Kalverla et al., various averaging methods are compared and it is shown that hourly averages align best with the instantaneous reanalysis data. However we discarded the lidar discussion completely since it was not adding much to the paper.

[AC] Discarded lidar discussion.

- P6L9: In the paper vertical wind speed profiles are fitted using logarithmic profile that has been corrected for stability. From the context I understand that Obukhov length L is the parameter that is being changed during the fitting procedure, but a more precise description of the methodology how the Obukhov length is acquired would be beneficial. If I understand correctly, the same procedure is applied for fitting the instantaneous and long-term mean profiles in Fig. 2. I would like to see some arguments why long-term means can be treated in the way authors do it here. I refer the authors to the paper "Kelly, Mark, and Sven-Erik Gryning. "Longterm mean wind profiles based on similarity theory." Boundary-layer meteorology 136.3 (2010): 377-390." for a discussion about long-term stability correction of wind profiles.

[AR] We are effectively fitting a mean value of the stability function (not using a mean value of L) which is the correct way to make the mean long term profile. The Kelly paper makes the point that it is not appropriate to calculate a mean L (based on flux measurements, etc) and then produce a mean profile.

[AC] Added: ".. following the approach recommended by Kelly et al. From this, a mean value of the Obukhov length $L$ can be inferred." to second paragraph of Sect 3.2.

- P5L18: Do I understand correctly that the parallel and perpendicular components

are calculated for a different reference wind direction in each sample, namely, in each sample the wind velocity at 100 m will have only the parallel component? If so, please expand the description of this procedure in the text.

[AR] Correct

[AC] Added: "As a result, the perpendicular wind speed profiles are zero at 100 m."

- P5L19-20 "each sample is normalized by the 90th percentile of its wind speeds at each height". This sentence is highly confusing, because one interpretation could be that at each height the distribution of wind speeds is constructed, and each height has its own normalization speed calculated as the 90th percentile of all wind speeds in given height. I suggest expanding the description of normalization to remove ambiguity.

[AC] Rephrased to: ".. the 90th percentile of the sample's wind velocity magnitudes is used to normalise the wind speed components."

- I think that Figure 2 is not successful in conveying the important information in an effective manner. The horizontal axis for PC1 and PC2 are not the same. The hodograph is small and the markings (dotted line vs uninterrupted line) are not explained. However, my biggest problem is the fact that the x-axis is chosen in such a way that the plots for parallel component and magnitude of the wind, which arguable are the most important features of the profile, are very small. In order to understand the results and conclusions I had to look at profiles that had the width of less than a centimeter (when printed out). I would like the most important features of profile to be plotted large and easy to understand. I would recommend rearranging Figure 2, dividing it into separate figures. To keep paper at a reasonable length, I suggest editing and shortening further sections of the manuscript. These issues continue in further Figures, such as Fig. 8. etc. I

recommend focusing on the important parts of the graphs, making them large and omitting less important information for brevity.

[AR] We have experimented a great deal with how to present these figures. We agree that they are not the best for depicting details in the profiles. However, they allow our results to be shown in a consistent and compact manner, while still showing the overall trends of the profiles. Our conclusions are mostly about the latter and therefore we feel that it is justified to keep the current lay-out of the figures.

[AC] Changed the horizontal axes for PC1 and PC2 to be the same. Hodograph dotted line explained in caption.

- Section 3.3. requires some editing. Many complicated metrics are introduced, but only briefly, which does not allow the reader to follow the discussion easily. For instance, the fact that higher values of silhouette score mean better similarity within cluster, should be mentioned when the metric is first introduced (P8L10), or at the start of the discussion paragraph (e.g. P10L12)

[AC] Added: "The dimensionless score ranges from -1 to 1: a negative value suggests that the sample is assigned to the wrong cluster, a value around zero indicates that the sample lies between two clusters, and a high value indicates that the sample is assigned to a distinct cluster". Also added expressions for the fit error metrics.

- P8L1: Why is "itk" used as a parameter name for the number of clusters? It is used twice and then never used again, for instance, in Fig. 5.a. Maybe it is better not to introduce such variable at all.

[AR] Sorry for the typo. It should have been an italic k, where k refers to the number of clusters (k-means).

[AC] Corrected
- Fig.5.(a). If the "cluster mag" and "cluster 2d" lines are identical, then maybe one of them can be omitted?

  [AR] They are not identical, but similar. We think it is helpful to show how the two error metrics for the same representation compare, as both metrics are used for drawing conclusions: 'mag' is used for comparing to the log fit and '2c' is equivalent to WCSS.

- Figure 9 and Figure 12: (a) panel could be omitted without loss of important information – all years are the same. Please make the rest of the plots larger – again I was forced to concentrate on very small portions of plots to arrive at conclusions. Maybe Figures 9 and 12 can be combined to save space for legends. I would also suggest using the same Richardson number bins for both figures to make interpretation easier.

  [AR] In contrast to the other panels, the absolute frequency is on the y-axis and serves to show which part of the total dataset is represented by each of the clusters. This is deemed necessary since the table listing the cluster frequencies is discarded (see next comment item).

  [AC] To make interpretation of the stability easier, the samples have been binned using stability classes in terms of the Obukhov length.

- The discussion of filtered vs full dataset, e.g. Table 2, could be omitted to shorten the paper.

  [AR] Agreed

  [AC] Table discarded

- Two different measures of stability are used in the paper – L (Obukhov length) and bulk Richardson number. For clusters (2) and (3) in Fig.8. L values that correspond to neutral stability conditions are reported (P11L13). From Fig.9. and the

discussion (P16L11-L17) they are associated with stable conditions. How do authors explain this discrepancy? Additionally, I would like to point out that clusters (2) and (3) are associated with higher windspeeds, and higher windspeeds typically are associated with higher frequency of neutral conditions, see for instance, Holtslag et al. 2014.

[AR] Due to these discrepancies between the fitted L to cluster-mean profiles and the recorded samples stabilities, we decided to change the fitting procedure such that the log profiles are fitted only to the wind speeds in the lower 200 m. This led to both approaches yielding more consistent results. Also the log profile is used in a more valid manner, i.e., the layer < 200 m approximates the surface layer.

[AC] Changed the fitting procedure. Converted the bulk Richardson number to L, allowing direct comparison.

- P11L13: Fitted Obukhov lengths are only reported for clusters (1)- (3). I understand why they should not be reported for cases where the fit cannot reproduce the shape of the profile, e.g. cluster (7). But what are the fitted Obukhov lengths for other clusters, such as (5) and (8)?

[AC] Added figures 9 and 11, showing how the Obukhov lengths found compare. We think stating the exact values of L is not necessary, as the fits serve most importantly to show to what extent the cluster-mean wind profile shapes deviate from non-adiabatic logarithmic profiles.

- The authors claim that clusters 4-7 indicate potential low-level jets. I do not disagree, but in my opinion, the evidence presented is not strong enough and is somewhat circumstantial. Mostly, because the amplitude of wind speed maxima in Figures 8 and 11 is quite small (except for cluster (7) where I agree that the jet-like profile is quite distinct). Sometimes in literature it is required that wind speed in jets is at least 3 m/s higher than the surrounding flow. Due to the normalization procedure it is hard to estimate how pronounced are these wind speed

maxima. The authors' position could be strengthened if more robust evidence to associate the clusters with jets could be provided – a case study or some wind speed profiles that are not normalized and show how distinct the jet actually is.

[AR] Our conclusion was stated somewhat boldly. The "low-level jets" description only served to describe the shape of the resulting profiles. We rather refrain from discussing what defines a low-level jet and therefore phrased our conclusions more mildly.

[AC] Rephrased "low-level jets" to "jet-like shapes".

- If the conditions for cluster (5) MMIJ DOWA are analyzed (Fig. 9), than one could conclude that they are very similar to cluster (1) for MMIJ DOWA, with the exception of wind direction, so all the interpretation for conditions for cluster (1) MMIJ DOWA could apply to cluster (5) as well, with the exception that the upwind flow is advected over the water for shorter time, as the prevailing directions for cluster (5) are from South. In fact, in PC1-PC2 axis (Fig. 7) the cluster (5) is next to cluster (1). The problem is that for MMC such conditions are impossible and therefore cluster (5) for MMC is not the same as cluster (5) for MMIJ, nor in the shape of the profile and nor in the location of cluster in the PC1-PC2 axis. Therefore, I cannot agree with the statement that cluster (5) for MMC is similar to MMIJ (P17L20).

[AR] Our statement was not phrased carefully and therefore misinterpreted.

[AC] Rephrased to: "The profile shapes for MMC-5–7 are (slightly) jet-shaped, as is the case for the offshore clusters MMIJ-4–7." Also, the parallels drawn by the reviewer for the offshore clusters MMIJ-1,5 are now explicitly described in Sect. 4.2.

- Table 3, column "Most frequent area". What is this classification based on? Are the orography and land use data used in DOWA available? If so, then adding this information to Figure 15 as a plot would strengthen the authors' argument,
because currently, although I agree that features seen in Figure 15 are probably related to orography, again I would argue that more evidence is needed to support the authors' claims. I would not expect the general reader to be familiar with the topography of the Netherlands and Germany

[AR] DOWA does not include such data, at least it is not published. The classification was based on 'quick and dirty' observations, so not precise and well supported by data. We agree that the topography was missing.

[AC] We have discarded the "Most frequent area" column in the table, as they were blunt conclusions. We have included information on the surface elevation to Figure 1 and 16, such that the claims in the text are now better supported.

**3 Technical comments**

- Paper referenced as "Sommerfeld et al.": "Improving mid-altitude mesoscale wind speed forecasts using LiDAR based observation nudging for Airborne Wind Energy Systems" does not seem to contain anything related to k-means clustering. Maybe a different paper with the same first author is meant instead: Sommerfeld, Markus, et al. "LiDAR based characterization of mid altitude wind conditions for airborne wind energy systems." Wind Energy 22.8 (2019): 1101-1120.

  [AR] Correct

  [AC] Changed reference

- Order of figures. Figure 8 is referenced in text before Figure 7. Figure 8 is referenced in P11L10. Figure 7 is referenced in P11L21.

  [AC] Corrected

- P1L19: "Deviating profiles are likely to occur". I suggest "deviations from the expected profile shape are likely to occur".

[Figure]

Interactive
comment

[AC] Replaced with: "Moreover, within this layer, not all wind profiles can be described well with these relationships."

- P4L11: "Better representation of the coastal morphology". "Coastal morphology" is the study of natural processes that change the shape of coastline, e.g., erosion or sediment transport. I suggest using "coastline" or "better resolution of coastline" instead.

  [AC] Replaced with "coastline"

- P16L9: "the wind profile is typically will mixed". Probably, "well mixed".

  [AC] Corrected

---

## Referee Report (RR1)

I thank the authors for their work to make the manuscript better. I think that great improvements have been made. I still have some minor suggestions to improve the readability and make the paper more understandable for the general reader.

1. I am confused by the use of parentheses as a way to soften the meaning of words or to make some of the words feel "optional". According to my understanding such use of parentheses is not considered good style. Please consider removing the parentheses in sentences such as P18L16 "we also see frequent (very) stable conditions". I think that here "we also see frequent stable or very stable conditions" is much more precise.

2. The overbar in $\bar{\theta}_v$ in formula (5) is hard to see. Please check that this gets corrected in the final proof.

3. I would suggest adding a subscript for the symbol for the reference height z in (6), to avoid confusing that with the z in formula (2), where z has the meaning of a coordinate.

4. I think that some additional sentences are still needed in describing the sample normalization procedure (P6L16-L19), because I still struggled to understand the details. I would suggest explicitly defining reference wind velocity for each sample as the wind speed and direction of that sample at 100 m height, explicitly noting that each sample has its own reference wind velocity, because my first association is that "reference" is something that is common for all samples. I would suggest rephrasing P6L17 as "the value of the perpendicular component of the wind speed profile is 0 for each sample and for all averaged profiles".

5. Caption of Figure 8. I would suggest: "In the hodograph, the lowest level is indicated by the dotted line connecting the lowest level to the origin of coordinates".

6. I would rephrase the statement that MMIJ-5 has "relatively large deviations from logarithmic profile" P9L21, because later authors state that its shape is well described by unstable logarithmic profile, similarly in the caption for Figure 8 I would like it to be pointed out that these are stability adjusted logarithmic fits. I am stressing this point because a slightly distracted reader might confuse what exactly is meant by "logarithmic profile" because in some cases this term describes only the neutrally stable profile shape.

---

## Referee Report (RR2)

**Clustering wind profile shapes to estimate airborne wind energy production - 2nd Referee Comment**

Markus Sommerfeld[1]

[1]Institute for Integrated Energy Systems, University of Victoria,British Columbia, Canada

**Correspondence:** Markus Sommerfeld (msommerf@uvic.ca)

**1 General comments**

This paper is a useful contribution to better understand the wind energy potential of airborne wind energy systems (AWESs). The investigated onshore and offshore wind regimes make it especially interesting for regions close to the shore such as the Netherlands which this papers wind data is based on. Simulated Dutch Offshore Wind Atlas (DOWA) data is normalized, transformed using principal component analysis (PCA) and clustered to generate generalized wind profiles which are then scaled and fed into a quasi-steady AWES model to estimate power curves and annual energy production (AEP).

The manuscript improved considerably from the previous submission. Its content is more focused and its language is much clearer than before. Following are some general comments and language corrections.

Language:

The language improved a lot since the previous submission and previous ambiguities and mistakes have been removed. The article seems rather wordy. I could not get the exact word count, but copy-pasting the text into a text editor resulted in over 12000 words. Certain sentence structures with the word ´´respectivly" as well as gerund forms repeat fairly often. I therefore recommend rewriting and shortening some paragraphs by combining sentences and simplifying long and complicated sentences. Some examples can be found in section 2.

Figures:

The added figures 9 and 13 do not add significant information and could be summarized as text boxes in figure 8 or 12 which they refer to, e.g. don't mention L value, but associated stability bin in text box in each profile plot 1-8

Wind data: The usage of $Ri$, $L$ and $\Psi$ to assess atmospheric stability throughout the paper is rather confusing. Since $Ri$ and $L$ are interchangeable, it might make sense to just use one of them.

**2 Specific comments including technical corrections**

**2.1 Abstract**

page 1

line 4: ´´vertical variation" sounds like vertical (w) component of wind velocity. Maybe clarify by writing: variation in height or variation of horizontal wind speed.

line 6: Why introduce AWE and AEP, but not DOWA abbreviation in abstract?

line 7-8 + 12: The abstract should include a summary of information found in the paper. It should not provide results or conclusions.

line 10: Add: " ... for each wind speed profile shape".

line 12: 4 cluster error relative to what?

**2.2 Introduction**

page 1

line 14: Grammar: Comparative between altitude and turbines is not parallel; Comparison of stronger and more persistent winds to what? Rewrite: "AWE systems use tethered flying devices to harness energy at higher altitudes, typically heights above 150 m (Malz et al., 2019; Salma et al., 2019), where wind is generally stronger and more persistent (steady) than at heights of (reachable to) tower-based wind turbines."

line 20: Add reference to validity of log and power law beyond surface layer

equation 1+2: Only time $u$ used for wind speed. Other times $v$ or is $\hat{v}$ only used for normalised wind speeds?

page 2

line 13: Wording: "assumptions... frequently violated in practice...". Not the assumptions are violated, it's the range of validity that is ignored.

page 3

line 1: Change sentence order. Before you write about log fit. Rephrase to 1 sentence: "The wind direction can also vary substantially with height in the lower atmosphere."

line 2: What are "...scalar quantities...""? Remove before comma

**2.2.1 Wind dataset**

page 4

line 9: "..., any dataset containing time series..." (singular and add any)

line 10: Rephrase: "..we focus on sensitivity of the AWE system power production ..."; add "..to the wind profile shape ..."

line 13: What does typically refer to ? power assessment of wind turbines?

line 20: The spatial resolution needed for this study (country wide) does not exist for lidar. This is a justification to use model data instead.

line 21: Are these "typical" sites? Did you compare them to other sites? Maybe just write "representative" or "exemplary".

line 24+25: Repetitive use of "grass-land". Maybe merge sentences; spelling: grassland

line 26: Could mention that "down-scaled" means higher resolution

line 27: Remove sentence and put in parenthesis in previous sentence to reduce wordiness of text: "The sites (shown in figure 1) were chosen...". Add reference to studies using these locations wind data.

**2.2.2 ERA5**

page 5

line 2: ERA5 reference linked to https://cds.climate.copernicus.eu/cdsapp#!/home. Could instead use:

i.
```
\begin{small}
@misc{website:era5,
Author = {Hans Hersbach and Dee Dick},
Month = {November},
Title = {{ERA5} reanalysis is in production},
url = {http://www.ecmwf.int/en/newsletter/147/news/era5-reanalysis-production},
Year = {2016},
Publisher = {{ECMWF} - European Center for Medium Range Weather Forecast},
    note = {last accessed: 22.10.2019}}
```

line 2: Add: reference to ECMWF

line 7: "... is performed on the DOWA data..."

line 7: Remove: "As is explained later on"

**2.2.3 DOWA**

page 5

line 10: Remove: "The"; spelling of "downscaled" different from above.

line 13: Add "grid spacing" and "grid points" for clarity.

page 6

line 1: Add reference to improved DOWA performance.

line 3: Clarify "routine weather stations"

**2.3 Clustering procedure**

**2.3.1 Prepossessing of the wind data**

page 6

line 11: Remove: "the" before "wind speed" similar to no "the" before "direction"

line 11: Remove: "the" after "Therefore", because general wind profiles and not specific ones.

line 13: Separate into 2 sentences. Explain what you mean by processed using its own properties.

line 14: Rewrite for clarity; for example: "The wind speed components are expressed as parallel and perpendicular components relative to their reference wind velocity at 100 m, similar to Kalverla et al. (2017) and Malz et al. (2020a), thereby making them independent of wind directions."

line 18 + 19: Active voice and clarity; for example: "Wind velocity components are normalised using each profile's 90th percentile wind speed, because it reduces the amount of outliers in comparison to using the maximum speed of each profile. These normalised and decomposed samples are referred to as wind profile shape. "

line 21: Use irregular,atypical or unconventional instead of eccentric, add: "...low wind speeds."

line 24: Clarify: wind resource representation. Is this the wind speed probability distribution similar to a Weibull distribution? ;

**2.3.2 Principal component analysis of the wind profile shape dataset**

page 6

line 27: Replace: "while" with "which"

page 7

line 7: Replace: "layer" with "height"

line 14: Explain Fig 2 PC unit vectors. Are these the unit vectors in PC1 and PC2 domain which have a length not equal to one in $\hat{v}$, $height$ domain? Is it that the length ($\sqrt{PC1_\parallel^2 + PC1_\perp^2} = 1$) of the parallel (orange) and perpendicular (blue) component at each height is one? Are the orange and blue lines the direction of the PC at each height? Add: $\hat{v}$ is normalised wind speed magnitude in text.

line 16 + 17: Word order: mostly characterises.

line 18: What shows the large contribution of both PCs? Is it that the $\hat{v}$ magnitude is high? Does it make sense that most variance is top and bottom? I would expect that profiles within one cluster have similar wind speeds at high altitudes as they are probably driven by similar large scale weather phenomenon or is this lost due to normalisation and PCA?

line 20: Explain wind profiles shapes along PC1 and PC2 more. Why did you chose minus and plus one standard deviation as multipliers? What do you want to show with these profiles?

line 22: What are the eigenvalues of PCs? What are the retained PCs?

line 32: add: "Figure 4g which shows onshore data will be ..."

page 8

figure 2: What are the orange and blue dashed lines in column 2 PC1 and PC2? I think you did not explain $\hat{v}$ anywhere in the text or the caption. What does it mean that both PC1 and PC2 have negative parallel components below 300m?

**2.3.3 Choosing the number of clusters**

page 8

        line 2: Replace: applied instead of employed

        line 3: Replace: "...represented by its centroid ..."

        line 9: Do wind profiles have such a structure?

page 9

        figure 4: Mention for offshore a) and onshore b) in caption. Put figs 2 and 11 in parenthesis? replace: "deviation away from their mean" ?

        line 1+2: Remove "Moreover". Add reference to elbow and silhouette method find appropriate k.

        line 3: Replace: applied instead of employed. Can add on how evaluated: by comparing the estimated AEP

        line 5: Reference: appropriate choices of k?

page 10

        figure 5 a): Remove: "the" before cost function and cluster cohesiveness. The error between which data? mean wind speed error between each profile and its respective cluster centroid?

        figure 5: Add: "k-means" before clustering ; remove: point before (b)

        line 5: Maybe use scaling instead of denormalisation; remove "of the cluster to which it is assigned"; replace "of" with "used in pre-processing"

        line 7: Add : $\varepsilon_{ij}$ here to make equation easiert to understand. Define between which data the error is calculated.

        line 8: Is the representation accuracy shown in Fig 5?

page 11

        line : Is the error only based on different 5 $\Psi$ values?

        line 5: Sentence needed?

        line 6: Add reference to $\Psi$ function; replace every with each

        line 14: Grammar; add (bottom) to Fig 5a

        line 15: Is " Note that..." sentence necessary here? Maybe move to where you introduce WCSS?

        line 19: Why not just wind speed error? $\overline{\varepsilon_i}$ never used in any equation

        line 20: Replace top and bottom with above and below.

        line 21: Replace vertical grid with dataset or rewrite

150      line 5: Limited to what? Rather a fixed number of clusters.

         line 6: What do you mean by: " ... and our aim to present a meaningful analysis and interpretation of the resulting clusters." and how did this affect your choice of k?

         line 8: Isn't it just the silhouette score and not the mean silhouette score in both sentences?

**2.4   Cluster wind resource representation**

155   page 13

         figure 7: What does the asterisk mean? Maybe rename axis label to $PC1_{onshore}$ / $PC1_{offshore}$ ?

         line 5: Even though clusters are not the same between on and offshore.

         line 6: Replace: full with entire

**2.4.1   Cluster representation for the offshore location**

160   page 13

         line 9: Replace: of with at; add: their centroids

         line 15: Add: backwards transformed from PC space to normalised wind velocity

         line 15: Explain: stability function is varied continuously?

      page 14

165      table 2: Replace: at with in?

         line 3: Is this sentence about a) ? remove: is, and & represented " In contrast to the other panels, the absolute frequency on the y-axis serves to show which part of the total dataset is presented "

         line 6: Replace: for with of

         line 7: Where is "here"? Figure 9, 10? Isn't it only one distribution?

170      line 9: Which data did you use? $L$ is always calculated using surface data (see: http://glossary.ametsoc.org/wiki/Obukhov_length)

      page 16

         figure 9: You could replace figure 9 and 13 with legend or text box in figure 8 and 12 to save space. Using a figure to show this seems unnecessary

175   page 17

         figure 10: Add a),b),c),d),e),f) to caption; Describe what is shown in a,b,c. What is the benefit of using this $v_{100}$ binning or what is the reason for having equal bin size for wind speed?

**2.4.2 Interpretation of the offshore cluster representation**

page 18

180       line : Maybe makes more sense to put interpretation in the same paragraph where figure is described? This way you have to flip back and forth a lot as

line 5: Add: magnitude of profiles? replace: well-described

line 6: Replace: shear and veer

line 8: Shorten: North-West direction

185       line 10: Equilibrium profile meaning a unstable shape?

line 25: Remove: "The frequent"; Rewrite: Winds at this location with a southerly component...

line 29: Remove: "... and gradually"

page 19

line 1: Replace: "abrupt kink" with "sharp bend"

190       line 5: Replace: "rather than" with "and less often with"

line 6: Replace: shear and veer

line 8: Shorten: North-West direction

line 10: Equilibrium profile meaning a unstable shape?

line 25: Remove: "The frequent"; Rewrite: Winds at this location with a southerly component...

195       line 29: Remove: "... and gradually"

**2.4.3 Comparing the on- and offshore cluster representations**

page 19

line 8: Rewrite: The onshore data at the met mast Cabauw is clustered using the same methodology.

line 9: Compared how and where?

200       line 12: Rewrite: "the mean profile shape below 200 m is in accordance with a stable logarithmic profile."

line 15: Increased relative to what?

line 16: Add reference to where they are plotted

line 16: If they share the same coordinate system does that mean that both locations have the same PCs?

line 24: "...clustering algorithm..."; ".... for each cluster."

205        line 25: Replace: more or less with onshore closely resembles offshore

line 28: Remove: again

line 29: Obukhov length determined how?

line 30: Meaning? "show an increase in wind shear". Increasing wind gradient from 1 to 3?

line 34: Replace: "turning" with "rotation" and "kink" with "sharp bend"

210   page 20

line 4: Rephrase: The frequency distribution over the first five onshore clusters is more balanced than the offshore clusters which show one distinct dominant cluster.

line 9: Rephrase: Convection occurs in the presence of daytime solar irradiation which leads to the development of well-mixed wind speed profiles.

215   line 12: Explain: "Patterns in times of occurrences" of what?

line 13: Explain: Meaning of " almost identical bin distributions" if wind speed distribution is different?

line 14: Explain: Reason for choosing wind speed bin limits this way?

line 15: Rephrase: Total frequency of each bin is roughly the same over the entire dataset. Explain: What is the benefit of this approach?

220   line 16: How can they show similar stability distributions if figure 9 and 13 are totally different?

line 22: Replace: "... start to be observed" with "are observed."

page 21 + 22

figure 11+12: See same figures for offshore

page 23

225   figure 13: Very strange that 6 out of 8 profile shapes are stable. Add to caption: fitted up to 200m

page 24

figure 14: Same as for previous similar figure. Rephrase: "Frequency distributions by time of occurrence..."

**2.4.4 Spatial frequency distribution of wind profile shape clusters**

page 25

230   line 2: Replace: "are" with "were"

line 3: Replace: "for" with "from"

line  10: Remove: "So"

line  12: Replace: "inherent" with "synonymous" Inherent turns the relationship between reducing error and increasing n clusters around.

line  21:Can you quantify how much they "look alike"?

line  22: 21.7% ?Reference Table 3 here already

line  27: Which terrain features? Elevation? Forests, cities?

page  26

table  3: How do you quantify similarity?

page  28

figure  16: Mention in caption: Each sample of every grid point is assigned to the cluster with the closest centroid.

**2.5   Efficient AWE production estimation using the cluster representation**

page  29

line  6: Add: power curve and AEP

line  7: Replace: "winds higher up" with "winds at higher altitudes"

line  11: Add: "wind speed distribution within the corresponding cluster"; Replace: "of the clusters" with "of each cluster"

**2.5.1   Constructing the power curves**

page  29

line  13: Never heard "construction" in the context of power curves. Replace throughout the document with derive or determine ?

line  15: Fix grammar (reel-out twice); mention change in angle of attack;

page  30

line  2: What is V3 kite?

line  4: Add: "average energy production"

page  31

line  1: Replace: "are exceeded" with "would be exceeded"? Simplify sentence: "Reel-out" repetition in same sentence. Why is reel-out duration important?

line 6+8: What are cycle settings? Are these the fixed constraint in table 5?

line 15-18: a bit wordy, could be shortened to 1 or 2 sentences.

line 17: Add "reference wind speed"; Remove: "to yield the absolute wind profile"

table 5: Is tether diameter and associated drag considered in the model?

line 20: Replace: "smallest" with "lowest"; "whole" with "entire"

line 1: Replace: "are exceeded" with "would be exceeded"?

page 32

line 9: Does this intersection depend on tether length?

figure 18: Replace: "from " with "in"; also reference which section describes these profiles in fig 12; Add: "pre-determined" before cut-in

line 12: Simplify with active voice: "The depicted trajectories highlight operational changes occurring at different reference wind speeds"; Specify how the approach / attitude changes. Is this the reason for the strange $v_{ref}$ in legend of Fig 19?

line 13: The fact that the tether length constraint is always active indicates that the global optimum is beyond this constraint. Can you comment on whether you tried out different settings or what the reasons could be for maxing out this constraint?

line 14: Same is true for lower wind seeds though?!

line 16: criterion is also a constraint?

line 19: They are similar. Are they cubic as we would expect from $P \sim \rho v^3$

page 33

figure 19: Add: "same normalised wind speed profile shape"; Approach = attitude? Specify what you mean by change in approach.

figure 20: Frequency better in %; Add: "scaled cluster-mean..."; Why did you aggregate 4 bins into 1? Not sure if you have to mention this here.

line 1: LLJ benefit because of high wind speed at low elevation angle hence low cosine losses?

**2.5.2 Estimating the annual energy production**

page 34

line 2: Replace "Constructed" with "derived" ?; Grammar: "... are used to calculate the average ..."

line 4: Is $P_i$ the power curve or power?

line 8: Is the numerical error so significant that you need to use 100 bins instead of the way more common 1m/s or 0.5m/s bins?

line 10: Isn't it that you just use the reference wind speed $v_{100m}$ to calculate the frequency and they make up these bins?

line 19: What do you mean by "...inaccuracy of the cluster wind resource representation"? Does it mean that fewer clusters lead to higher inaccuracies because of averaging or misrepresentation of the entiere wind resource?

line 22: Rewrite: It's hard to understand how MMC and ML relate to each other and what 16 and 32 clusters mean. Does it mean that 16 MMC clusters have the same difference to converged value as 32 ML clusters? Doesn't this indicate that you need twice as many clusters to achieve similar quality using the ML approach? Which makes sense as it combines on and offshore locations and therefore different flow regimes?

line 28: Reference figure 20 here again. 50 optimisations per power curve because of the step size of $\Delta v_{scale}$ you chose? Maybe mention step size for clarification.

line 29: Remove of: "...half the number"; Replace; ".. can be used" with "yield similar results with half the computational cost"

line 30: Grammar: "used for generating" with "used to generate"; Rewrite "in comparison to brute force, where 8760 optimisations ...."

page 35

line 1:"orders of magnitude lower"

figure 21:"Comparison of AEP conversion between MMC onshore and ML offshore"

**2.5.3 Conclusion**

page 35

line 1: Remove: "have"; add: "a set of normalised wind profile shapes"

line 6: Did you quantify the occurrence of LLJs? There are also other ways of doing that. You approach allows the inclusion of wind profile shapes into the wind resource description (equivalent to wind speed distribution/ Weibull for conventional turbines).

line 7: Grammar: "We demonstrated this methodology for two reference locations on and offshore based on the DOWA dataset."

line 9: What do you mean by "expressed in terms of wind velocity at 100 m?" and where did you do it?

line 10: "... profile shape variance."

line 15: Rewrite: "The DOWA dataset is partitioned using k-means clustering. The resulting cluster-mean wind profile shapes are used to represent the wind resource, thereby reducing the wide range of wind conditions to a reasonable number of wind profile shapes."

line 18: Rewrite: "Although some variability is lost by only using the mean cluster profiles, ..."

320      line 21: Order: "... 8 offshore clusters show 3 monotonic ..."

page 36

line 1: Replace: "entire DOWA domain"

line 3: Replace: "... between the profile shape and terrain."

line 3: Replace: "... in contrast to 8760 for an ..."

325      line 16: Grammar: "...ML enables an assessment..."

line 18: Replace: "...in estimating the AEP..."

---

## Editor Decision (ED1)

I thank the authors for their work to make the manuscript better. I think that great improvements have been made. I still have some minor suggestions to improve the readability and make the paper more understandable for the general reader.

1. I am confused by the use of parentheses as a way to soften the meaning of words or to make some of the words feel "optional". According to my understanding such use of parentheses is not considered good style. Please consider removing the parentheses in sentences such as P18L16 "we also see frequent (very) stable conditions". I think that here "we also see frequent stable or very stable conditions" is much more precise.

2. The overbar in $\bar{\theta}_v$ in formula (5) is hard to see. Please check that this gets corrected in the final proof.

3. I would suggest adding a subscript for the symbol for the reference height z in (6), to avoid confusing that with the z in formula (2), where z has the meaning of a coordinate.

4. I think that some additional sentences are still needed in describing the sample normalization procedure (P6L16-L19), because I still struggled to understand the details. I would suggest explicitly defining reference wind velocity for each sample as the wind speed and direction of that sample at 100 m height, explicitly noting that each sample has its own reference wind velocity, because my first association is that "reference" is something that is common for all samples. I would suggest rephrasing P6L17 as "the value of the perpendicular component of the wind speed profile is 0 for each sample and for all averaged profiles".

5. Caption of Figure 8. I would suggest: "In the hodograph, the lowest level is indicated by the dotted line connecting the lowest level to the origin of coordinates".

6. I would rephrase the statement that MMIJ-5 has "relatively large deviations from logarithmic profile" P9L21, because later authors state that its shape is well described by unstable logarithmic profile, similarly in the caption for Figure 8 I would like it to be pointed out that these are stability adjusted logarithmic fits. I am stressing this point because a slightly distracted reader might confuse what exactly is meant by "logarithmic profile" because in some cases this term describes only the neutrally stable profile shape.

**Clustering wind profile shapes to estimate airborne wind energy production - 2nd Referee Comment**

Markus Sommerfeld[1]

[1]Institute for Integrated Energy Systems, University of Victoria,British Columbia, Canada

**Correspondence:** Markus Sommerfeld (msommerf@uvic.ca)

**1 General comments**

This paper is a useful contribution to better understand the wind energy potential of airborne wind energy systems (AWESs). The investigated onshore and offshore wind regimes make it especially interesting for regions close to the shore such as the Netherlands which this papers wind data is based on. Simulated Dutch Offshore Wind Atlas (DOWA) data is normalized, transformed using principal component analysis (PCA) and clustered to generate generalized wind profiles which are then scaled and fed into a quasi-steady AWES model to estimate power curves and annual energy production (AEP).

The manuscript improved considerably from the previous submission. Its content is more focused and its language is much clearer than before. Following are some general comments and language corrections.

Language:

The language improved a lot since the previous submission and previous ambiguities and mistakes have been removed. The article seems rather wordy. I could not get the exact word count, but copy-pasting the text into a text editor resulted in over 12000 words. Certain sentence structures with the word ´´respectivly'' as well as gerund forms repeat fairly often. I therefore recommend rewriting and shortening some paragraphs by combining sentences and simplifying long and complicated sentences. Some examples can be found in section 2.

Figures:

The added figures 9 and 13 do not add significant information and could be summarized as text boxes in figure 8 or 12 which they refer to, e.g. don't mention L value, but associated stability bin in text box in each profile plot 1-8

Wind data: The usage of $Ri$, $L$ and $\Psi$ to assess atmospheric stability throughout the paper is rather confusing. Since $Ri$ and $L$ are interchangeable, it might make sense to just use one of them.

**2 Specific comments including technical corrections**

**2.1 Abstract**

page 1

line 4: ´´vertical variation'' sounds like vertical (w) component of wind velocity. Maybe clarify by writing: variation in height or variation of horizontal wind speed.

line 6: Why introduce AWE and AEP, but not DOWA abbreviation in abstract?

line 7-8 + 12: The abstract should include a summary of information found in the paper. It should not provide results or conclusions.

line 10: Add: " ... for each wind speed profile shape".

line 12: 4 cluster error relative to what?

**2.2 Introduction**

page 1

line 14: Grammar: Comparative between altitude and turbines is not parallel; Comparison of stronger and more persistent winds to what? Rewrite: "AWE systems use tethered flying devices to harness energy at higher altitudes, typically heights above 150 m (Malz et al., 2019; Salma et al., 2019), where wind is generally stronger and more persistent (steady) than at heights of (reachable to) tower-based wind turbines."

line 20: Add reference to validity of log and power law beyond surface layer

equation 1+2: Only time $u$ used for wind speed. Other times $v$ or is $\hat{v}$ only used for normalised wind speeds?

page 2

line 13: Wording: "assumptions... frequently violated in practice...". Not the assumptions are violated, it's the range of validity that is ignored.

page 3

line 1: Change sentence order. Before you write about log fit. Rephrase to 1 sentence: "The wind direction can also vary substantially with height in the lower atmosphere."

line 2: What are "...scalar quantities..."? Remove before comma

**2.2.1 Wind dataset**

page 4

line 9: "..., any dataset containing time series..." (singular and add any)

line 10: Rephrase: "..we focus on sensitivity of the AWE system power production ..."; add "..to the wind profile shape ..."

line 13: What does typically refer to ? power assessment of wind turbines?

line 20: The spatial resolution needed for this study (country wide) does not exist for lidar. This is a justification to use model data instead.

line 21: Are these "typical" sites? Did you compare them to other sites? Maybe just write "representative" or "exemplary".

line 24+25: Repetitive use of "grass-land". Maybe merge sentences; spelling: grassland

line 26: Could mention that "down-scaled" means higher resolution

line 27: Remove sentence and put in parenthesis in previous sentence to reduce wordiness of text: "The sites (shown in figure 1) were chosen...". Add reference to studies using these locations wind data.

**2.2.2 ERA5**

page 5

line 2: ERA5 reference linked to https://cds.climate.copernicus.eu/cdsapp#!/home. Could instead use:

     i.

```
\begin{small}
@misc{website:era5,
Author = {Hans Hersbach and Dee Dick},
Month = {November},
Title = {{ERA5} reanalysis is in production},
url = {http://www.ecmwf.int/en/newsletter/147/news/era5-reanalysis-production},
Year = {2016},
Publisher = {{ECMWF} - European Center for Medium Range Weather Forecast},
  note = {last accessed: 22.10.2019}}
```

line 2: Add: reference to ECMWF

line 7: "... is performed on the DOWA data..."

line 7: Remove: "As is explained later on"

**2.2.3 DOWA**

page 5

line 10: Remove: "The"; spelling of "downscaled" different from above.

line 13: Add "grid spacing" and "grid points" for clarity.

page 6

line 1: Add reference to improved DOWA performance.

line 3: Clarify "routine weather stations"

**2.3 Clustering procedure**

**2.3.1 Prepossessing of the wind data**

page 6

line 11: Remove: "the" before "wind speed" similar to no "the" before "direction"

line 11: Remove: "the" after "Therefore", because general wind profiles and not specific ones.

line 13: Separate into 2 sentences. Explain what you mean by processed using its own properties.

line 14: Rewrite for clarity; for example: "The wind speed components are expressed as parallel and perpendicular components relative to their reference wind velocity at 100 m, similar to Kalverla et al. (2017) and Malz et al. (2020a), thereby making them independent of wind directions."

line 18 + 19: Active voice and clarity; for example: "Wind velocity components are normalised using each profile's 90th percentile wind speed, because it reduces the amount of outliers in comparison to using the maximum speed of each profile. These normalised and decomposed samples are referred to as wind profile shape. "

line 21: Use irregular,atypical or unconventional instead of eccentric, add: "...low wind speeds."

line 24: Clarify: wind resource representation. Is this the wind speed probability distribution similar to a Weibull distribution? ;

**2.3.2 Principal component analysis of the wind profile shape dataset**

page 6

line 27: Replace: "while" with "which"

page 7

line 7: Replace: "layer" with "height"

line 14: Explain Fig 2 PC unit vectors. Are these the unit vectors in PC1 and PC2 domain which have a length not equal to one in $\hat{v}$, $height$ domain? Is it that the length ($\sqrt{PC1_\parallel^2 + PC1_\perp^2} = 1$) of the parallel (orange) and perpendicular (blue) component at each height is one? Are the orange and blue lines the direction of the PC at each height? Add: $\hat{v}$ is normalised wind speed magnitude in text.

line 16 + 17: Word order: mostly characterises.

line 18: What shows the large contribution of both PCs? Is it that the $\hat{v}$ magnitude is high? Does it make sense that most variance is top and bottom? I would expect that profiles within one cluster have similar wind speeds at high altitudes as they are probably driven by similar large scale weather phenomenon or is this lost due to normalisation and PCA?

line 20: Explain wind profiles shapes along PC1 and PC2 more. Why did you chose minus and plus one standard deviation as multipliers? What do you want to show with these profiles?

line 22: What are the eigenvalues of PCs? What are the retained PCs?

line 32: add: "Figure 4g which shows onshore data will be ..."

page 8

figure 2: What are the orange and blue dashed lines in column 2 PC1 and PC2? I think you did not explain $\hat{v}$ anywhere in the text or the caption. What does it mean that both PC1 and PC2 have negative parallel components below 300m?

**2.3.3 Choosing the number of clusters**

page 8

    line 2: Replace: applied instead of employed

    line 3: Replace: "...represented by its centroid ..."

    line 9: Do wind profiles have such a structure?

page 9

    figure 4: Mention for offshore a) and onshore b) in caption. Put figs 2 and 11 in parenthesis? replace: "deviation away from their mean" ?

    line 1+2: Remove "Moreover". Add reference to elbow and silhouette method find appropriate k.

    line 3: Replace: applied instead of employed. Can add on how evaluated: by comparing the estimated AEP

    line 5: Reference: appropriate choices of k?

page 10

    figure 5 a): Remove: "the" before cost function and cluster cohesiveness. The error between which data? mean wind speed error between each profile and its respective cluster centroid?

    figure 5: Add: "k-means" before clustering ; remove: point before (b)

    line 5: Maybe use scaling instead of denormalisation; remove "of the cluster to which it is assigned"; replace "of" with "used in pre-processing"

    line 7: Add : $\varepsilon_{ij}$ here to make equation easiert to understand. Define between which data the error is calculated.

    line 8: Is the representation accuracy shown in Fig 5?

page 11

    line : Is the error only based on different 5 $\Psi$ values?

    line 5: Sentence needed?

    line 6: Add reference to $\Psi$ function; replace every with each

    line 14: Grammar; add (bottom) to Fig 5a

    line 15: Is " Note that..." sentence necessary here? Maybe move to where you introduce WCSS?

    line 19: Why not just wind speed error? $\overline{\varepsilon_i}$ never used in any equation

    line 20: Replace top and bottom with above and below.

    line 21: Replace vertical grid with dataset or rewrite

150    line  5: Limited to what? Rather a fixed number of clusters.

       line  6: What do you mean by: " ... and our aim to present a meaningful analysis and interpretation of the resulting clusters." and how did this affect your choice of k?

       line  8: Isn't it just the silhouette score and not the mean silhouette score in both sentences?

**2.4    Cluster wind resource representation**

155    page  13

       figure  7: What does the asterisk mean? Maybe rename axis label to $PC1_{onshore}$ / $PC1_{offshore}$ ?

       line  5: Even though clusters are not the same between on and offshore.

       line  6: Replace: full with entire

**2.4.1    Cluster representation for the offshore location**

160    page  13

       line  9: Replace: of with at; add: their centroids

       line  15: Add: backwards transformed from PC space to normalised wind velocity

       line  15: Explain: stability function is varied continuously?

       page  14

165    table  2: Replace: at with in?

       line  3: Is this sentence about a) ? remove: is, and & represented " In contrast to the other panels, the absolute frequency on the y-axis serves to show which part of the total dataset is presented "

       line  6: Replace: for with of

       line  7: Where is "here"? Figure 9, 10? Isn't it only one distribution?

170    line  9: Which data did you use? $L$ is always calculated using surface data (see: http://glossary.ametsoc.org/wiki/Obukhov_length)

       page  16

       figure  9: You could replace figure 9 and 13 with legend or text box in figure 8 and 12 to save space. Using a figure to show this seems unnecessary

175    page  17

       figure  10: Add a),b),c),d),e),f) to caption; Describe what is shown in a,b,c. What is the benefit of using this $v_{100}$ binning or what is the reason for having equal bin size for wind speed?

**2.4.2   Interpretation of the offshore cluster representation**

page  18

180      line  : Maybe makes more sense to put interpretation in the same paragraph where figure is described? This way you have to flip back and forth a lot as

line  5: Add: magnitude of profiles? replace: well-described

line  6: Replace: shear and veer

line  8: Shorten: North-West direction

185      line  10: Equilibrium profile meaning a unstable shape?

line  25: Remove: "The frequent"; Rewrite: Winds at this location with a southerly component...

line  29: Remove: "... and gradually"

page  19

line  1: Replace: "abrupt kink" with "sharp bend"

190      line  5: Replace: "rather than" with "and less often with"

line  6: Replace: shear and veer

line  8: Shorten: North-West direction

line  10: Equilibrium profile meaning a unstable shape?

line  25: Remove: "The frequent"; Rewrite: Winds at this location with a southerly component...

195      line  29: Remove: "... and gradually"

**2.4.3   Comparing the on- and offshore cluster representations**

page  19

line  8: Rewrite: The onshore data at the met mast Cabauw is clustered using the same methodology.

line  9: Compared how and where?

200      line  12: Rewrite: "the mean profile shape below 200 m is in accordance with a stable logarithmic profile."

line  15: Increased relative to what?

line  16: Add reference to where they are plotted

line  16: If they share the same coordinate system does that mean that both locations have the same PCs?

line  24: "...clustering algorithm..."; ".... for each cluster."

205        line 25: Replace: more or less with onshore closely resembles offshore

line 28: Remove: again

line 29: Obukhov length determined how?

line 30: Meaning? "show an increase in wind shear". Increasing wind gradient from 1 to 3?

line 34: Replace: "turning" with "rotation" and "kink" with "sharp bend"

210   page 20

line 4: Rephrase: The frequency distribution over the first five onshore clusters is more balanced than the offshore clusters which show one distinct dominant cluster.

line 9: Rephrase: Convection occurs in the presence of daytime solar irradiation which leads to the development of well-mixed wind speed profiles.

215   line 12: Explain: "Patterns in times of occurrences" of what?

line 13: Explain: Meaning of " almost identical bin distributions" if wind speed distribution is different?

line 14: Explain: Reason for choosing wind speed bin limits this way?

line 15: Rephrase: Total frequency of each bin is roughly the same over the entire dataset. Explain: What is the benefit of this approach?

220   line 16: How can they show similar stability distributions if figure 9 and 13 are totally different?

line 22: Replace: "... start to be observed" with "are observed."

page 21 + 22

figure 11+12: See same figures for offshore

page 23

225   figure 13: Very strange that 6 out of 8 profile shapes are stable. Add to caption: fitted up to 200m

page 24

figure 14: Same as for previous similar figure. Rephrase: "Frequency distributions by time of occurrence..."

**2.4.4 Spatial frequency distribution of wind profile shape clusters**

page 25

230   line 2: Replace: "are" with "were"

line 3: Replace: "for" with "from"

line 10: Remove: "So"

line 12: Replace: "inherent" with "synonymous" Inherent turns the relationship between reducing error and increasing n clusters around.

line 21:Can you quantify how much they "look alike"?

line 22: 21.7% ?Reference Table 3 here already

line 27: Which terrain features? Elevation? Forests, cities?

page 26

table 3: How do you quantify similarity?

page 28

figure 16: Mention in caption: Each sample of every grid point is assigned to the cluster with the closest centroid.

**2.5 Efficient AWE production estimation using the cluster representation**

page 29

line 6: Add: power curve and AEP

line 7: Replace: "winds higher up" with "winds at higher altitudes"

line 11: Add: "wind speed distribution within the corresponding cluster"; Replace: "of the clusters" with "of each cluster"

**2.5.1 Constructing the power curves**

page 29

line 13: Never heard "construction" in the context of power curves. Replace throughout the document with derive or determine ?

line 15: Fix grammar (reel-out twice); mention change in angle of attack;

page 30

line 2: What is V3 kite?

line 4: Add: "average energy production"

page 31

line 1: Replace: "are exceeded" with "would be exceeded"? Simplify sentence: "Reel-out" repetition in same sentence. Why is reel-out duration important?

line 6+8: What are cycle settings? Are these the fixed constraint in table 5?

line 15-18: a bit wordy, could be shortened to 1 or 2 sentences.

line 17: Add "reference wind speed"; Remove: "to yield the absolute wind profile"

table 5: Is tether diameter and associated drag considered in the model?

line 20: Replace: "smallest" with "lowest"; "whole" with "entire"

line 1: Replace: "are exceeded" with "would be exceeded"?

page 32

line 9: Does this intersection depend on tether length?

figure 18: Replace: "from " with "in"; also reference which section describes these profiles in fig 12; Add: "pre-determined" before cut-in

line 12: Simplify with active voice: "The depicted trajectories highlight operational changes occurring at different reference wind speeds"; Specify how the approach / attitude changes. Is this the reason for the strange $v_{ref}$ in legend of Fig 19?

line 13: The fact that the tether length constraint is always active indicates that the global optimum is beyond this constraint. Can you comment on whether you tried out different settings or what the reasons could be for maxing out this constraint?

line 14: Same is true for lower wind seeds though?!

line 16: criterion is also a constraint?

line 19: They are similar. Are they cubic as we would expect from $P \sim \rho v^3$

page 33

figure 19: Add: "same normalised wind speed profile shape"; Approach = attitude? Specify what you mean by change in approach.

figure 20: Frequency better in %; Add: "scaled cluster-mean..."; Why did you aggregate 4 bins into 1? Not sure if you have to mention this here.

line 1: LLJ benefit because of high wind speed at low elevation angle hence low cosine losses?

**2.5.2 Estimating the annual energy production**

page 34

line 2: Replace "Constructed" with "derived" ?; Grammar: "... are used to calculate the average ..."

line 4: Is $P_i$ the power curve or power?

line 8: Is the numerical error so significant that you need to use 100 bins instead of the way more common 1m/s or 0.5m/s bins?

line 10: Isn't it that you just use the reference wind speed $v_{100m}$ to calculate the frequency and they make up these bins?

line 19: What do you mean by "...inaccuracy of the cluster wind resource representation"? Does it mean that fewer clusters lead to higher inaccuracies because of averaging or misrepresentation of the entiere wind resource?

line 22: Rewrite: It's hard to understand how MMC and ML relate to each other and what 16 and 32 clusters mean. Does it mean that 16 MMC clusters have the same difference to converged value as 32 ML clusters? Doesn't this indicate that you need twice as many clusters to achieve similar quality using the ML approach? Which makes sense as it combines on and offshore locations and therefore different flow regimes?

line 28: Reference figure 20 here again. 50 optimisations per power curve because of the step size of $\Delta v_{scale}$ you chose? Maybe mention step size for clarification.

line 29: Remove of: "...half the number"; Replace; ".. can be used" with "yield similar results with half the computational cost"

line 30: Grammar: "used for generating" with "used to generate"; Rewrite "in comparison to brute force, where 8760 optimisations ...."

page 35

line 1:"orders of magnitude lower"

figure 21:"Comparison of AEP conversion between MMC onshore and ML offshore"

**2.5.3 Conclusion**

page 35

line 1: Remove: "have"; add: "a set of normalised wind profile shapes"

line 6: Did you quantify the occurrence of LLJs? There are also other ways of doing that. You approach allows the inclusion of wind profile shapes into the wind resource description (equivalent to wind speed distribution/ Weibull for conventional turbines).

line 7: Grammar: "We demonstrated this methodology for two reference locations on and offshore based on the DOWA dataset."

line 9: What do you mean by "expressed in terms of wind velocity at 100 m?" and where did you do it?

line 10: "... profile shape variance."

line 15: Rewrite: "The DOWA dataset is partitioned using k-means clustering. The resulting cluster-mean wind profile shapes are used to represent the wind resource, thereby reducing the wide range of wind conditions to a reasonable number of wind profile shapes."

line  18: Rewrite: "Although some variability is lost by only using the mean cluster profiles, ..."

320     line  21: Order: "... 8 offshore clusters show 3 monotonic ..."

page  36

line  1: Replace: "entire DOWA domain"

line  3: Replace: "... between the profile shape and terrain."

line  3: Replace: "... in contrast to 8760 for an ..."

325     line  16: Grammar: "...ML enables an assessment..."

line  18: Replace: "...in estimating the AEP..."

---

## Author Response (AR2)

**Response to 2nd review of referee #1**

Mark Schelbergen, Peter C. Kalverla, Roland Schmehl, and Simon J. Watson

Thank you for another comprehensive and useful list of comments. We feel that they were very helpful for removing the last flaws in the paper. The most important changes to the paper include:

1. The profiles in the 2nd column of Fig. 2 and 11 are now better explained.

2. Section 4.3 includes an explanation on why only 1 cluster groups the unstable profile shapes.

3. A lot of unclear relationships in the text are clarified.

We respond to the referee comments by including our answers below the original comments. Our answers are preceded by one or both of the following labels:

1. [AR] = author's response

2. [AC] = author's changes in manuscript

**1 General comments**

This paper is a useful contribution to better understand the wind energy potential of airborne wind energy systems (AWESs). The investigated onshore and offshore wind regimes make it especially interesting for regions close to the shore such as the Netherlands which this papers wind data is based on. Simulated Dutch Offshore Wind Atlas (DOWA) data is normalized, transformed using principal component analysis (PCA) and clustered to generate generalized wind profiles which are then scaled and fed into a quasi-steady AWES model to estimate power curves and annual energy production (AEP).
The manuscript improved considerably from the previous submission. Its content is more focused and its language is much clearer than before. Following are some general comments and language corrections.

Language:
The language improved a lot since the previous submission and previous ambiguities and mistakes have been removed. The article seems rather wordy. I could not get the exact word count, but copy-pasting the text into a text editor resulted in over 12000 words. Certain sentence structures with the word ´´respectively" as well as gerund forms repeat fairly often. I therefore recommend rewriting and shortening some paragraphs by combining sentences and simplifying long and complicated sentences. Some examples can be found in section 2.

[AR] Improved the language as suggested by the reviewer in section 2.

Figures:

 The added figures 9 and 13 do not add significant information and could be summarized as text boxes in figure 8 or 12 which they refer to, e.g. don't mention L value, but associated stability bin in text box in each profile plot 1-8

[AR] These plots were added to depict how the L found for each shape compares to the others, this information would be lost by using the approach suggested by the reviewer. This information is used throughout e.g. Sec. 4.2.

Wind data: The usage of $Ri$, $L$ and $\Psi$ to assess atmospheric stability throughout the paper is rather confusing. Since $Ri$ and $L$ are interchangeable, it might make sense to just use one of them.

[AR] We only present results in terms of $L$. $Ri$ is only introduced for explaining how we get to $L$ from the ERA5 data.

**2 Specific comments including technical corrections**

**2.1 Abstract**

page 1

line 4: ´´vertical variation" sounds like vertical (w) component of wind velocity. Maybe clarify by writing: variation in
 height or variation of horizontal wind speed.

[AC] Changed to: "... variation ... with height."

line 6: Why introduce AWE and AEP, but not DOWA abbreviation in abstract?

[AR] I don't use DOWA multiple times in the abstract, in contrast to AWE and AEP.

line 7-8 + 12: The abstract should include a summary of information found in the paper. It should not provide results or
 conclusions.

[AR] To my understanding the abstract should include information of the whole paper, including results and conclusions.

line 10: Add: " ... for each wind speed profile shape".

[AC] Changed

 line 12: 4 cluster error relative to what?

[AC] Changed to: "... the difference in AEP with respect to the converged value is within three percent for four or more clusters."

**2.2 Introduction**

page 1

line 14: Grammar: Comparative between altitude and turbines is not parallel; Comparison of stronger and more persistent winds to what? Rewrite: "AWE systems use tethered flying devices to harness energy at higher altitudes, typically heights above 150 m (Malz et al., 2019; Salma et al., 2019), where wind is generally stronger and more persistent (steady) than at heights of (reachable to) tower-based wind turbines."

[AC] Rephrased

line 20: Add reference to validity of log and power law beyond surface layer

[AR] Such references are given below seperately for the two relationships.

equation 1+2: Only time $u$ used for wind speed. Other times $v$ or is $\hat{v}$ only used for normalised wind speeds?

[AC] Changed to $v$

page 2

line 13: Wording: "assumptions... frequently violated in practice...". Not the assumptions are violated, it's the range of validity that is ignored.

[AC] Rephrased

page 3

line 1: Change sentence order. Before you write about log fit. Rephrase to 1 sentence: "The wind direction can also vary substantially with height in the lower atmosphere."

[AC] Rephrased

line 2: What are "...scalar quantities..."? Remove before comma

[AC] Removed

**2.2.1 Wind dataset**

page 4

line 9: "..., any dataset containing time series..." (singular and add any)

[AC] Corrected

line 10: Rephrase: "..we focus on sensitivity of the AWE system power production ..."; add "..to the wind profile shape ..."

[AC] Corrected

line 13: What does typically refer to ? power assessment of wind turbines?

[AC] Rephrased

line 20: The spatial resolution needed for this study (country wide) does not exist for lidar. This is a justification to use model data instead.

[AR] As is implied.

line 21: Are these "typical" sites? Did you compare them to other sites? Maybe just write "representative" or "exemplary".

[AC] Left out typical

line 24+25: Repetitive use of "grass-land". Maybe merge sentences; spelling: grassland

[AC] Corrected

line 26: Could mention that "down-scaled" means higher resolution

[AC] Sentence removed

line 27: Remove sentence and put in parenthesis in previous sentence to reduce wordiness of text: "The sites (shown in figure 1) were chosen...". Add reference to studies using these locations wind data.

[AC] Corrected

**2.2.2 ERA5**

page 5

line 2: ERA5 reference linked to https://cds.climate.copernicus.eu/cdsapp#!/home. Could instead use:

i.
```
\begin{small}
@misc{website:era5,
Author = {Hans Hersbach and Dee Dick},
Month = {November},
Title = {{ERA5} reanalysis is in production},
url = {http://www.ecmwf.int/en/newsletter/147/news/era5-reanalysis-production},
Year = {2016},
Publisher = {{ECMWF} - European Center for Medium Range Weather Forecast},
note = {last accessed: 22.10.2019}}
```

[AC] Cited as requested by ECMWF: https://confluence.ecmwf.int/display/CKB/ERA5

line 2: Add: reference to ECMWF

[AR] Is considered not essential after having cited ERA5.

line 7: "... is performed on the DOWA data..."

[AC] Modified

line 7: Remove: "As is explained later on"

[AC] Removed

**2.2.3 DOWA**

page 5

line 10: Remove: "The"; spelling of "downscaled" different from above.

[AC] Modified

line 13: Add "grid spacing" and "grid points" for clarity.

[AC] Rephrased

page 6

line 1: Add reference to improved DOWA performance.

[AR] The last sentence covers such a reference: Kalverla 2019

line 3: Clarify "routine weather stations"

[AC] the KNMI's network of automated weather stations.

**2.3 Clustering procedure**

**2.3.1 Prepossessing of the wind data**

page 6

line 11: Remove: "the" before "wind speed" similar to no "the" before "direction"

[AC] Removed

line 11: Remove: "the" after "Therefore", because general wind profiles and not specific ones.

[AC] Removed

line 13: Separate into 2 sentences. Explain what you mean by processed using its own properties.

[AC] Removed

line 14: Rewrite for clarity; for example: "The wind speed components are expressed as parallel and perpendicular components relative to their reference wind velocity at 100 m, similar to Kalverla et al. (2017) and Malz et al. (2020a), thereby making them independent of wind directions."

[AC] Rephrased, also taking into account comments of referee 1

line 18 + 19: Active voice and clarity; for example: "Wind velocity components are normalised using each profile's 90th percentile wind speed, because it reduces the amount of outliers in comparison to using the maximum speed of each profile. These normalised and decomposed samples are referred to as wind profile shape."

[AC] Rephrased

line 21: Use irregular,atypical or unconventional instead of eccentric, add: "...low wind speeds."

30    [AC] used irregular

line 24: Clarify: wind resource representation. Is this the wind speed probability distribution similar to a Weibull distribution? ;

[AC] removed

**2.3.2 Principal component analysis of the wind profile shape dataset**

35 page 6

line 27: Replace: "while" with "which"

[AR] Would result in incorrect language. I don't see why I could not use "while" here.

page 7

line 7: Replace: "layer" with "height"

[AC] Rephrased

line 14: Explain Fig 2 PC unit vectors. Are these the unit vectors in PC1 and PC2 domain which have a length not equal

5    to one in $\hat{v}$, $height$ domain? Is it that the length ($\sqrt{PC1_{\parallel}^2 + PC1_{\perp}^2} = 1$) of the parallel (orange) and perpendicular (blue) component at each height is one? Are the orange and blue lines the direction of the PC at each height? Add: $\hat{v}$ is normalised wind speed magnitude in text.

[AC] Explanation expanded

line 16 + 17: Word order: mostly characterises.

10    [AC] modified

line 18: What shows the large contribution of both PCs? Is it that the $\hat{v}$ magnitude is high? Does it make sense that most variance is top and bottom? I would expect that profiles within one cluster have similar wind speeds at high altitudes as they are probably driven by similar large scale weather phenomenon or is this lost due to normalisation and PCA?

15    [AC] A large PC coefficient. We explain it better in the text now.

[AR] We expect most variance at both ends. For a large part this is explained by the pre-processing approach that is used, i.e., the choice of the reference height.

"I would expect that profiles within one cluster have similar wind speeds at high altitudes as they are probably driven by similar large scale weather phenomenon or is this lost due to normalisation and PCA?" - I'm not sure why you would expect this. But indeed I would expect the correlation between the clusters and large scale phenomena is

20    weakened by the two steps in the preprocessing, so both dissecting the reference wind direction and normalisation.

line 20: Explain wind profiles shapes along PC1 and PC2 more. Why did you chose minus and plus one standard deviation as multipliers? What do you want to show with these profiles?

[AC] Explanation expanded

[AR] We want to illustrate how the PCs can be physically interpreted.

line 22: What are the eigenvalues of PCs? What are the retained PCs?

[AR] They follow from the PC analysis. It's believed to be beyond the scope of the paper to explain the details on this.

[AC] Left out sentence that mentions the eigenvalues.

line 32: add: "Figure 4g which shows onshore data will be ..."

[AC] modified

page 8

figure 2: What are the orange and blue dashed lines in column 2 PC1 and PC2? I think you did not explain $\hat{v}$ anywhere in the text or the caption. What does it mean that both PC1 and PC2 have negative parallel components below 300m?

[AC] Dashed lines were explained in last sentence. $\tilde{v}$ is now introduced in text, it is not deemed necessary to also state it in the caption. That the parallel wind speed component at this height decreases for an increase in the PC. The PC coefficients are now also better explained in the text.

**2.3.3   Choosing the number of clusters**

page 8

line 2: Replace: applied instead of employed

[AC] Changed

line 3: Replace: "...represented by its centroid ..."

[AC] Changed

line 9: Do wind profiles have such a structure?

[AC] No, expanded in text.

page 9

figure 4: Mention for offshore a) and onshore b) in caption. Put figs 2 and 11 in parenthesis? replace: "deviation away from their mean" ?

[AC] Corrected

line 1+2: Remove "Moreover". Add reference to elbow and silhouette method find appropriate k.

[AR] Methods are explained in next paragraphs so reference is deemed unnecessary.

[AC] Removed moreover

line 3: Replace: applied instead of employed. Can add on how evaluated: by comparing the estimated AEP

[AC] Replaced

line 5: Reference: appropriate choices of k?

[AR] Not deemed necessary

page 10

figure 5 a): Remove: "the" before cost function and cluster cohesiveness. The error between which data? mean wind speed error between each profile and its respective cluster centroid?

[AR] See text for error definition

[AC] "The"'s removed

figure 5: Add: "k-means" before clustering ; remove: point before (b)

[AC] Corrected

line 5: Maybe use scaling instead of denormalisation; remove "of the cluster to which it is assigned"; replace "of" with "used in pre-processing"

[AC] Corrected, kept de-normalisation as it is more informative than scaling.

line 7: Add : $\varepsilon_{ij}$ here to make equation easiert to understand. Define between which data the error is calculated.

[AC] Reformulated

line 8: Is the representation accuracy shown in Fig 5?

[AC] Yes, reference to figure added in text.

page 11

line : Is the error only based on different 5 $\Psi$ values?

[AR] Actually only the values for L are restricted to 5 values. $\Psi$ is a function of L and the height.

line 5: Sentence needed?

[AC] Reformulated

line 6: Add reference to $\Psi$ function; replace every with each

[AR] Introduced in introduction

line 14: Grammar; add (bottom) to Fig 5a

[AR] Grammar?

[AC] Added "The lower panel of Fig. 5a ..."

line 15: Is " Note that..." sentence necessary here? Maybe move to where you introduce WCSS?

[AC] Sentence removed

line 19: Why not just wind speed error? $\overline{\varepsilon_i}$ never used in any equation

[AC] Reformulated. Expression removed.

line 20: Replace top and bottom with above and below.

[AC] Added "... of the vertical grid."

line 21: Replace vertical grid with dataset or rewrite

[AC] Rewritten "... of the vertical grid points of DOWA."

page 12

line 5: Limited to what? Rather a fixed number of clusters.

[AC] Rephrased

line 6: What do you mean by: " ... and our aim to present a meaningful analysis and interpretation of the resulting clusters." and how did this affect your choice of k?

[AR] Using many clusters would make it hard to find relations between clusters and atmospheric phenomena manually.

[AC] Rephrased a little

line 8: Isn't it just the silhouette score and not the mean silhouette score in both sentences?

[AR] No, the mean is calculated over all samples in a cluster.

**2.4 Cluster wind resource representation**

page 13

figure 7: What does the asterisk mean? Maybe rename axis label to $PC1_{onshore}$ / $PC1_{offshore}$ ?

[AR] It refers to the average of the two locations.

[AC] Rephrased, hopefully that clarifies it better.

line 5: Even though clusters are not the same between on and offshore.

[AC] Rephrased a little

line 6: Replace: full with entire

[AC] Replaced

**2.4.1    Cluster representation for the offshore location**

page  13

30          line  9: Replace: of with at; add: their centroids

              [AC] Corrected

          line  15: Add: backwards transformed from PC space to normalised wind velocity

              [AR] This is implied by the definition of the cluster-mean wind profile shape introduced in Sec. 3.3.

          line  15: Explain: stability function is varied continuously?

35              [AC] Reformulated

page  14

          table  2: Replace: at with in?

              [AC] Rephrased

          line  3: Is this sentence about a) ? remove: is, and & represented " In contrast to the other panels, the absolute frequency
              on the y-axis serves to show which part of the total dataset is presented "

5               [AR] Rephrased

          line  6: Replace: for with of

              [AC] Replaced

          line  7: Where is "here"? Figure 9, 10? Isn't it only one distribution?

              [AC] Rephrased

10          line  9: Which data did you use? $L$ is always calculated using surface data (see: http://glossary.ametsoc.org/wiki/Obukhov_
              length)

              [AR] We infer $L$ from $\mathrm{Ri_B}$. The latter we calculate over an elevated layer. So the approximation of $L$ is not based
              on surface data. However, it is only an approximation.

page  16

15          figure  9: You could replace figure 9 and 13 with legend or text box in figure 8 and 12 to save space. Using a figure to show
              this seems unnecessary

              [AR] These plots were added to depict how the L found for each shape compares to the others, this information
              would be lost by using the approach suggested by the reviewer. This information is used throughout e.g. Sec. 4.2.

page  17

20      figure  10: Add a),b),c),d),e),f) to caption; Describe what is shown in a,b,c. What is the benefit of using this $v_{100}$ binning
        or what is the reason for having equal bin size for wind speed?

        [AR] Difference between a,b,c is easily inferred from the legend.

        [AC] Letters added.

**2.4.2   Interpretation of the offshore cluster representation**

25   page  18

        line : Maybe makes more sense to put interpretation in the same paragraph where figure is described? This way you
        have to flip back and forth a lot as

        [AC] The current order is used to have a nice flow of information.

        line  5: Add: magnitude of profiles? replace: well-described

30          [AC] Modified

        line  6: Replace: shear and veer

            [AC] Replaced

        line  8: Shorten: North-West direction

            [AR] Current text is more precise.

35      line  10: Equilibrium profile meaning a unstable shape?

            TBD

        line  25: Remove: "The frequent"; Rewrite: Winds at this location with a southerly component...

            [AC] Rewritten

        line  29: Remove: "... and gradually"

            [AC] Removed

     page  19

        line  1: Replace: "abrupt kink" with "sharp bend"

5           [AC] Replaced

        line  5: Replace: "rather than" with "and less often with"

            [AC] Rephrased "... more often for winds with a westerly than southerly component."

**2.4.3   Comparing the on- and offshore cluster representations**

     page  19

10       line 8: Rewrite: The onshore data at the met mast Cabauw is clustered using the same methodology.

        [AC] Rewritten

      line 9: Compared how and where?

        [AC] Rephrased

      line 12: Rewrite: "the mean profile shape below 200 m is in accordance with a stable logarithmic profile."

15         [AC] Rewritten

      line 15: Increased relative to what?

        [AR] The offshore mean shape as implied by context.

      line 16: Add reference to where they are plotted

        [AR] Referred to at the start of the paragraph

20       line 16: If they share the same coordinate system does that mean that both locations have the same PCs?

        [AR] No, the location average PCs were used for the coordinate system.

        [AC] Rephrased

      line 24: "...clustering algorithm..."; ".... for each cluster."

        [AC] Corrected

25       line 25: Replace: more or less with onshore closely resembles offshore

        [AC] Rephrased - align covers statement better

      line 28: Remove: again

        [AC] Removed

      line 29: Obukhov length determined how?

30         [AC] Rephrased

      line 30: Meaning? "show an increase in wind shear". Increasing wind gradient from 1 to 3?

        [AR] Yes, the reader is ought to understand at this point.

      line 34: Replace: "turning" with "rotation" and "kink" with "sharp bend"

        [AC] Rephrased

35  page 20

      line 4: Rephrase: The frequency distribution over the first five onshore clusters is more balanced than the offshore clusters which show one distinct dominant cluster.

        [AC] Rephrased

line 9: Rephrase: Convection occurs in the presence of daytime solar irradiation which leads to the development of well-mixed wind speed profiles.

[AC] Rephrased

line 12: Explain: "Patterns in times of occurrences" of what?

[AC] Added reference to figs.

line 13: Explain: Meaning of " almost identical bin distributions" if wind speed distribution is different?

[AC] Rephrased a little.

line 14: Explain: Reason for choosing wind speed bin limits this way?

[AC] Added "... thereby the distributions of the individual clusters are easily related to the uniform general distribution and compared among one another."

line 15: Rephrase: Total frequency of each bin is roughly the same over the entire dataset. Explain: What is the benefit of this approach?

[AR] See above

line 16: How can they show similar stability distributions if figure 9 and 13 are totally different?

[AC] Added more precise description: "In the case of the stability distributions, the onshore location shows a tendency to more stable conditions for all clusters."

line 22: Replace: "... start to be observed" with "are observed."

[AC] Replaced

page 21 + 22

figure 11+12: See same figures for offshore

[AR] No Figure 12 offshore equivalent (Figure 8) are given.

page 23

figure 13: Very strange that 6 out of 8 profile shapes are stable. Add to caption: fitted up to 200m

[AR] Not so strange: there is little diversity in the shape of the unstable profiles, therefore all the associated samples are grouped together by the clustering. Added more text on this in Sect. 4.3

page 24

figure 14: Same as for previous similar figure. Rephrase: "Frequency distributions by time of occurrence..."

[AC] Corrected

**2.4.4 Spatial frequency distribution of wind profile shape clusters**

page 25

line 2: Replace: "are" with "were"

[AC] Corrected

line 3: Replace: "for" with "from"

[AC] Corrected

line 10: Remove: "So"

[AC] Removed

line 12: Replace: "inherent" with "synonymous" Inherent turns the relationship between reducing error and increasing n clusters around.

[AC] Rephrased: "increasing the number of clusters reduces the error"

line 21:Can you quantify how much they "look alike"?

[AR] You could, but I don't think such a metric would be very intuitive.

line 22: 21.7% ?Reference Table 3 here already

[AC] Pushed introducing the table more forward.

line 27: Which terrain features? Elevation? Forests, cities?

[AC] "Terrain features" replaced by "orographic features"

page 26

table 3: How do you quantify similarity?

[AR] I don't.

page 28

figure 16: Mention in caption: Each sample of every grid point is assigned to the cluster with the closest centroid.

[AR] Should be clear from the text already.

**2.5 Efficient AWE production estimation using the cluster representation**

page 29

line 6: Add: power curve and AEP

[AR] I don't think this would help the reader understand. If so more explanation would be required as we actually promote using multiple power curves.

line 7: Replace: "winds higher up" with "winds at higher altitudes"

[AC] Replaced

line 11: Add: "wind speed distribution within the corresponding cluster"; Replace: "of the clusters" with "of each cluster"

[AC] Phrased as "Each power curve together with the corresponding cluster-specific wind speed distribution yields the AEP contribution of the respective cluster."

**2.5.1 Constructing the power curves**

page 29

line 13: Never heard "construction" in the context of power curves. Replace throughout the document with derive or determine ?

[AC] Using derive now.

line 15: Fix grammar (reel-out twice); mention change in angle of attack;

[AC] Rephrased

page 30

line 2: What is V3 kite?

[AC] V3 removed

line 4: Add: "average energy production"

[AC] Rephrased: "The proposed AEP estimation requires characterising the maximum mean cycle power for a large variety of wind conditions."

page 31

line 1: Replace: "are exceeded" with "would be exceeded"? Simplify sentence: "Reel-out" repetition in same sentence. Why is reel-out duration important?

[AC] Rephrased: "the reeling speed is kept zero as long as the tether force does not exceed its limit" and "During reel-out, the tether force should yield a high power, while increasing the fraction of time spent producing energy."

line 6+8: What are cycle settings? Are these the fixed constraint in table 5?

[AC] Rephrased: "Table 5 lists the cycle setting parameters, which are used as optimisation variables, together with their respective limits."

line 15-18: a bit wordy, could be shortened to 1 or 2 sentences.

[AC] Rephrased

line 17: Add "reference wind speed"; Remove: "to yield the absolute wind profile"

[AC] Removed

table 5: Is tether diameter and associated drag considered in the model?

[AR] Yes, fixed as it is a design parameter.

line 20: Replace: "smallest" with "lowest"; "whole" with "entire"

[AC] Replaced

page 32

line 9: Does this intersection depend on tether length?

[AR] Yes, this is implied by relating the intersection height to the kite height earlier.

figure 18: Replace: "from " with "in"; also reference which section describes these profiles in fig 12; Add: "pre-determined" before cut-in

[AC] Modified

line 12: Simplify with active voice: "The depicted trajectories highlight operational changes occurring at different reference wind speeds"; Specify how the approach / attitude changes. Is this the reason for the strange $v_{ref}$ in legend of Fig 19?

[AC] Rephrased

[AR] The remaining of the paragraph describes these approach changes.

line 13: The fact that the tether length constraint is always active indicates that the global optimum is beyond this constraint. Can you comment on whether you tried out different settings or what the reasons could be for maxing out this constraint?

[AR] The constraint was chosen as it is a representative maximum value for the Kitepower system. The use of a different maximum was not extensively assessed.

line 14: Same is true for lower wind seeds though?!

[AC] replace "for" with "below"

line 16: criterion is also a constraint?

[AC] Yes, rephrased

line 19: They are similar. Are they cubic as we would expect from $P \sim \rho v^3$

[AR] Doesn't appear to be cubic.

page 33

figure 19: Add: "same normalised wind speed profile shape"; Approach = attitude? Specify what you mean by change in approach.

[AC] Added "cluster-mean wind profile shape"

[AR] Explained in text

figure 20: Frequency better in %; Add: "scaled cluster-mean..."; Why did you aggregate 4 bins into 1? Not sure if you have to mention this here.

[AC] Added "scaled"

[AR] It's only for illustrative purposes, therefore I felt that the figure was the right place to mention this.

line 1: LLJ benefit because of high wind speed at low elevation angle hence low cosine losses?

[AR] In the case of a LLJ and a fixed minimum and maximum tether length (which is more or less the case in my study as the pumping tether length coincides with its upper bound for all wind speeds), the kite does not necessarily see higher wind speeds for an increase in elevation angle, in contrast to monotonic profiles. The cosine loss does not play a big role here, as we do not observe a tendency to increase the reel-out power by increasing the elevation angle (at least not at low wind speeds). At a certain point the elevation angle is increased, but this is driven by keeping the duty cycle high.

**2.5.2 Estimating the annual energy production**

page 34

line 2: Replace "Constructed" with "derived" ?; Grammar: "... are used to calculate the average ..."

[AC] Replaced

line 4: Is $P_i$ the power curve or power?

[AC] Replaced by "maximal mean cycle power"

line 8: Is the numerical error so significant that you need to use 100 bins instead of the way more common 1m/s or 0.5m/s bins?

[AR] It is significant for the AEP convergence analysis.

line 10: Isn't it that you just use the reference wind speed $v_{100m}$ to calculate the frequency and they make up these bins?

[AR] No the frequency is determined on the basis of the normalisation wind speeds.

[AC] Rewritten paragraph

line 19: What do you mean by "...inaccuracy of the cluster wind resource representation"? Does it mean that fewer clusters lead to higher inaccuracies because of averaging or misrepresentation of the entire wind resource?

[AC] Yes, however, I wanted to emphasize here that it's not just the wind resource that introduces errors, therefore rephrased the sentence.

line 22: Rewrite: It's hard to understand how MMC and ML relate to each other and what 16 and 32 clusters mean. Does it mean that 16 MMC clusters have the same difference to converged value as 32 ML clusters? Doesn't this indicate that you need twice as many clusters to achieve similar quality using the ML approach? Which makes sense as it combines on and offshore locations and therefore different flow regimes?

[AC] Rephrased: "The AEP error at 16 clusters for the MMC representation is similar to the AEP error at 32 clusters for the ML representation, which suggests that the ML representation needs twice the number clusters to yield the same accuracy as the MMC representation."

line 28: Reference figure 20 here again. 50 optimisations per power curve because of the step size of $\Delta v_{scale}$ you chose? Maybe mention step size for clarification.

[AC] Rephrased

line 29: Remove of: "...half the number"; Replace; ".. can be used" with "yield similar results with half the computational cost"

[AC] Corrected

line 30: Grammar: "used for generating" with "used to generate"; Rewrite "in comparison to brute force, where 8760 optimisations ...."

[AC] Rewritten

page 35

line 1:"orders of magnitude lower"

[AC] Corrected

figure 21:"Comparison of AEP conversion between MMC onshore and ML offshore"

[AR] I believe "conversion" should be "convergence"

[AC] Rephrased "Comparison of the AEP convergence with increasing number of clusters for the onshore reference location using the MMC and ML cluster wind resource representations."

**2.5.3   Conclusion**

page 35

line 1: Remove: "have"; add: "a set of normalised wind profile shapes"

[AC] Removed "have"

[AR] Shape already implies normalised

line 6: Did you quantify the occurrence of LLJs? There are also other ways of doing that. You approach allows the inclusion of wind profile shapes into the wind resource description (equivalent to wind speed distribution/ Weibull for conventional turbines).

[AC] Rephrased

line 7: Grammar: "We demonstrated this methodology for two reference locations on and offshore based on the DOWA dataset."

[AC] Changed

line 9: What do you mean by "expressed in terms of wind velocity at 100 m?" and where did you do it?

[AC] Rewritten

line 10: "... profile shape variance."

[AC] added "... in the dataset"

line 15: Rewrite: "The DOWA dataset is partitioned using k-means clustering. The resulting cluster-mean wind profile shapes are used to represent the wind resource, thereby reducing the wide range of wind conditions to a reasonable number of wind profile shapes."

[AC] Rewritten

line 18: Rewrite: "Although some variability is lost by only using the mean cluster profiles, ..."

[AC] Sentence removed

line 21: Order: "... 8 offshore clusters show 3 monotonic ..."

[AC] Not sure what's meant exactly, but rewritten a bit.

page 36

line 1: Replace: "entire DOWA domain"

[AC] Replaced

line 3: Replace: "... between the profile shape and terrain."

[AC] Replaced

line 3: Replace: "... in contrast to 8760 for an ..."

[AR] "against" works here

line 16: Grammar: "...ML enables an assessment..."

[AC] Corrected

line 18: Replace: "...in estimating the AEP..."

[AC] Replaced

**Response to 2$^{\text{nd}}$ review of referee #2**

Mark Schelbergen, Peter C. Kalverla, Roland Schmehl, and Simon J. Watson

Thank you for another useful list of comments. We feel that they were very helpful for removing the last flaws in the paper. The most important changes made to the paper based on the comments of referee #1 are:

1. The profiles in the 2nd column of Fig. 2 and 11 are now better explained.

2. Section 4.3 includes an explanation on why only 1 cluster groups the unstable profile shapes.

3. A lot of unclear relationships in the text are clarified.

We respond to the referee comments by including our answers below the original comments. Our answers are preceded by one or both of the following labels:

1. [AR] = author's response

2. [AC] = author's changes in manuscript

**Referee comments**

1. I am confused by the use of parentheses as a way to soften the meaning of words or to make some of the words feel "optional". According to my understanding such use of parentheses is not considered good style. Please consider removing the parentheses in sentences such as P18L16 "we also see frequent (very) stable conditions". I think that here "we also see frequent stable or very stable conditions" is much more precise.

   [AC] Agreed, parentheses used in this way are removed.

2. The overbar in formula (5) is hard to see. Please check that this gets corrected in the final proof.

   [AC] Increased spacing between overbar and the fraction deliminator.

3. I would suggest adding a subscript for the symbol for the reference height z in (6), to avoid confusing that with the z in formula (2), where z has the meaning of a coordinate.

   [AC] We use $\bar{z}$ now instead.

4. I think that some additional sentences are still needed in describing the sample normalization procedure (P6L16-L19), because I still struggled to understand the details. I would suggest explicitly defining reference wind velocity for each

sample as the wind speed and direction of that sample at 100 m height, explicitly noting that each sample has its own reference wind velocity, because my first association is that "reference" is something that is common for all samples. I would suggest rephrasing P6L17 as "the value of the perpendicular component of the wind speed profile is 0 for each sample and for all averaged profiles".

[AC] Rephrased, now only referring to a reference height in stead of a reference velocity: "... the wind speed components are expressed as parallel and perpendicular components relative to the wind velocity at a reference height, which we have chosen to be 100 m. As a result, the value for the perpendicular wind speed at 100 m is zero and the reformatted wind profile is independent of the wind direction at 100 m."

5. Caption of Figure 8. I would suggest: "In the hodograph, the lowest level is indicated by the dotted line connecting the lowest level to the origin of coordinates".

[AC] Rephrased to: "In each hodograph, the lower end of the profile is indicated by the dotted line connecting the lowest height point to the origin."

6. I would rephrase the statement that MMIJ-5 has "relatively large deviations from logarithmic profile" P9L21, because later authors state that its shape is well described by unstable logarithmic profile, similarly in the caption for Figure 8 I would like it to be pointed out that these are stability adjusted logarithmic fits. I am stressing this point because a slightly distracted reader might confuse what exactly is meant by "logarithmic profile" because in some cases this term describes only the neutrally stable profile shape.

[AC] Replaced by: "Because wind speed increases monotonically with height in the logarithmic wind profile relationship, it can not describe these type of profile shapes.". Specified in the captions of figs. 2, 8, 11, and 12 that non-adiabatic log profiles are fitted.

**Clustering wind profile shapes to estimate airborne wind energy production**

Mark Schelbergen[1], Peter C. Kalverla[2], Roland Schmehl[1], and Simon J. Watson[1]

[1]Faculty of Aerospace Engineering, Delft University of Technology, Kluyverweg 1, 2629 HS Delft, The Netherlands
[2]Meteorology and Air Quality Section, Wageningen University, PO box 47, 6700 AA Wageningen, The Netherlands

**Correspondence:** Mark Schelbergen (m.schelbergen@tudelft.nl)

**Abstract.** Airborne wind energy (AWE) systems harness energy at heights beyond the reach of tower-based wind turbines. To estimate the annual energy production (AEP), measured or modelled wind speed statistics close to the ground are commonly extrapolated to higher altitudes, introducing substantial uncertainties. This study proposes a clustering procedure for obtaining wind statistics for an extended height range from modelled datasets that include the  variation of the wind speed and direction _with height_. K-means clustering is used to identify a set of wind profile shapes that characterise the wind resource. The methodology is demonstrated using the Dutch Offshore Wind Atlas for the locations of the met masts IJmuiden and Cabauw, 85 km off the Dutch coast in the North Sea and in the centre of the Netherlands, respectively. The cluster-mean wind profile shapes and the corresponding temporal cycles, wind properties, and atmospheric stability are in good agreement with literature. Finally, it is demonstrated how a set of wind profile shapes  _is_ used to estimate the AEP of a small-scale pumping AWE system located at Cabauw, which  _requires_ the derivation of a separate power curve for each _wind profile_ shape. Studying the relationship between the  _estimated_ AEP and the number of  _site-specific_ clusters used for the calculation shows that the  _difference in AEP relative to_ the converged value _is less than three percent_ for four or more clusters.

**1 Introduction**

Airborne wind energy (AWE) systems  _employ_ tethered flying devices to harness energy  _above the operational height range of_ tower-based wind turbines. _Typically these devices operate_ above 150 m (Malz et al., 2019; Salma et al., 2019), where wind is generally stronger and more persistent  _than in the surface layer. To estimate the_ annual energy production (AEP), _measured or modelled wind speed statistics close to the ground are commonly extrapolated to higher altitudes to obtain the_ wind speed statistics in the full operational height range of the AWE system  using either the wind profile power law or the logarithmic profile (e.g., Heilmann and Houle, 2013). This way of representing the wind resource introduces substantial uncertainties since the aforementioned wind profile relationships are not strictly valid beyond the surface layer. Moreover, within this layer, not all wind profiles can be described well with these relationships.

The power law is a simple empirical relationship which can be used to relate the wind speed $\underline{u}~v$ at one height $z_1$ to that at a different height $z_2$ and has the form:

$$\underline{u}v(z_2) = \underline{u}v(z_1)\left(\frac{z_2}{z_1}\right)^{\alpha} \quad , \tag{1}$$

where $\alpha$ is an empirical shear exponent factor related to the surface properties. The power law is normally applied up to around
5 100–200 m (Peterson and Hennessey, 1978), and does not offer enough flexibility to describe the variety of measured wind profiles (e.g., Park et al., 2014).

The logarithmic wind profile is frequently used to estimate the variation in wind speed with height over a flat surface. This profile is based on physical arguments and a form of the profile has been well established based on Monin-Obukhov similarity theory (Monin and Obukhov, 1954). In this non-adiabatic form, the mean wind speed $\underline{u}~v$ at height $z$ is given by:

[revised manuscript text omitted]